# A net-zero emissions strategy for China's power sector using carbon-capture utilization and storage

Jing-Li Fan [1,2], Zezheng Li[1,2], Xi Huang[1,2], Kai Li [1,2], Xian Zhang [3] ✉, Xi Lu [4,5], Jianzhong Wu [6], Klaus Hubacek [7] & Bo Shen [8]

Decarbonized power systems are critical to mitigate climate change, yet methods to achieve a reliable and resilient near-zero power system are still under exploration. This study develops an hourly power system simulation model considering high-resolution geological constraints for carbon-capture-utilization-and-storage to explore the optimal solution for a reliable and resilient near-zero power system. This is applied to 31 provinces in China by simulating 10,450 scenarios combining different electricity storage durations and interprovincial transmission capacities, with various shares of abated fossil power with carbon-capture-utilization-and-storage. Here, we show that allowing up to 20% abated fossil fuel power generation in the power system could reduce the national total power shortage rate by up to 9.0 percentages in 2050 compared with a zero fossil fuel system. A lowest-cost scenario with 16% abated fossil fuel power generation in the system even causes 2.5% lower investment costs in the network (or $16.8 billion), and also increases system resilience by reducing power shortage during extreme climatic events.

Decarbonization of energy systems, especially the power system that accounts for up to 39.6% of global carbon emissions[1], plays an important role in mitigating climate change. The power system will likely experience a profound transformation to achieve zero carbon emissions in the future. The latest Sixth Assessment Report of the United Nations Intergovernmental Panel on Climate Change (IPCC) states that "in pathways limiting climate warming to 1.5 °C, almost all electricity will need to rely on the supply from zero- or low-carbon sources in 2050, such as renewables or fossil fuels with carbon-capture and storage, combined with increased electrification of the energy demand"[2]. With the rapid decline in the cost of renewable power generation[3], an extremely high proportion of renewable or even 100%

renewable energy, such as solar photovoltaic (PV) and wind power, has been widely considered an effective method for future net-zero carbon power systems[4,5].

Although wind and solar resources are widely available with low operating costs, their intermittent nature seriously threatens the stable and reliable electricity supply[6,7]. To mitigate this risk, energy storage must be widely deployed[8]. Moreover, renewables are usually unevenly distributed, and electricity load centers are often located far from supply sources, especially in large economies such as China[9,10]. As a result, there is an urgent need to build long-distance high-voltage infrastructures[11,12] to transmit surplus power from resource-rich areas to electricity demand centers. From a spatiotemporal perspective, a

[1]Centre for Sustainable Development and Energy Policy Research, School of Energy and Mining Engineering, China University of Mining & Technology, Beijing 100083, China. [2]State Key Laboratory of Coal Resources and Safe Mining (China University of Mining and Technology), Beijing 100083, China. [3]The Administrative Centre for China's Agenda 21, Ministry of Science and Technology, Beijing 100038, China. [4]State Key Joint Laboratory of Environment Simulation and Pollution Control, School of Environment, Tsinghua University, Beijing 10084, China. [5]Institute for Carbon Neutrality, Tsinghua University, Beijing 100084, China. [6]School of Engineering, Cardiff University, Cardiff CF24 3AA, UK. [7]Integrated Research on Energy, Environment & Society (IREES), Energy and Sustainability Research Institute Groningen (ESRIG), University of Groningen, Groningen 9747 AG, the Netherlands. [8]Energy Technology Area, Lawrence Berkeley National Laboratory, Berkeley, CA 94720, USA. ✉e-mail: zhangxian_ama@163.com

100% or near-100% renewable power system may incur higher costs due to the high investment in energy storage[13] and high-voltage infrastructures[14,15], otherwise suffering from a low reliability (defined as the degree to meet the ideal electricity demand under normal circumstances, with 99.9% as the current standard for Chinese cities)[4,9,11,12,16]. In addition, since wind, solar and hydropower are all climate or weather-sensitive, renewable power generation is generally regarded as one of the most vulnerable sectors to weather extremes[17], threatening resilience (defined as the degree to meet the ideal electricity demand during weather events[18,19]; see "Methods" section) of a 100% or near-100% renewable power system.

A high share of renewable power generation combined with fossil fuels involving carbon capture, utilization, and storage (CCUS) could be an alternative to 100% renewable power in the absence of a sufficient storage capacity and interprovincial transmission to ensure deep decarbonization of the future power generation system[20,21] and maintain system reliability and resilience. First, as one of the firm low-carbon electricity sources (e.g., nuclear power, hydropower, coal-fired power with CCUS, and natural gas-fired power with CCUS)[22], abated fossil fuel power generation with CCUS in high-renewable power systems could partially replace variable renewable energy and lower the associated need for the construction of energy storage or high-voltage infrastructures to improve the reliability of power systems. Second, in contrast to other low-carbon power, fossil fuel power generation with CCUS is less vulnerable due to its stable thermal supply and flexibility to generate power as needed[23]. For instance, the electricity supply obtained from many nuclear and hydropower plants in France was replaced by natural gas-fired power due to the European drought in the spring and summer of 2022. Third, although CCUS currently remains expensive with a global $CO_2$ capture capacity of only 36.6 Mt per year in 2021[24], its growth has been evident in recent years, with the number of demonstration projects under development or operation worldwide growing from 43 in 2018 to 136 in 2021[24,25]. This is reflected in the considered IPCC scenarios, with almost all integrated assessment model (IAM) scenarios incorporating CCUS under limiting global warming to 1.5 °C or 2 °C relative to preindustrial levels[26,27], as the CCUS option generally yields lower costs in reducing carbon emissions than nuclear and renewable options under these scenarios[13,22,28] and provides a viable solution for carbon lock-in of fossil fuel energy infrastructure[7,29,30], stranded assets, and industry employment losses[31,32], although previous research has considered IAM-specific modeling assumptions (e.g., the application of general equilibrium theory-based IAMs)[33]. Therefore, it is important to quantify the carbon emission reduction effectiveness of the high-renewable power system combined with abated fossil fuel power generation involving CCUS via a comparison to the 100% renewable power system, especially from system reliability and resilience perspectives.

In previous research, the significance of 100% renewable or fossil fuel power generation with CCUS in future low-carbon power systems has been investigated separately. Certain scholars have emphasized the feasibility and reliability of 100% (or near-100%) renewable power systems[34] at the global[9,11,35] or national level (e.g., USA[4,12] and Germany[34]). For instance, Bogdanov et al.[35] stated that a 100% renewable electricity system in 2050 is both technically and economically feasible for all regions worldwide; Dowling et al.[11] highlighted that long-duration storage (>10 h) could reduce the cost of 100% reliable wind–PV cell systems; and Brown and Botterud[4] and MacDonald et al.[12] demonstrated that interstate high-voltage transmission expansion is necessary to achieve a decarbonized power system dominated by renewables in the US at a lower cost. In contrast, IAM-based research (e.g., Rogelj et al.[36], Jacobson et al.[37]), in which studies depend on underlying structural constraints[33], bottom-up industry models[13,21,22], and power system optimization models[38,39], has highlighted the inevitability of fossil fuel power generation with CCUS as a complement to renewables for deep decarbonization of the power system

from an economic perspective. However, few researchers have compared the overall cost-effectiveness performance of the 100% renewable system to that of the system with a high share of renewables combined with abated fossil fuel power generation under the same system modeling framework to obtain a reliable and resilient near-zero power system, except for limited analysis from a single reliability[10,40] or resilience[41,42] perspective or comparative perspective of two individual technologies rather than under the same power system model framework[43]. More importantly, all power system optimization models in earlier studies generally lacked detailed facility and geological constraints of abated fossil fuel power plants, with notable exceptions incorporating high-resolution temporal features of variable renewable power[10,40,44].

As a country rich in coal resources, China hosts >50% of the world's coal-fired power generation capacity[45], thereby emitting 37% of global power sector carbon dioxide emissions[46]. In 2020, China committed to achieving carbon neutrality by 2060 and set a target to reach a nonfossil energy consumption proportion of 80% by then[45]. Decarbonizing the power sector in China is vital for both global climate mitigation and achieving its carbon neutrality goal. Moreover, due to the unique situation in China in terms of economic development[47], renewable endowment[9] and geological storage potential of $CO_2$[48], power system strategies for other counties[4] are not directly applicable. Several recent modeling studies on China's power system have achieved numerous advances, such as improving the resolution from yearly to hourly electricity supply–demand systems and from national to provincial levels[10,40]. However, they ignored the availability constraints of fossil fuel power generation with CCUS, whose potential could be high but greatly dependent on the distribution of geophysical conditions[49–51]. In these studies, the hourly power was aggregated into only dozens of categories due to the high computational complexity, which may cause biased results.

In this paper, we constructed a high-resolution integrated power system assessment model considering the hourly electricity supply–demand balance by combining hourly variable renewables that vary across provinces with geologically constrained fossil fuel power generation involving CCUS, as well as energy storage and long-distance power transmission, and then applies the established model in the design of the future decarbonized electric power system architecture in China and 31 provinces (except for Hong Kong, Macau, and Taiwan) in 2050. This model integrates six interlinked modules (see Methods and Supplementary Fig. 1): (1) an hour-by-hour prediction model for the electricity demand in 31 Chinese provinces in 2050; (2) an hour-by-hour estimation model for the solar PV and wind power generation potential in 31 Chinese provinces; (3) a CCUS source–sink optimal matching model for retrofitting the existing fossil fuel power plants in China; (4) an integrated simulation model configured with the hourly power system supply–demand balance in 31 provinces in 2050, which specifies energy storage duration and interprovincial power transmission capacity in combination with $CO_2$ geological storage-limited fossil fuel power generation involving CCUS; this model was used to analyze the reliability (i.e., by deducting the electricity shortage rate from 100%) of 10,450 combination scenarios under different storage durations, transmission capacities and shares of fossil fuel power generation with CCUS in 2050 (Supplementary Fig. 2); (5) a cost-competitive analysis model for the decarbonized power system in 2050, which was used to identify the lowest-cost power mix; and (6) a simulation model for the impact of representative weather extremes (snowstorms, sandstorms, droughts, and heat waves) on power generation and corresponding power shortages, which was used to analyze the resilience of the future power system.

We highlight three major findings. (1) A high proportion of renewables combined with fossil fuel power generation involving CCUS in 2050 could offset the transmission capacity and short-term storage requirements, resulting in a lower cost to achieve a certain

power shortage rate (or power system reliability) relative to a zero-fossil fuel power generation system. Specifically, to achieve the lowest national total power shortage rate (the ratio of the unmet electricity demand to the ideal demand) of 0.07% with a zero-fossil fuel power generation system at five times the reference transmission level and 12 h of energy storage, 20% abated fossil fuel power generation with CCUS only requires a 3.5 times higher transmission level with 8 h of energy storage, corresponding to a 3.0% decrease in the levelized cost of energy (LCOE) relative to a zero-fossil fuel power generation system. (2) As the penetration rate of abated fossil fuel power generation technology involving CCUS increases from 0%–20% in the 2050 power generation system (as an integer), the system cost corresponding to certain reliability of 99.9% would first decrease from $679.2 (or an LCOE of 46.78 $/MWh) to $662.4 billion (or an LCOE of 45.64 $/MWh) (or by 2.5%) and then increase to $663.2 (or an LCOE of 45.69 $/MWh), with the lowest-cost power system configuration typically involving 16% abated fossil fuel power generation. (3) A high-renewable power system combined with 16% abated fossil fuel power generation involving CCUS (i.e., the lowest-cost electric power system architecture based on our 10,450 scenarios) remains more resilient under extreme weather conditions than a zero-fossil fuel power system; and if historical snowstorms, sandstorms, droughts, and heat waves were to again occur in China, power shortages in affected regions would be 54%, 56%, 57% and 68% lower, respectively, under the abated fossil fuel power generation scenario than under the zero-fossil fuel power generation scenarios. This study provides an important reference for the design of economical, reliable, and resilient near-zero power systems worldwide.

## Results

### Unmet electricity demand in a zero-fossil fuel power system

By 2050, the nonfossil energy (onshore wind, offshore wind, solar PV, hydropower, and nuclear) power generation potential (equal to the sum of the corresponding hourly maximum power output potential values) in China will reach 90,076 billion kWh, of which variable renewables (solar and wind power in this study) will account for 96% or 6.2 times the total projected electricity demand (as expressed in Eq. (1)) for that year (Fig. 1i, Supplementary Fig. 3a). When the power generation potential is accounted for separately from the perspective of each province, i.e., ignoring the electricity supply from inter-provincial transmission, the electricity supply–demand balance in China widely varies across provinces, with the supply-to-demand ratio ranging from 0.21-254.6 and the total unmet electricity accounting for 18.1% of the national demand (determined with Supplementary Equation (16)). Specifically, 17 of 31 provinces in China will exhibit a higher nonfossil power generation potential than their electricity demand by 2050, with supply-to-demand ratios ranging from 1.04–254.6, while the remaining 14 provinces will exhibit a deficit, with supply-to-demand ratios ranging from 0.21-0.98 (Supplementary Fig. 3b). For instance, Guangdong, Shandong, and Jiangsu will achieve the highest electricity consumption in 2050, accounting for 26% of the national total amount, while their nonfossil power generation potential will only account for 3% of the total nonfossil power generation potential of 31 provinces, indicating high electricity supply deficiency risks. Xinjiang, Inner Mongolia, and Tibet, in contrast, will reach the highest nonfossil power generation potential (72% of the national total amount), but their electricity consumption will account for only 8% of the national total amount in 2050, resulting in a substantial electricity supply excess.

Further comparing the hourly nonfossil power output to the disaggregated hourly electricity demand without power transmission and energy storage, China could experience a national total power shortage rate of up to 35.4% (the sum of the power shortages in 31 provinces as a share of the national ideal electricity demand; the lower the power shortage rate, the higher the system reliability; see "Methods" section

for a detailed definition), and all provinces could face power shortages, ranging from 0.4%-81.5% across the 31 provinces (Supplementary Fig. 3b). This indicates that simply aggregating the hourly nonfossil power output could result in considerable underestimation of electricity supply shortages, especially in areas with high electricity consumption. For instance, even when several provinces are aggregated into regions, such as Beijing-Tianjin, East Coast, and South Coast, the hourly variable renewable power output is overall much lower than the electricity demand (Fig. 1c, d, e), resulting in 8760, 7860, and 5673 h of power shortages (the sum of the hours with a lower nonfossil power generation than the electricity demand), respectively. Other regions, Northeast, Northwest, and Southwest China, could also suffer 1680, 780, and 841 h of power shortages, respectively, although they jointly account for 87.4% of the total wind power generation potential and 95.0% of the total solar power generation potential in China (Fig. 1b, f, h, respectively).

Electricity storage and power transmission, particularly their combination, could potentially lower provincial power shortages as well as national total power shortages (defined in Eq. (14)). On the one hand, cross-provincial or cross-regional power transmission in China, e.g., large-scale west–east power transmission lines, could assist in balancing spatial variations in the renewable electricity supply. For this reason, China built or planned interprovincial power transmission infrastructures with a total capacity of ~385.6 GW by 2021, including 31 ultrahigh-voltage (UHV, i.e., ±800 kV direct current and 1000 kV alternating current) lines and hundreds of other high-voltage lines (i.e., 1 kV-750 kV)[10,52,53]. By 2050, maintaining a zero-fossil fuel power system with the current transmission capacity in the absence of energy storage could reduce the national total power shortage rate in China from 35.3% to 28.1%. Measurement of the enhanced reference transmission capacity (employed as the reference scenario in this study) by artificially adding 35 UHV infrastructure channels to the current transmission capacity (Supplementary Table 1) could result in a national total power shortage rate of 21.8%, which could be further reduced to 11.8% with a fivefold increase in the interprovincial reference transmission capacity (Fig. 2a). On the other hand, short- or long-term energy storage (e.g., the use of low-cost flow batteries, Li-ion batteries, compressed air energy storage, pumped hydroelectric storage, and hydrogen energy storage[8,11]), particularly in renewable resource-rich areas, could stabilize intermittent local wind power and solar PV energy, despite most exhibiting a lower technology readiness level[54], higher cost (Supplementary Table 2), greater geographic limitations, or lower installed capacity advancement than other low-carbon technologies such as nuclear and energy efficiency improvement. For instance, under the reference transmission scenario, allowing a maximum short-term storage capacity of 6 h could reduce the national total power shortage rate from 21.8% to 7.2%, and allowing 12 h of capacity could further reduce this value to 6.6% (Fig. 2a).

Nevertheless, it is challenging to overcome power shortages with either short-term energy storage or power transmission in a zero-fossil fuel power system. Specifically, the national total power shortage rate based solely on the maximum power transmission capacity and short-term energy storage, i.e., 11.8% at up to five times the reference transmission level and 6.7% at 12 h of storage (Fig. 2a), respectively, remains well above the national critical standard for electricity supply reliability (the degree of the electricity supply meeting the electricity demand, determined as 1 minus the power shortage rate) in typical Chinese cities (99.9% or 0.1% in terms of the national total power shortage rate). The lowest national total power shortage rate of 0.07% could be obtained when the maximum short-term energy storage and power transmission are fully utilized concurrently (Fig. 2a), but this may incur very high economic costs due to the very high capital investment cost of transmission and storage infrastructures, with relevant levelized electricity supply costs, power

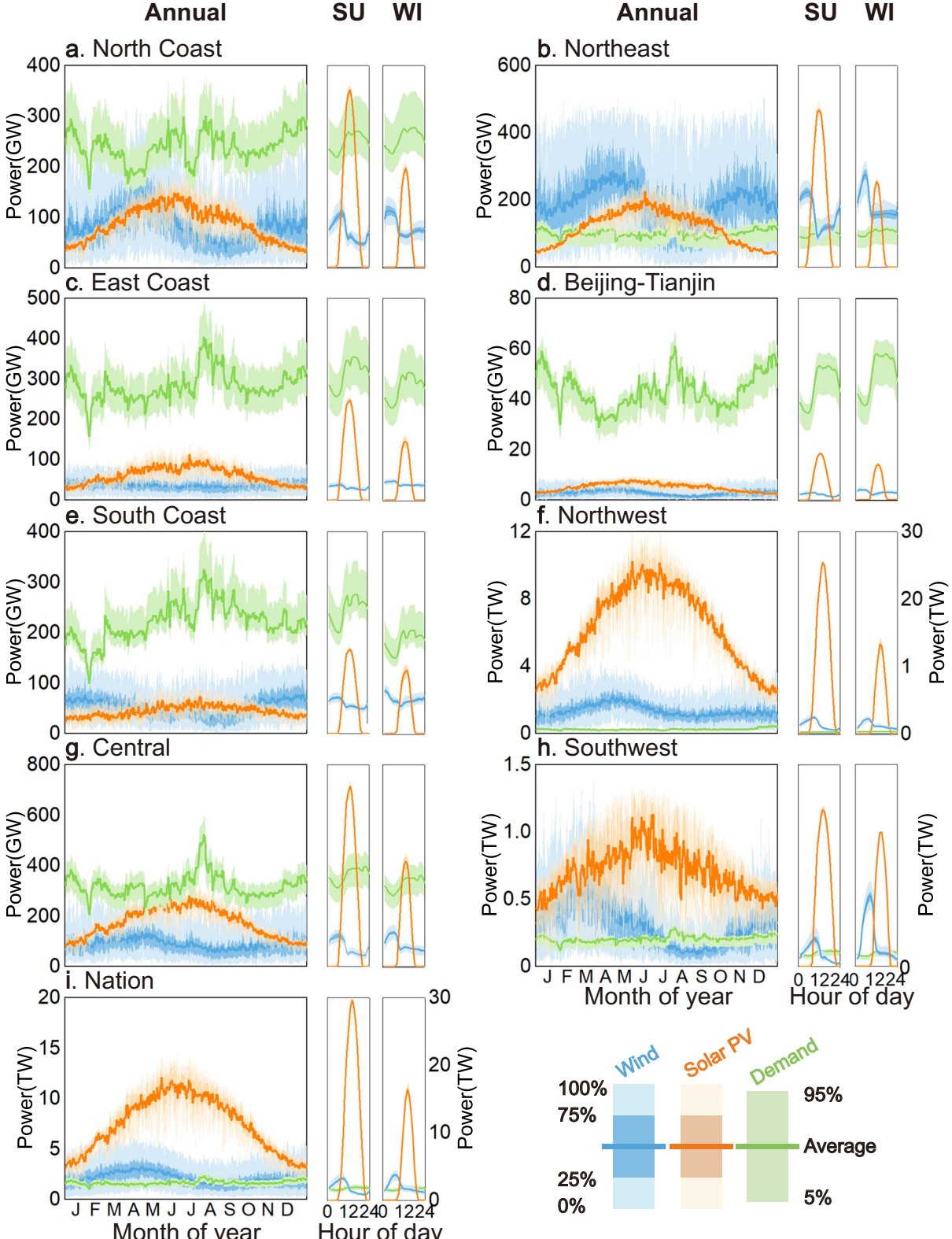

**Fig. 1 | Daily and hourly variabilities of the wind power and solar PV generation potentials and the predicted electricity demand in 2050.** Each panel covers a different region: **a** North Coast; **b** Northeast China; **c** East Coast; **d** Beijing-Tianjin; **e** South Coast; **f** Northwest China; **g** Central China; **h** Southwest China; **i** Nation (the sum of 31 provinces). The cyan and orange curves in each panel denote the wind power and solar PV generation potentials, respectively, and the green curves in each panel denote the predicted electricity demand in each region in 2050. SU and WI denote summer and winter, respectively. The left-to-right columns for each region show the daily variabilities of the power generation potential and predicted electricity demand in the entire year, and the corresponding hourly variabilities in summer (June, July, and August) and winter (December, January, and February). The lines indicate the mean values, the dark shading indicates the inner 50% range (25th to 75th percentiles), and the light shading indicates the outer 50% range (0th–100th percentiles). The samples of power generation potentials are from 1980–2019 for wind and 2010–2019 for solar PV.

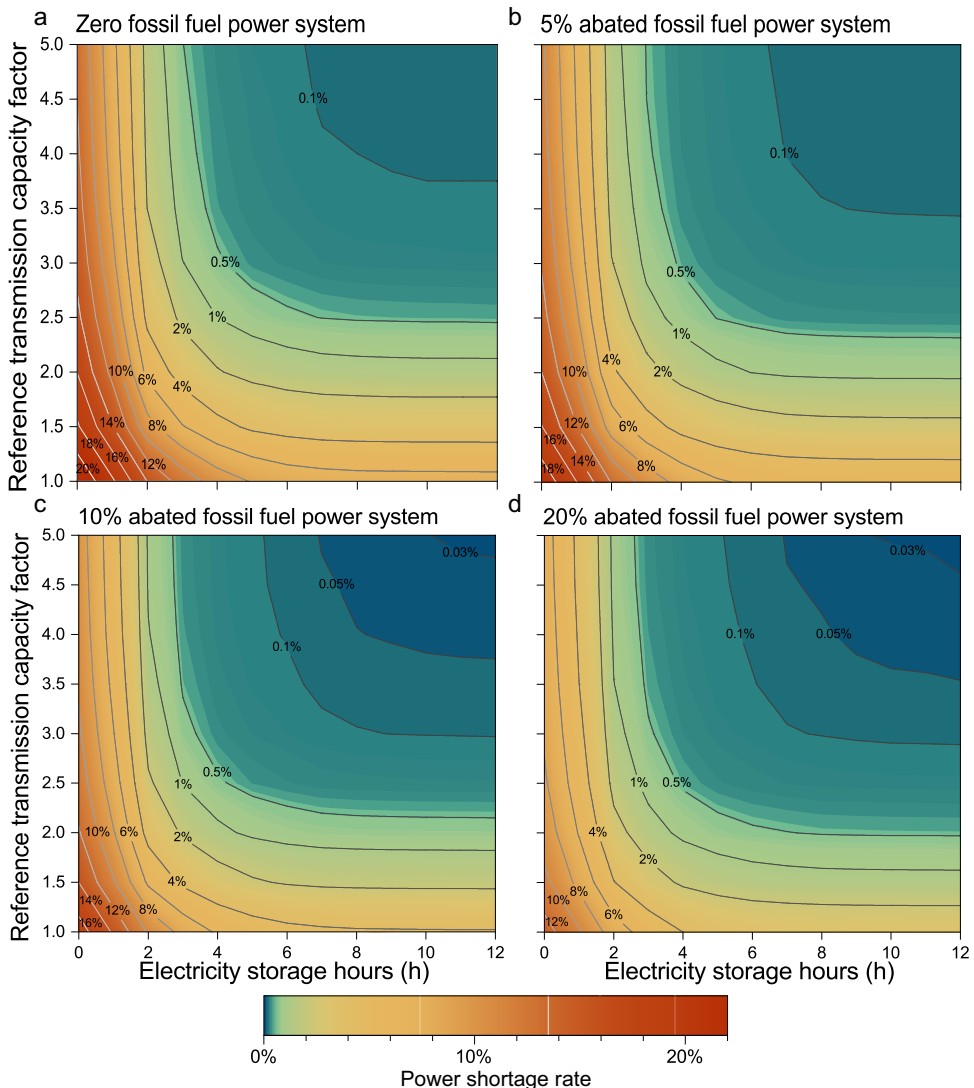

**Fig. 2 | Relationship between the national total power shortage rate, transmission capacity, and short-term energy storage duration.** The impact of the energy storage duration and transmission capacity on the national total power shortage rate in China in 2050 is explored by considering 10,450 scenarios with 0-24 h of short-term energy storage, 1–10 times the reference transmission capacity (0.5 intervals), and 21 abated fossil fuel share scenarios (0%-20% at 1% intervals) or zero-fossil fuel power generation with long-term energy storage. Only the results associated with durations of 0-12 h and 1-5 times the reference transmission capacity are shown in this figure, while four representative shares of abated fossil fuel power generation, namely, 0%, 5%, 10%, and 20%, are included (the long-term energy storage scenario is shown in Supplementary Fig. 6). The figure shows the national total power shortage rates for the various combinations of the transmission capacity and short-term energy storage duration, with the share of abated fossil fuel power generation varying in each panel: **a** 0%; **b** 5%; **c** 10%; **d** 20%. A warmer color indicates more severe power shortages, while a cooler color indicates less severe shortages. The lines denote the combination of the transmission capacity and energy storage duration for a certain power shortage level.

transmission costs, and energy storage costs of 47.24 USD/MWh, $64.7, and $71.0 billion, respectively (Supplementary Table 3). Interestingly, to achieve a possible lower national total power shortage rate (even if above the critical standard 0.1%) in a zero-fossil fuel power generation system, short-term energy storage and power transmission must at least reach certain capacities at the same time. For instance, to achieve a national total power shortage rate lower than 2%, the transmission capacity must be more than doubled, while the storage duration must exceed 5 h. Moreover, to realize a national total power shortage rate lower than 0.5%, these two thresholds are 2.5 times and 7 h, respectively. Overall, a fully nonfossil power system could hardly achieve satisfactory power reliability in 2050 unless short-term energy storage and transmission facilities are widely developed (as indicated by the limited colder color area representing a national total power shortage rate lower than 0.1% in Fig. 2a) or long-term storage is included (as below).

## Abated fossil fuel power generation improves power system reliability

By constructing a full-chain CCUS source–sink matching optimization model (see "Methods" section) under the constraints of the $CO_2$ geological storage potential, injection rate capacity and a maximum transport distance of 500 km, as well as suitable size (≥300 MW) and remaining life (≥15 years) criteria (Supplementary Fig. 4a), 718 of 944 coal-fired power plants (907.8 of 949.1 GW) and 58 of 165 (53.6 of 55.8 GW) natural gas-fired power plants were selected as CCUS retrofit candidates, and they were matched with 5471 storage sites (20 × 20 km² per site) across China, including 4926 deep saline aquifer sites and 545 oil field sites (for enhanced oil recovery (EOR)) in 17 onshore basins and 4 offshore basins (Supplementary Fig. 5a). The resulting CCUS supply curve representing the relationship between the source–sink distance and the cumulative installed capacity of CCUS retrofitted power plants shows that some fossil fuel power

plants can access $CO_2$ storage sites within very short transport distances (Supplementary Fig. 4b). For instance, 178.4 GW obtained from 141 plants could match storage sites within 100 km (Supplementary Fig. 4b), including four oil fields that can provide extra benefits resulting from enhanced oil discovery and 137 deep saline aquifer storage sites. These infrastructure and geographical strengths provide the premise for combining a high share of renewable power with abated fossil power generation involving CCUS.

To explore the effect of abated fossil power generation technology involving CCUS on the power system reliability, we simulated 9500 scenarios with 20 fossil fuel power generation shares (1%-20%, at 1% intervals) combined with 0-24 h of short-term energy storage (at 1-hour intervals) and 1–10 times the reference transmission capacity (0.5 intervals), reaching a total of 10,450 scenarios after including the abovementioned 950 zero-fossil fuel power generation scenarios with or without long-term energy storage (Supplementary Fig. 2 and Methods). For simplicity, we only focus on the simulation results for representative abated fossil fuel power generation shares (5%, 10%, and 20%) considering a maximum short-term energy storage of 12 h and five times the reference transmission capacity since higher capacities only slightly affect the results, while the long-term energy storage scenario was used for comparison. Our simulations showed that a high share of renewables combined with abated fossil power generation involving CCUS could effectively improve the electricity supply reliability when maintaining the transmission capacity and short-term energy storage unchanged or reduce the need for transmission or the energy storage to reach the same power reliability. Under the 5%, 10%, and 20% abated fossil fuel power generation scenarios, for instance, the maximum national total power shortage rates at the reference transmission without energy storage were 19.4%, 16.9%, and 12.8%, respectively, which are 2.4%, 4.3% and 9% lower, respectively, than the value of 21.8% under the zero-fossil fuel power generation scenario (Fig. 2b–d, respectively). The minimum national total power shortage scenario, represented by five times the reference transmission capacity combined with 12 h of energy storage, yielded national total power shortage rates of 0.06%, 0.03%, and 0.03% under the 5%, 10%, and 20% abated fossil fuel power generation scenarios, respectively, which are 0.01%, 0.04% and 0.04% lower, respectively, than that under the zero-fossil fuel power generation scenario (Fig. 2b–d, respectively). In addition, to approximate the highest power system reliability under the zero-fossil fuel power generation scenario (i.e., an electricity shortage rate of 0.07% at five times the reference transmission capacity and 12 h of energy storage), only 4.5-fold transmission with 10-hour storage, 3.5-fold transmission with 8-hour storage, and 3.5-fold transmission with 7-hour storage were needed under the 5%, 10%, and 20% abated fossil fuel power generation scenarios, respectively, corresponding to 1.1%, 2.5%, and 2.8% decreases in the LCOE, respectively. These infrastructure construction cost savings for energy storage and transmission (-\$9.5–\$36.7 billion), combined with the potentially avoided stranded assets due to CCUS retrofitted fossil fuel power plants[55,56] (\$4.2–\$16.8 billion according to the value of 520.3\$/kW·for coal-fired power plants and the value of 334.6\$/kW for gas-fired power plants[53]), represent the dual benefits of multisource power systems in China.

## Optimal power system structure

The costs were then calculated for the 10,450 simulated scenarios (Methods). As shown in Fig. 3a, the minimum system cost satisfying the power reliability requirements (a national total power shortage rate of 0.1%) first decreased and then increased as the share of abated fossil fuel power generation with CCUS was increased from 0% to 20% without long-term energy storage. As a result, the 2050 power generation system in China attained the lowest cost of \$662 billion, which is 2.5% lower than that of the zero-fossil fuel power system, and this power system includes 16% abated fossil fuel power generation with

CCUS (14.9% for coal-fired power plants and 1.1% for natural gas-fired power plants), 30.0% variable wind power (the sum of onshore and offshore sources) and 30.5% PV power, and 12.6% hydropower, 11.1% nuclear power and 8.6% energy storage-aided generation (Fig. 3b). To overcome the intermittent and uneven distribution of variable renewable resources, a maximum short-term storage capacity of 8 h and a transmission capacity that is 3 times the reference capacity should be adopted under the lowest-cost scenario (Supplementary Fig. 6b). CCUS retrofitted fossil fuel plants (coal and natural gas) are concentrated in several provinces in central regions, such as Jiangsu, Henan and Hebei, but they are also scattered in the Northeast China, Northwest China and South Coast regions (Fig. 3c), with most plants matched with storage sites between 9.7 and 146.8 km (5th to 95th percentiles), of which 13 fossil power plants in the southern coastal provinces were matched with offshore storage sites (Fig. 3c). In the regional electricity generation composition, CCUS retrofitted fossil fuel power generation accounted for up to 68%, 40% and 31% of the total electricity supply in the Beijing-Tianjin, East Coast, and North Coast regions, respectively, corresponding to 57% of the national fossil fuel power generation involving CCUS (Fig. 3b).

Our simulations revealed a specific optimal electricity supply structure (without considering long-term energy storage) in each province with matched hour-by-hour interprovincial transmission or at any downscaling level. For instance, at the regional level, the Northwest and Southwest China regions, with high variable renewable power generation potential, could supply up to 2196 and 313 TWh of electricity, respectively, to other regions while meeting their own electricity demand, accounting for 81% and 11%, respectively, of the total interregional transmission (Fig. 3b). Regarding seasonal changes, the Northeast and Northwest China regions generate less electricity in summer than in winter (−8.4% and −7.6%, respectively), while other regions, conforming with national features[57,58], generate more electricity in summer than in winter (2.6% to 26.7%, respectively). This occurs because more space heating (by electricity) is needed in the Northeast and Northwest China regions due to the cold winter weather conditions, in addition to the need for higher transmission power to the eastern regions where renewable sources are scarce in winter (Fig. 1a, c, g). Moreover, almost all regions exhibit higher solar PV power generation in summer than in winter, with seasonal differences ranging from 0 to 91% (national: 37%), and higher wind power generation in winter than in summer, ranging from 17 to 41% (national: 27%).

Long-term energy storage technology (e.g., hydrogen and thermal energy storage) may play an essential role in sustaining electricity supply reliability, similar to the role of fossil fuel power generation with CCUS. We simulated a set of scenarios considering a zero-fossil fuel power system with long-term hydrogen storage for comparison. The results showed that a minimum combination of the transmission capacity and short-term energy storage is required to ensure a relatively low power shortage rate under the zero-fossil fuel power system with long-term hydrogen storage, similar to all other scenarios. For instance, meeting the 0.1% national total power shortage rate with the lowest cost necessitates the construction of facilities providing 6 h of short-term energy storage and 4 times the reference transmission capacity (Supplementary Fig. 6a), corresponding to a system cost of \$820 billion and a levelized cost of 47.15 USD/MWh, which are 19.3% and 3.2% higher, respectively, than the high-renewable scenario with 16% abated fossil fuel power generation (Supplementary Fig. 7). However, if the national electricity supply reliability standard were further enhanced (i.e., lower than 0.1% of power shortage), long-term energy storage would play a more important role than abated fossil fuel power generation involving CCUS, as shown by the larger area towards the right-upper corner representing the same power shortage level in Supplementary Fig. 6a than Fig. 6b, or even result in a lower levelized cost to meet the higher power shortage rate standard, e.g.,

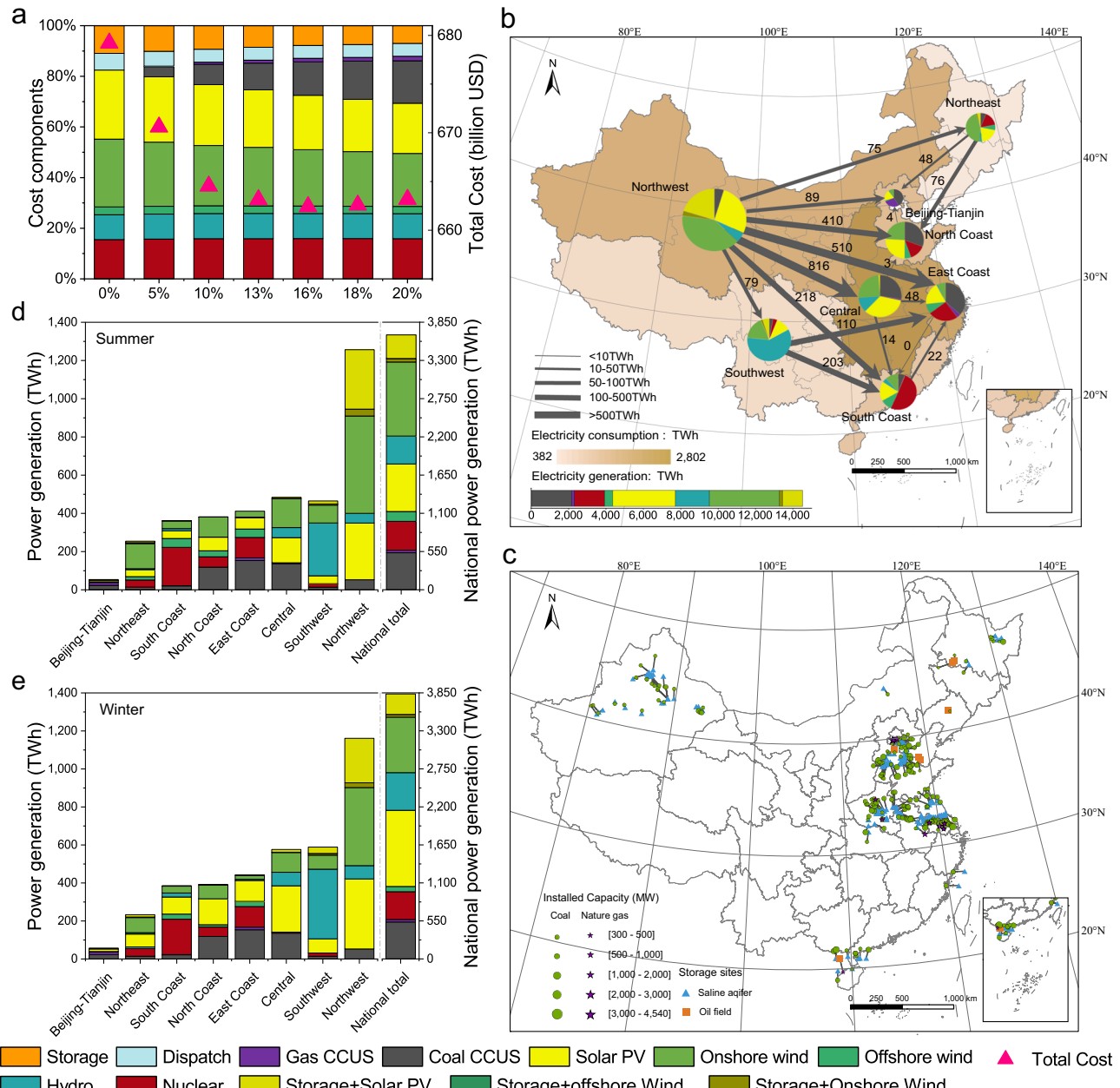

**Fig. 3 | Selection of the lowest-cost power system configuration in 2050 and the regional-level distribution. a** The lowest cost of the power system and the cost composition for different CCUS retrofitted power proportions at an acceptable 0.1% national total power shortage rate, indicating that the optimal power mix should include 16% abated fossil fuel power generation with CCUS retrofitting. **b** Interregional power transmission under the optimal power configuration system. **c** Distribution of fossil power plants and their matched storage sites under the optimal power system after CCUS source-sink matching. Regional and national summer (**d** June, July, and August) and winter (**e** December, January, and February) power generation compositions. Data Credits: All the provincial boundaries are from the Ministry of Civil Affairs of the People's Republic of China (http://xzqh.mca.gov.cn/map, Map Content Approval Number: GS (2022)1873).

with at lowest 47.19 and 48.21 USD/MWh to achieve a 0.01% national total power shortage rate under the zero-fossil fuel power system with long-term energy storage and high-renewable scenario with 16% abated fossil fuel power generation, respectively. Therefore, long-term energy storage technology is a wise option to improve the power system supply reliability in the face of more stringent national shortage rate standards in the future.

## Power system resilience to extreme climatic events
Considering the vulnerability of variable renewable energy to weather variability (e.g., wind speed and irradiance), we measured the power system resilience to historical extreme climatic events by simulating and comparing the impacts of snowstorms, sandstorms, droughts, and

heat waves on power shortages under power systems using zero-fossil fuel power generation and a high share of renewables combined with 16% abated fossil fuel power generation involving CCUS (i.e., the lowest-cost scenario). As shown in Fig. 4, both types of power systems are likely to be affected by these extreme climatic events, but the impact would be much less in the case of a high share of renewables combined with abated fossil power generation involving CCUS.

In the case of snowstorms, the affected areas would exhibit a total power shortage rate over 10% of 396 h during the disaster period under the zero-fossil fuel power system, with a maximum single-hour power shortage rate of 44%, accounting for 75% of the main affected period (January 14-February 4), seriously affecting the power grid stability (Fig. 4a). With a high share of renewables combined with 16%

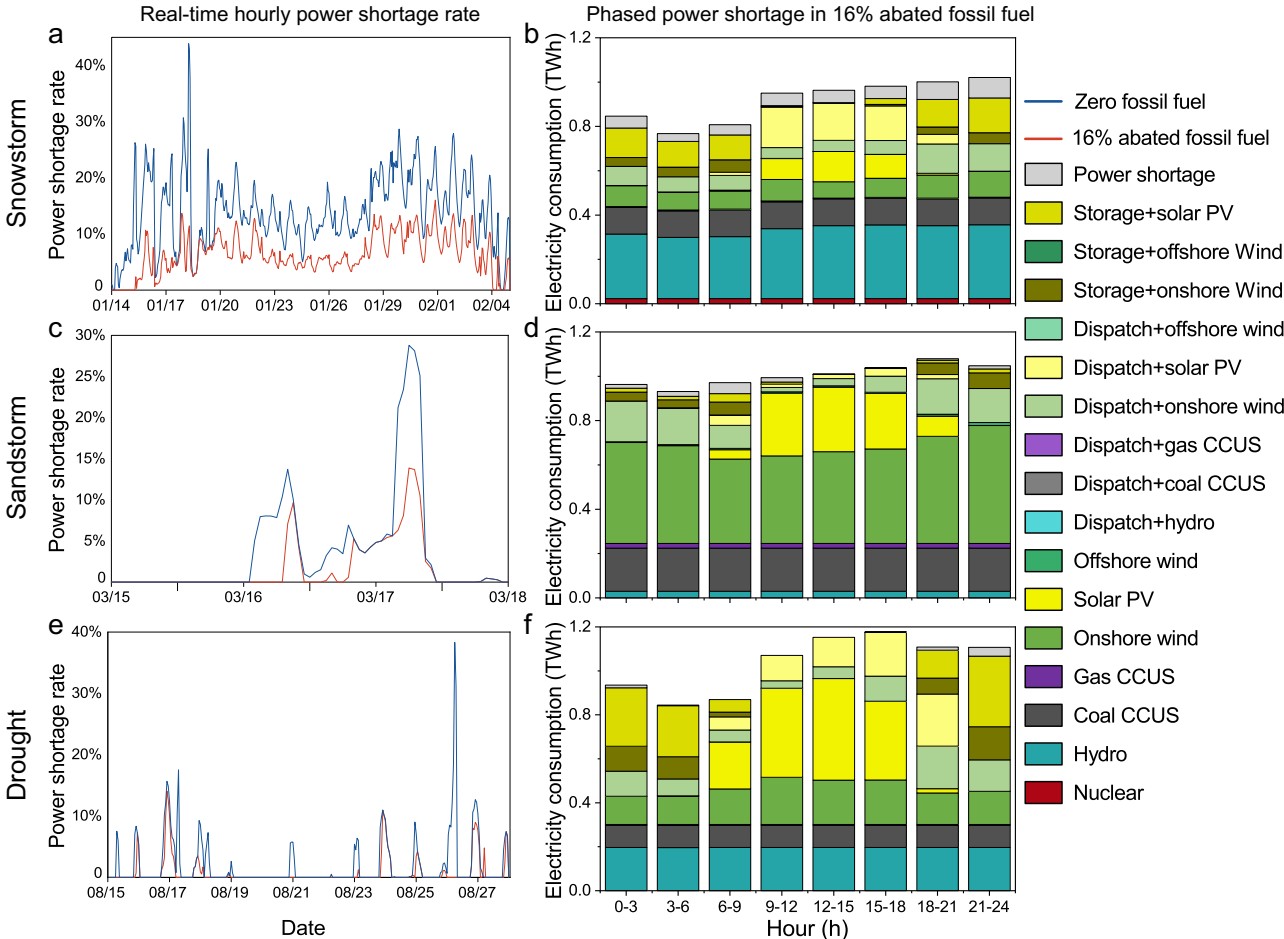

**Fig. 4 | Comparison of the power shortage rates over the event time in the affected provinces under snowstorms, sandstorms, and droughts.** The line chart indicates the real-time power shortage under disasters (**a** snowstorms; **c** sandstorms; **e** droughts), and the bar chart shows the composition of the electricity consumption in a 3-hour cycle under disasters (**b** snowstorms; **d** sandstorms;

**f** droughts). Note that **b**, **d**, **f** only show the electricity consumption under the optimal power system. The event times and intensities and the affected provinces are sourced from actual disasters in China, i.e., 14 January to 4 February 2008, for snowstorms, 15–17 March, 2021, for sandstorms, and 12–27 August 2022, for droughts.

abated fossil fuel power generation, the occurrences of hourly power shortage rates over 10% would decrease to 101, accounting for just 19% of the main affected period with a maximum single-hour power shortage rate of 16% (Fig. 4a). Correspondingly, a high share of renewable power combined with 16% abated fossil fuel power generation could yield a much lower total power shortage of 10.4 TWh in affected provinces (i.e., regional power shortages), decreasing by 22.9 TWh or 54% relative to the zero-fossil fuel power system (Supplementary Fig. 8b). At the provincial level, the abated fossil fuel power plants with CCUS were more effective for Anhui, Hebei, and Guangxi than other affected provinces in alleviating the power shortages under zero-fossil fuel power system, decreasing the provincial power shortage rate from 48%, 9.5%, and 4% to 0.03%, 1.58%, and 1.04%, respectively (Supplementary Fig. 8a).

In comparison, sandstorms and droughts are less likely to impact the power system in terms of both the occurrence of adversely affected hours with a power shortage rate over 10% (8 and 18 h, respectively) and total power shortages (0.9 and 2.1 TWh, respectively) in the affected provinces (i.e., regional power shortages), but a high share of renewables combined with an abated fossil fuel power generation system is more resilient than a zero-fossil fuel power system (Fig. 4c, e). For instance, the areas affected by sandstorm and drought events would experience 5 and 13 more hours with hourly power shortage rates over 10% under the zero-fossil fuel power system than under the high-renewable power system with abated fossil fuel power

generation, and the regional total power shortages could be reduced by 56% and 57% during these events (from 0.9 to 0.4 TWh and 2.1 to 0.9 TWh, respectively) under the abated fossil fuel power generation system, respectively (Fig. 4d, f and Supplementary Fig. 8d, f). Notably, electricity shortages due to sandstorms will not occur until 16 March under the zero-fossil fuel power system although these weather events started earlier on 15 March, and in the case involving a high share of renewables combined with abated fossil fuel power generation, the start of severe electricity shortages could be further delayed by ~7 h (Fig. 4c). This is mainly due to the strong winds associated with these sandstorms initially increasing wind power for energy storage, which could stabilize the electricity supply for several additional hours, especially when fossil fuel power was included in the system. At the provincial level, Xinjiang, Gansu, and Ningxia under sandstorms and Sichuan under drought events exhibited more effective than other affected provinces in alleviating the power shortages under zero-fossil fuel power system, with the cumulative power shortage rates decreasing from 6.61%, 6.47%, 41.93%, and 5.4% to 0%, 4.57%, 19.0%, and 3.0%, respectively (Supplementary Fig. 8c, e). Heat waves exerted the least impact on the electricity supply system under either the zero-fossil fuel power generation system or the abated fossil fuel power generation with CCUS system (Supplementary Figs. 8g, h and 9).

Importantly, adopting abated fossil fuel power generation could alleviate extreme power shortages at certain moments during weather

events. For instance, the highest hourly power shortage rates in the zero-fossil fuel power system during snowstorms (43.9%), sandstorms (28.8%), and droughts (38.3%) may be reduced by 33.2%, 14.9%, and 38.3%, respectively, under a 16% abated fossil fuel power generation system (Fig. 4a, c, e). Nevertheless, a high proportion of renewables combined with CCUS retrofitted fossil fuel power generation remained adversely affected almost throughout the entire 24 h a day during snowstorms and sandstorms. As shown in Fig. 4b, d, power shortages were observed throughout the day and night during snowstorms, with the power shortage rate peaking at 21–24 pm, accounting for up to 9.1% of total electricity demand during this period (Fig. 4b), while the most severely affected periods during sandstorms mainly ranged from 0–12 am, with the 6–9 am period exhibiting the highest power shortage rate of up to 4.9% of total electricity demand during this period (Fig. 4d). However, power shortages during sandstorms from 12 am to 6 pm were negligible, mainly because the extra solar power stored in the morning could be discharged in the afternoon to meet the electricity demand even though these sandstorms severely affected the PV output. In contrast to snowstorms and sandstorms, a high proportion of renewables with abated fossil fuel power generation under droughts could generate slight power shortages, and the power shortage rate over 1% only occur from midnight to 3 am and 18–24 pm, with a maximum value up to 3.6% from 21–24 pm (Fig. 4f). These results were confirmed by a similar performance in the zero-fossil fuel case (Supplementary Fig. 8f).

## Discussion

In this study, we constructed a high-resolution comprehensive simulation model for hourly power system optimization and applied it to evaluate deep decarbonization options for China's power system in 2050. We compared the impacts of various systems with a high share of renewables combined with abated fossil fuel power generation involving CCUS to that of the zero-fossil fuel power system on electricity supply reliability and resilience. In order to minimize electricity supply shortages at a very high temporal resolution, the model configuration considers future power system compositions based on the estimated hour-by-hour power output potential of different power source types, as well as the predicted hourly electricity demand in 31 Chinese provinces in 2050. Then, 10,450 scenarios based on various combinations of short-term energy storage duration, transmission capacity, and share of abated fossil fuel power generation or use of long-term energy storage were simulated to configure the electricity supply structure considering electricity shortages and corresponding system costs. After determining the lowest-cost high-renewable power structure with abated fossil fuel power generation, we further simulated the impacts of extreme climate events on power shortages under the zero-fossil fuel power system and a 16% abated fossil fuel power generation system. Through these simulations, the optimal power structure considering both reliability and resilience could be derived, which is useful for the Chinese power sector to develop long-term decarbonization pathways toward the 2060 carbon neutrality goal.

As the carbon dioxide capture rate does not reach 100%, typically up to 90% for fossil fuel power generation-related flue gas[59], the optimal high-renewable share power system configured in this study cannot achieve net-zero emissions via sole reliance on fossil energy power generation with CCUS. A simple and feasible way to achieve a net-zero power system is to co-fire fossil fuels and biomass energy sources at CCUS retrofit-ready plants to offset the uncaptured $CO_2$ benefits from the negative emissions of bioenergy with CCUS (BECCS). By simulating the optimal matching of surrounding biomass resources (agriculture residues, forest residues, and energy plants) with candidate CCUS fossil fuel-fired power plants, we found that 166 out of 196 candidate CCUS fossil fuel-fired power plants could be matched with biomass resources within a 50-km radius (Supplementary Fig. 10). The optimal power system could achieve net-zero emissions at an average

co-firing ratio of 13% (see "Methods" and Supplementary Fig. 10). Based on available engineering experiences and research evidence[60,61], a co-firing ratio of less than 20% could result in very low costs, as no retrofitting of coal-based boilers is needed. Thus, China could achieve a net-zero power system by 2050 as a result of the partial mitigation contribution of BECCS associated with biomass and coal co-firing. Furthermore, if existing fossil fuel-fired power plants could be fully converted into dedicated biomass-fired power plants with sufficient biomass resources and geological sequestration, this system could achieve considerable net negative emissions. Following the lead of selected developed countries (e.g., the 45Q credit system in the US and carbon taxes in Norway)[32], China's incentive policies for CCUS, as well as biomass and coal co-firing with CCUS, should be enhanced to promote deployment at fossil fuel-fired power generation facilities.

Finally, differentiated measures should be implemented in the various Chinese regions (provinces) according to their dependence on cross-regional (interprovincial) power transmission. At the regional level (Supplementary Fig. 11a, b), the Beijing-Tianjin and East Coast regions are net electricity importers with the highest external electricity dependence (i.e., the imported electricity as a percentage of domestic electricity consumption) at 40% and 32%, respectively, while the Northwest, Southwest and Northeast China regions are net electricity exporters, with exported electricity exceeding domestic consumption levels of 109%, 16% and 8%, respectively. At the provincial level (Supplementary Fig. 11c, d), Chongqing and Shanghai have the highest external dependence, at 71% and 60%, respectively, while Xinjiang and Inner Mongolia are net electricity exporters, with exported electricity exceeding domestic consumption levels of 176% and 144%, respectively. Overall, China must construct more long-distance transmission infrastructures in the different regions or provinces with high external electricity dependence while prioritizing different energy storage technology options, including short- and long-term energy storage systems, in regions or provinces that are net electricity exporters. Moreover, the electricity supply mix in these net electricity exporter regions is dominated by variable renewable energy sources, making them more vulnerable to extreme climate events. Therefore, local governments in these areas should enhance the adaptability of their power systems by implementing proactive measures such as grid reinforcement, long-term energy storage, or backup fossil energy power generation to provide an emergency electricity supply.

## Methods
### Research framework

In this paper, we constructed an integrated model comprising six modules that correspond to the six steps of the research framework (Supplementary Fig. 1). In the first step, the real-time hourly potential of the nonfossil power output in 31 Chinese provinces was estimated. A downscaling approach combining historical hour-by-hour climate information and different engineering calculations was used to downscale the aggregate annual wind and solar PV power output potentials in each province to the real-time hourly level. The installed capacity potential of nuclear power and hydropower in 2050 was month-adjusted from existing estimates. In the second step, the hourly electricity demand in each province in 2050 was predicted. In this step, an econometric model was first developed and then downscaled to the hourly level, i.e., the hourly electricity demand in each province in 2050 was projected by combining datasets of the typical workday and non-workday electricity loads for each province with the hourly electricity load in a typical reference province. In the third step, a CCUS source–sink matching model was developed that could be used to identify the optimal links between fossil plants and suitable storage sites under a given fossil fuel share of electricity consumption. In the fourth step, an integrated optimal near-zero power system simulation model was established to assess the reliability associated with the real-

time hourly electricity demand. In general, the real-time hourly electricity demand could be satisfied through four electricity supply sources, i.e., the local real-time hourly electricity supply, real-time hourly electricity dispatch, energy storage discharging, and power transmission from other provinces via energy storage discharging. In this step, we simulated 10,450 cases combining different transmission capacities, power storage durations (including short-term energy storage ranging from 0 to 24 h and long-term energy storage without energy storage duration limitations), and specified shares of fossil fuel power generation to assess their reliability under the framework of a near-zero or net-zero power system, as well as the relationships with the abated fossil power generation share. In the fifth step, a cost-competitive analysis of the different scenarios was performed to finally determine the lowest-cost power system composition in China in 2050. In the sixth step, the effect of abated fossil power generation with CCUS on the resilience of power system was examined by characterizing and comparing the impacts of typical climate events on CCUS retrofitted fossil fuel power generation systems and the zero-fossil power generation system.

## Assessment of the nonfossil fuel power potential

The hourly power output potential of onshore wind, offshore wind, solar PV, and stable nonfossil energy sources was projected separately. A downscaling approach based on real-time hourly climate information for recent decades was used to refine the provincial variable renewable output potential to the hourly scale. The specific prediction methods for each electricity supply source are described in Supplementary Note 1.

## Assessment of abated fossil fuel power generation with CCUS

Referring to Fan et al.[62], we developed a CCUS source–sink matching model based on a multiobjective optimization model with extended $CO_2$ emission sources from existing coal-fired power generation plants to existing coal- and gas-fired power plants. We also updated the storage site database in this study by expanding onshore storage sites to onshore and offshore storage sites. After source–sink matching, we obtained the distribution of power plants that could be prioritized for CCUS project retrofits and used this distribution to determine the maximum hourly generation potential in each province for different fossil fuel shares, with the data and assumptions referenced in Supplementary Note 2.

## Hourly electricity demand predictions by province in 2050

In this study, the electricity demand in 2050 was projected based on econometric models, accounting for different future socioeconomic development scenarios. Considering the future demand-side response (lowering the actual electricity demand) and capacity margin requirements (increasing the actual grid generation demand), a lower total demand scenario was chosen, suggesting that the demand response effect is greater than the capacity margin effect. Under this scenario, the total future electricity demand in China in 2050 would reach 14.53 trillion kWh, as expressed in Eq. (1).

$$NEIED = \sum_n E_n \cdot PS_n \cdot NEP \qquad (1)$$

where $NEIED$ is the national entire ideal electricity demand in 2050 (PWh), $E_n$ is the predicted electricity consumption per-capita in 2050 in province $n$ (PWh), $PS_n$ is the population share of province $n$ (%), and $NEP$ is the national total population in 2050.

In regard to annual provincial electricity demand prediction, a fixed-effects multiple regression model using the per-capita electricity consumption as the dependent variable was developed based on electricity consumption data for 30 Chinese provinces from 1995 to 2019. Then, the per-capita electricity demand in 2050 in each province

was predicted according to the model estimations and the future projections of the independent variables. Combined with the future predicted population of China, the final projected electricity demand in 2050 in each province was further calculated. The econometric model is expressed in Eq. (2).

$$\ln E_{n,y} = \beta_1 \ln GDP_{n,y} + \beta_2 HDD_{n,y} + \beta_3 CDD_{n,y} + \beta_4 \ln SI_{n,y} \\ + \beta_5 \ln EPI_{n,y} + \ln \varepsilon \qquad (2)$$

where $E_{n,y}$ is the electricity consumption per-capita in year $y$ in province $n$ (PWh), $GDP_{n,y}$ is the gross domestic product (GDP) per-capita in year $y$ in province $n$ at 1990 constant prices (USD), $HDD_{n,y}$ is the value of the heating degree days in year $y$ in province $n$ (°C·d) (as expressed in Eq. (3)), $CDD_{n,y}$ is the value of the cooling degree days in year $y$ in province $n$ (°C·d) (as expressed in Eq. (4)), $SI_{n,y}$ is the ratio of the value added of the secondary industry to the GDP in year $y$ in province $n$ (%), $EPI_{n,y}$ is the electricity price index in year $y$ in province $n$ based on the consumption price index of the electricity and heat producing industry, $\beta_1 - \beta_5$ are the regression coefficients of each independent variable, and $\varepsilon$ is the random error term.

$$HDD_{n,y} = \sum_d \left( T^t - T_{d,n,y} \right), T^t > T_{d,n,y} \qquad (3)$$

$$CDD_{n,y} = \sum_d \left( T_{d,n,y} - T^t \right), T^t \le T_{d,n,y} \qquad (4)$$

where $T^t$ is the temperature threshold (°C), and $T_{d,n,y}$ is the average daily temperature on day $d$ in year $y$ in province $n$ (°C).

Compared to the literature on electricity demand predictions with Chinese power system models[40,54,63], we used a more accurate and refined method to predict the hourly electricity demand in each province in China in 2050.

First, according to the monthly electricity consumption and the hourly electricity load on typical workdays and nonworkdays in each province in 2019 (the representative provinces in the eight regions are shown in Supplementary Fig. 12), the corresponding hourly electricity load on workdays and nonworkdays in each month were calculated by Eqs. (5)–(6). Second, considering the number of workdays and nonworkdays in each month, the average hourly baseline electricity load at the same hour in each month was obtained by Eq. (7). Third, due to the data availability, Anhui Province was selected as the reference province, and its hourly actual electricity consumption in 2019 was used to calculate the proportion of the electricity load in each hour of each day in each month relative to the electricity load in the same hour of the same month, as expressed in Eq. (8). Then, the hourly electricity demand in each province for the whole year was calculated referring to the hourly electricity demand variation proportion in Anhui Province determined by Eq. (9). Finally, the real-time hourly electricity demand in each province in 2050 was predicted using the multiplicator of the annual provincial electricity demand relative to 2019. The specific equations are described below.

The hourly electricity load on workdays and nonworkdays in each month in each province is expressed in Eqs. (5)–(6).

$$EC_{n,m,t}^w = TEL_{n,t}^w \cdot \frac{EC_{n,m}}{\sum_t (TEL_{n,t}^w \cdot D_m^w + TEL_{n,t}^{nw} \cdot D_m^{nw})} \qquad (5)$$

$$EC_{n,m,t}^{nw} = TEL_{n,t}^{nw} \cdot \frac{EC_{n,m}}{\sum_t (TEL_{n,t}^w \cdot D_m^w + TEL_{n,t}^{nw} \cdot D_m^{nw})} \qquad (6)$$

where $EC_{n,m,t}^w$ is the electricity consumption at hour $t$ on workdays of month $m$ in province $n$ (MWh), $TEL_{n,t}^w$ is the typical electricity load at

hour $t$ on workdays in province $n$ (MW), $EC_{n,m}$ is the electricity consumption in month $m$ in province $n$ in 2019 (MWh), $D_m^w$ is the number of workdays in month $m$, $D_m^{nw}$ is the number of nonworkdays in month $m$, $EC_{n,m,t}^{nw}$ is the electricity load at hour $t$ on nonworkdays in month $m$ in province $n$ (MWh), and $TEL_{n,t}^{nw}$ is the typical electricity load at hour $t$ on nonworkdays in province $n$ (MW).

The average hourly baseline electricity load in each province can be obtained with Eq. (7).

$$AEC_{n,m,t} = \frac{EC_{n,m,t}^w \cdot D_m^w + EC_{n,m,t}^{nw} \cdot D_m^{nw}}{D_m^{all}} \quad (7)$$

where $AEC_{n,m,t}$ is the average electricity consumption at hour $t$ in month $m$ in province $n$ (MWh).

The variation ratio of the actual hourly electricity consumption to the average electricity consumption in Anhui Province is expressed in Eq. (8).

$$PP_{m,d,t} = EC_{m,d,t} / \left( \frac{\sum_d EC_{m,d,t}}{D_m^{all}} \right) \quad (8)$$

where $PP_{m,d,t}$ is the variation proportion of the actual electricity consumption in Anhui Province in 2019 to the average electricity demand at hour $t$ on day $d$ in month $m$ and $EC_{m,d,t}$ is the actual electricity consumption in 2019 in Anhui Province at hour $t$ on day $d$ in month $m$ (MWh).

The hourly electricity demand in the other provinces is defined in Eq. (9).

$$ED_{n,m,d,t} = AEC_{n,m,t} \cdot PP_{m,d,t} \quad (9)$$

where $ED_{n,m,d,t}$ is the electricity demand in province $n$ at hour $t$ on day $d$ in month $m$ (MWh).

Finally, the real-time hourly electricity demand in 2050 in each province can be estimated using the multiplicator derived from the provincial electricity demand in 2050 relative to 2019 estimated by econometric models.

### Optimal near-zero power system simulation model

In this paper, an optimal near-zero power system simulation model was established, which incorporates the 2050 hourly electricity demand, nonfossil fuel power output potential predictions, and the optimal layout of CCUS source–sink matching for fossil fuel power plants, as well as power transmission and energy storage (including short- and long-term energy storage). This model was calculated using MATLAB software. Since the future electricity supply structure in China will fundamentally differ from the current one, 35 new interprovincial transmission routes was added to the existing 50 ones, i.e., a total of 85 transmission routes, comprising the reference transmission capacity under the scenario framework (Supplementary Table 1).

Considering the uncertainty in the future near-zero power system structure, we considered a total of 10,450 scenarios (19 × 25 × 22) in this study based on different transmission capacity times (1–10 times at 0.5 intervals), short-term energy storage durations (0–24 h at 1-hour intervals[9]), and abated fossil fuel shares (0%–20% as an integer) or zero-fossil fuel with long-term energy storage for comparison (the inclusion mechanism for long-term energy storage is described in the Supplementary Note 3). On this basis, the hourly energy mix and power shortage in each province were simulated under different scenarios.

### Power shortage rate definition

In this study, we defined four categories of power shortage rates, including the national total power shortage rate, provincial total power

shortage rate, national hourly power shortage rate, and provincial hourly power shortage rate, as described below.

First, the national total power shortage rate represents the cumulative gap between the provincial hourly electricity supply not meeting the ideal hourly electricity demand divided by the national overall ideal electricity demand, as expressed in Eq. (10).

$$TPS^N = \frac{\sum_n \sum_t (IED_{n,t} - ES_{n,t})}{\sum_n \sum_t IED_{n,t}} \quad (10)$$

where $TPS^N$ is the national total power shortage rate, $ES_{n,t}$ is the electricity supply at hour $t$ in province $n$, including the local real-time hourly electricity supply via power generation, real-time hourly dispatch electricity supply via power generation, local energy storage discharging electricity supply, and hourly dispatch via energy storage discharging electricity supply, as expressed in Eq. (15), and $IED_{n,t}$ is the ideal electricity demand in 2050 at hour $t$ in province $n$, as defined in Eq. (9).

Second, the provincial total power shortage rate represents the cumulative gap in a specific province between the hourly electricity supply not meeting the ideal hourly electricity demand divided by the provincial overall ideal electricity demand, as expressed in Eq. (11).

$$TPS_n^P = \frac{\sum_t (IED_{n,t} - ES_{n,t})}{\sum_t IED_{n,t}} \quad (11)$$

where $TPS_n^P$ is the total power shortage rate in province $n$.

Third, the national hourly power shortage rate represents the gap between the national hourly electricity supply not meeting the ideal hourly electricity demand divided by the national ideal hourly electricity demand, as defined in Eq. (12).

$$HPS_t^N = \frac{\sum_n (IED_{n,t} - ES_{n,t})}{\sum_n IED_{n,t}} \quad (12)$$

where $HPS_t^N$ is the national hourly power shortage rate at hour $t$.

Finally, the provincial hourly power shortage rate represents the gap in a specific province between the hourly electricity supply not meeting the ideal hourly electricity demand divided by the provincial ideal hourly electricity demand, as defined in Eq. (13).

$$HPS_{n,t}^P = \frac{IED_{n,t} - ES_{n,t}}{IED_{n,t}} \quad (13)$$

where $HPS_{n,t}^P$ is the hourly power shortage rate at hour $t$ in province $n$.

### Assumptions and configuration of the power system simulation model

In this study, an optimal simulation model for the future near-zero power system was constructed involving seven power generation technologies (nuclear power, hydropower, onshore wind, offshore wind, solar PV technology, coal-fired power with CCUS, and natural gas-fired power with CCUS), and four electricity supply sources of the local real-time hourly power output, real-time hourly dispatch, local energy storage discharging, and power transmission from other provinces via energy storage discharge (Supplementary Fig. 2). To ensure a more realistic power system composition and to simplify the modeling process, the following assumptions were made:

1. Due to the difficulty of obtaining grid transmission lines within a province and simplifying the model to ensure a manageable optimization issue, the power system simulations only considered interprovincial power transmission, which enabled each province to be regarded as a single node. Moreover, the same type of power generation unit within a given province was considered a single unit.

2. Considering the grid preference for various electricity supply sources and the cost of power generation technology, the power system was assumed to prioritize the use of steady-state power sources, and thus, the priority for the adoption of the various power generation technologies was nuclear power > hydropower > abated fossil fuel power > variable renewable energy power (including onshore wind power, offshore wind power, and solar power).

3. Energy storage was classified as short-term (within 24 h) and long-term (without time constraints) energy storage. Due to the higher cost of long-term energy storage, priority was given to short-term energy storage discharge while supplying electricity.

4. Considering the different costs of the various electricity supplies, the priority for the use of the electricity supply in each province was as follows: real-time hourly electricity supply from local power generation > real-time hourly dispatch electricity supply via power generation > local short-term energy storage discharging > hourly dispatch via short-term energy storage discharging > local long-term energy storage discharging > hourly dispatch via long-term energy storage discharging.

5. To reduce power shortages, real-time hourly dispatch was prioritized for supplying the province with the most severe power shortages where the local real-time hourly electricity supply cannot meet the provincial electricity demand and transmission lines are available.

6. To improve the number of utilization hours and increase power generation in provinces with better resource conditions, provinces with the highest remaining variable renewable energy power generation potential after meeting their local real-time hourly electricity supply and real-time hourly dispatch were prioritized for energy storage discharging and subsequent dispatch.

7. In terms of short-term energy storage charging, only variable renewable energy storage was examined, and different storage durations were assumed as the upper limit for continuous energy storage charging in each province, without considering the time constraint of energy storage discharging.

The objective of the optimal near-zero power system simulation model was to ensure a minimum national shortage rate under the scenarios with different abated fossil fuel shares, transmission capacities, and storage durations. The objective is defined in Eq. (14).

$$\min NPS = \sum_n \sum_t \left(IED_{n,t} - ES_{n,t}\right) \tag{14}$$

where $NPS$ is the national power shortage (MWh), and $ES_{n,t}$ includes four electricity supply sources, as expressed in Eq. (15).

$$ES_{n,t} = ES_{n,t}^l + ES_{n,t}^d + ES_{n,t}^s + ES_{n,t}^{sd} \tag{15}$$

where $ES_{n,t}^l$ is the local real-time hourly electricity supply via local power generation at hour $t$ in province $n$, $ES_{n,t}^d$ is the real-time hourly electricity supply from other provinces via power generation to province $n$ at hour $t$ through hourly dispatch, $ES_{n,t}^s$ is the hourly electricity supply from energy storage discharging at hour $t$ in province $n$, and $ES_{n,t}^{sd}$ is the hourly electricity supply from other provinces to province $n$ at hour $t$ through hourly dispatch via energy storage discharging.

The overall constraint of an optimal near-zero power system is to ensure that the hourly electricity supply is lower than the electricity demand, as determined in Eq. (16).

$$ES_{n,t} \leq IED_{n,t} \tag{16}$$

where $ES_{n,t}$ is the total electricity supply at hour $t$ in province $n$, including four electricity supply sources. Based on the hourly electricity supply sources, the optimal near-zero power simulation model was divided into four modules: local real-time hourly electricity supply via power generation module, real-time hourly dispatch electricity supply via power generation module, local energy storage discharging electricity supply module, and hourly dispatch via energy storage discharging electricity supply module. Note that energy storage in the above modules does not include long-term energy storage. The main constraints of the four modules are as follows:

First, in the local real-time hourly electricity supply via power generation module, local power generation was prioritized to meet the local electricity demand, with the main constraints including the following: the local real-time hourly electricity supply via power generation in each province must not exceed its electricity demand, and it must not exceed the total power generation potential of local power generation technologies, as expressed in Eqs. (17) and (18), respectively.

$$ES_{n,t}^l \leq IED_{n,t} \tag{17}$$

$$ES_{n,t}^l \leq \sum_z PGP_{z,n,t} \tag{18}$$

where $PGP_{z,n,t}$ is the power generation potential of the various power generation technologies $z$ at hour $t$ in province $n$, where $z = 1$ is onshore wind power, $z = 2$ is offshore wind power, $z = 3$ is solar PV power, $z = 4$ is nuclear power, $z = 5$ is hydropower, $z = 6$ is coal-fired power with CCUS, and $z = 7$ is natural gas-fired power with CCUS.

Second, if the local real-time hourly electricity supply from power generation is insufficient to meet the local real-time hourly electricity demand, the real-time hourly dispatch electricity supply via the power generation module will be needed, with the main constraints including the following: the dispatched electricity should not exceed the local unmet electricity demand by local power generation; the maximum dispatched electricity along each route is constrained by its designed transmission capacity, and it should not exceed the upper limit of that available from the outflow province, as expressed in Eqs. (19)–(21).

$$ES_{n,t}^d \leq IED_{n,t} - ES_{n,t}^l \tag{19}$$

$$ES_{n,t}^d \leq \sum_{n'} X_{n',n} \cdot DC_{n',n} \tag{20}$$

$$ES_{n,t}^d \leq \sum_{n'} X_{n',n} \cdot \left(\sum_z PGP_{z,n',t} - IED_{n',t} - EDS_{n',n,t}^d\right) \tag{21}$$

where $X_{n',n}$ is a binary variable and is assigned a value of 1 if there is a transmission line from dispatched outflow province $n'$ to dispatched inflow province $n$. Otherwise, a value of 0 is assigned. $DC_{n',n}$ is the maximum transmission capacity from dispatched outflow province $n'$ to dispatched inflow province $n$, $PGP_{z,n',t}$ is the power generation potential of the various power generation technologies $z$ at hour $t$ in province $n'$, $IED_{n',t}$ is the ideal electricity demand in 2050 at hour $t$ in province $n'$, and $EDS_{n',n,t}^d$ is the accumulation of electricity dispatch via power generation from province $n'$ to the other provinces prioritized over province $n$ at hour $t$ (if province $n$ is the most electricity-deficient province and all other provinces are prioritized to supply electricity to province $n$, $EDS_{n',n,t}^d$ is 0).

Third, if the local real-time hourly electricity supply from power generation and real-time hourly dispatch electricity supply from power generation in other provinces still cannot meet the local electricity demand, the local energy storage discharging electricity supply

module will be needed, with the main constraints including the following: the local energy storage discharging electricity supply should be lower than the remaining electricity demand after local hourly power generation and dispatch electricity supply and should be less than the amount of energy storage charging minus the amount of electricity discharged, as expressed in Eqs. (22) and (23), respectively.

$$ES_{n,t}^{s} \leq IED_{n,t} - ES_{n,t}^{l} - ES_{n,t}^{d} \qquad (22)$$

$$ES_{n,t}^{s} \leq \sum_{h=t-h0-H+1}^{t-h0} \left(RWP_{n,h} + RPV_{n,h}\right) - \sum_{h=t-h0+1}^{t-1} \left(ES_{n,h}^{s} + EDS_{n,h}^{sd}\right)$$

$$(23)$$

where $h$ is an auxiliary variable related to $t$ for simulating the process of energy storage charging and discharging, $h0$ is the number of hours from the end of energy storage charging to hour $t$, $H$ is the maximum number of energy storage hours, which ranges from 1 to 24 hours according to the different scenarios, $RWP_{n,h}$ is the remaining power generation potential of wind power after the local real-time hourly electricity supply and real-time hourly dispatch electricity supply at hour $h$ in province $n$, $RPV_{n,h}$ is the remaining power generation potential of solar PV power after the local real-time hourly electricity supply and real-time hourly dispatch electricity supply at hour $h$ in province $n$, $ES_{n,h}^{s}$ is the local energy storage discharging electricity supply at hour $h$ in province $n$, and $EDS_{n,h}^{sd}$ is the electricity dispatch via energy storage discharging from province $n$ at hour $h$.

Fourth, if all the above electricity supply sources cannot meet the local electricity demand, dispatch via energy storage discharging electricity supply module will be needed, with the main constraints including the following: the dispatch via energy storage discharging electricity supply should not exceed the local remaining electricity demand, should not exceed the remaining capacity of each transmission line, and should not exceed the amount of energy storage charging minus the amount of electricity discharged in the electricity outflow provinces, as expressed in Eqs. (24)–(26).

$$ES_{n,t}^{sd} \leq IED_{n,t} - ES_{n,t}^{d} - ES_{n,t}^{s} \qquad (24)$$

$$ES_{n,t}^{sd} \leq \sum_{n'} X_{n',n} \cdot \left(DC_{n',n} - ES_{n',n,t}^{d}\right) \qquad (25)$$

$$ES_{n,t}^{sd} \leq \sum_{n'} X_{n',n} \cdot \left(\sum_{h=t-h0-H+1}^{t-h0} \left(RWP_{n',h} + RPV_{n',h}\right) - \sum_{h=t-h0+1}^{t-1} \left(ES_{n',h}^{s} + EDS_{n',h}^{sd}\right) - \left(ES_{n',t}^{s} + EDS_{n',n,t}^{sd\_\circ}\right)\right) \qquad (26)$$

where $ES_{n',n,t}^{d}$ is the real-time hourly dispatch electricity supply via power generation from province $n'$ to province $n$ at hour $t$, $RWP_{n',h}$ is the remaining power generation potential of wind power after the local real-time hourly electricity supply and real-time hourly dispatch electricity supply at hour $h$ in province $n'$, $RPV_{n',h}$ is the remaining power generation potential of solar PV after the local real-time hourly electricity supply and real-time hourly dispatch electricity supply at hour $h$ in province $n'$, $ES_{n',h}^{s}$ is the local energy storage discharging electricity supply at hour $h$ in province $n'$, $EDS_{n',h}^{sd}$ is the electricity dispatch via energy storage discharging from province $n'$ at hour $h$, $ES_{n',t}^{s}$ is the local energy storage discharging electricity supply at hour $t$ in province $n'$, and $EDS_{n',n,t}^{sd\_\circ}$ is the accumulated electricity dispatch via energy storage discharging dispatched from province $n'$ to province $n$ at hour $t$. Specific calculations are provided in the Supplementary Note 3.

## Cost-competitive analysis of the near-zero power system

To evaluate the economics of the power system, we calculated the costs of the overall power system and its components under all scenarios and selected the optimal scenario characterized by the lowest total cost and total power shortage lower than 0.1% (i.e., ensuring a general level of the electricity supply reliability of 99.9% in Chinese cities). The total cost of the power system includes the cost of non-fossil fuel power generation, the cost of abated fossil fuel power generation with CCUS, the cost of short-term energy storage, the cost of hydrogen energy, and the cost of power transmission (all costs in this study were adjusted to 2020 constant prices), as expressed in Eq. (27).

$$COST = LCOE^{nc} \cdot EC^{nc} + LCOE^{hp} \cdot EC^{hp} + LCOE^{pv} \cdot EC^{pv} + LCOE^{on-wp}$$
$$\cdot EC^{on-wp} + LCOE^{off-wp} \cdot EC^{off-wp} + LCOE^{cpccs} \cdot EC^{cp}$$
$$+ LCOE^{ngccs} \cdot EC^{ng} + LCOE^{es} \cdot EC^{es} + (H2^{uc} \cdot (1 - PEC^{H2}) + H2^{s})$$
$$\cdot H2^{c} + LCOE^{H2} \cdot EC^{H2} + \sum_{I} COST^{UT} \cdot LC_{I} \cdot Cap_{I}^{T}$$

$$(27)$$

where $COST$ is the total cost of the power system (at 2020 constant prices) (USD), $LCOE^{nc}$, $LCOE^{hp}$, $LCOE^{pv}$, $LCOE^{on-wp}$, $LCOE^{off-wp}$, $LCOE^{cpccs}$, $LCOE^{ngccs}$, and $LCOE^{es}$ are the LCOEs of nuclear power, hydropower, solar PV power, onshore wind power, offshore wind power, coal-fired power with CCUS, natural gas-fired power with CCUS, and short-term energy storage in 2050 (USD/kWh), respectively, $EC^{nc}$, $EC^{hp}$, $EC^{pv}$, $EC^{on-wp}$, $EC^{off-wp}$, $EC^{cp}$, $EC^{ng}$, $EC^{es}$, and $EC^{H2}$ are the total electricity consumption levels of nuclear power, hydropower, solar PV power, onshore wind power, offshore wind power, coal-fired power with CCUS, natural gas-fired power with CCUS, short-term energy storage, and hydrogen in 2050 (kWh), respectively (under the long-term energy storage scenario, the additional consumption of additional variable renewable electricity due to the production of hydrogen is captured by $EC^{pv}$, $EC^{on-wp}$, and $EC^{off-wp}$), $H2^{uc}$ is the unit production cost of hydrogen, $PEC^{H2}$ is the proportion of the electricity costs in hydrogen production (%), $H2^{s}$ is the cost of hydrogen storage in 2050 (USD/kg), $H2^{c}$ is the hydrogen consumption in 2050 (kg), $LCOE^{H2}$ is the LCOE of the electricity generated from hydrogen in addition to the cost of fuel in 2050 (USD/kWh), $I$ is the candidate transmission route in this study, with a total of 85, $COST^{UT}$ is the unit transmission cost (USD/km·GW), $LC_{I}$ is the length of power transmission route $I$ (km), and $Cap_{I}^{T}$ is the maximum hourly utilization capacity of route $I$ (GW). Detailed cost data are provided in Supplementary Table 5. The cost calculations of nonfossil fuel power generation, abated fossil fuel power generation, short-term energy storage, hydrogen energy, and power transmission are presented in Supplementary Note 4. The LCOE of the near-zero-carbon power system was obtained by dividing the total levelized cost of the power system by the electricity consumption in 2050.

## Modeling the impacts of extreme weather events

We then investigated the effects of extreme weather events (snowstorms, sandstorms, droughts, and heat waves) on the overall system resilience under the near-zero power system while incorporating abated fossil fuel power generation with CCUS. The 2008 snowstorm in southern China was chosen as a reference disaster since it was the most severe and widespread rain, snow, and freezing natural disaster in China since 2000. The 2021 sandstorm in northern China was adopted as another reference disaster since it was the most powerful and extensive sandstorm in China in the previous decade. Since 2022, heat waves and droughts have become more extreme climate crises plaguing the Chinese power grid. The 2022 heat wave in Southeast and Northwest China showed the highest intensity since the 21st century, and the 2022 drought seriously impacted southern China. Therefore,

these two disasters were also introduced as representative extreme weather events. The mechanism of the impact of each event on the power system is as follows:

The simulation of the 2008 snowstorm impact incorporated the seven most severely affected provinces (including Anhui, Jiangxi, Hubei, Hunan, Guangxi, Sichuan, and Guizhou, as shown in Supplementary Fig. 13a) associated with their climatic conditions (hourly radiation intensity, temperature, wind speed, snowfall, and snow depth) during the snowstorm. Snowstorms often impose three typical effects on the near-zero power system (refer to Supplementary Note 5 for details).

The simulation of the 2021 sandstorm impact incorporated the eight most severely affected provinces (including Xinjiang, western Inner Mongolia, Gansu, Shanxi, Hebei, Beijing, Tianjin, and Ningxia, as shown in Supplementary Fig. 13a) associated with their climatic conditions (hourly radiation intensity, temperature, and wind speed) during the sandstorm. Sandstorms often generate two typical effects on the near-zero power system (refer to Supplementary Note 5 for details).

The simulation of the 2022 drought impact incorporated the six most severely affected provinces (including Sichuan, Chongqing, Hubei, Hunan, Jiangxi, and Anhui, as shown in Supplementary Fig. 13b) associated with their climatic conditions (hourly radiation intensity, temperature, wind speed, and drought level) during the drought. Droughts often impose three typical effects on the near-zero power system (refer to Supplementary Note 5 for details).

The simulation of the 2022 heat wave impact incorporated the 14 most severely affected provinces (including Hunan, Zhejiang, Chongqing, Jiangxi, Jiangsu, Anhui, Shanghai, Guangdong, Sichuan, Xinjiang, Henan, Hubei, Fujian, and Hainan, as shown in Supplementary Fig. 13b) associated with their climatic conditions (hourly radiation intensity, temperature, and wind speed) during the heat wave. Heat waves often exert three typical effects on the near-zero power system (please refer to Supplementary Note 5 for details).

### Reliability and resilience of the power system

Both the reliability and resilience of the power system can be measured by power shortages. Referring to previous research[9] and national standards for the electricity supply (e.g., 99.9% for cities in China)[16], reliability was defined as the ability of all generating units connected to the grid to meet the electricity demand in normal years (i.e., without extreme weather events), quantified as one minus the power shortage rate. Resilience mainly measures the power system ability to withstand power shortages and restore the electricity supply in a timely manner during extreme weather events[18,19]. Here, the power shortage degrees in affected areas during extreme weather periods were used to indicate the power system resilience, such as power shortage hours, highest power shortage rate, and total power shortage. Considering that zero-fossil fuel power generation and destroyed power transmission infrastructures can be recovered or rebuilt artificially during or after a climatic disaster (e.g., snowstorms), we assumed that the affected power system could gradually restore the normal electricity supply. A detailed demonstration of Lyapunov's observability and controllability for the model is provided in Supplementary Note 6, and the impact of renewable energy costs on the system cost is provided in Supplementary Note 7.

### CO$_2$ emissions accounting boundary

Here, we set the boundary to the direct CO$_2$ emission reduction related to all types of low-carbon technologies involved in the whole power system. Specifically, the indirect CO$_2$ emissions originating from wind power, solar PV, hydropower, and nuclear power were not considered, and only the remaining CO$_2$ emissions that could not be captured by CCUS retrofits for coal-fired and gas-fired power plants were calculated (10% of the total emissions). As a result, we introduced negative emissions through the coal and biomass co-firing system coupled with CCUS to achieve complete net-zero emissions of the Chinese power system under the same emissions accounting framework.

### Optimal matching model used for the coal and biomass co-firing system

Aiming at the coal and biomass co-firing system coupled with CCUS, we developed an optimal matching model to determine the optimal links between CCUS-qualified coal-fired power plants and surrounding biomass feedstocks[64]. This optimal matching model incorporates three biomass resources (agricultural residues, forest residues, and energy crops). The biomass data were spatialized into a $1 \times 1$ km$^2$ grid that could be individually and optimally selected by CCUS-qualified coal-fired power plants. The total amount of agricultural residues and forest residues were assumed to remain constant over time (as in 2015), while the total amount of future energy crops was estimated based on the distribution of suitable marginal land and the associated per-unit yield.

In this model, the matching mechanism was expressed by objective functions that minimized the total biomass transportation distance while maximizing the total biomass feedstocks constrained by the biomass availability (collection radius ≤50 km and maximum co-firing ratio ≤40%). This model introduces two important matching priorities for power plants with higher energy consumption and biomass feedstocks located closer to power plants, as well as the one-way matching rule (i.e., biomass feedstocks from each grid can only be matched with one power plant, while each power plant can receive biomass from multiple sourcing sites). The objective functions are defined in Eqs. (28)−(29).

$$\min FB_1 = \sum_p \sum_b m_{p,b} \cdot DB_{p,b} \tag{28}$$

$$\max FB_2 = \sum_p \sum_b \sum_k BQ_{p,b,k} \cdot T_k \tag{29}$$

where $FB_1$ is the total biomass transportation distance (km), and $m_{p,b}$ is a binary variable describing if biomass grid $b$ can be optimally linked to power plant $p$. If this is the case, $m_{p,b} = 1$, and $m_{p,b} = 0$ otherwise. Moreover, $DB_{p,b}$ is the straight-line distance between biomass grid $b$ and power plant $p$ (km), and $FB_2$ denotes the total biomass feedstocks, both calculated from all surrounding biomass grids to the CCUS-qualified coal-fired power plants (tce). $BQ_{p,b,k}$ is the amount of biomass $k$ ($k = 1, 2, 3$, representing agricultural residues, forest residues, and energy crops, respectively) in biomass grid $b$ linked to power plant $p$ (t), and $T_k$ is the conversion coefficient of biomass $k$ into standard coal, with values of 0.51, 0.57, and 0.52 tce/t, respectively.

The optimal matching model was constrained by the biomass availability associated with the maximum collection radius and the maximum co-firing ratio, as expressed in Eqs. (30)−(31).

$$\sum_b \sum_k BQ_{p,b,k} \cdot T_k \leq Cap_p^c \cdot h_p^c \cdot PGCC \cdot \theta \tag{30}$$

$$DB_{p,b} \leq R^{\max} \tag{31}$$

where $PGCC$ is the standard coal consumption per kWh of electricity generation in province (g/kWh), $\theta$ is the maximum co-firing ratio of 40% by heat, and $R^{\max}$ is the maximum biomass collection radius of 50 km.

The total biomass feedstocks linked to a CCUS-qualified power plant were acquired by aggregating those from all potentially surrounding biomass grids, as defined in Eq. (32).

$$ABQ_p = \sum_b \sum_k m_{p,b} \cdot BQ_{p,b,k} \cdot T_k \tag{32}$$

where $ABQ_p$ denotes the total biomass feedstocks linked to power plant $p$ (tce).

## Data availability
Power supply and demand data generated in this study have been deposited in the Figshare platform [https://doi.org/10.6084/m9.figshare.23614473], and can be obtained from fan@cumtb.edu.cn upon request. Corresponding data sources are listed in Supplementary Note 8.

## Code availability
Codes used in this study can be obtained from fan@cumtb.edu.cn upon request and are available at https://doi.org/10.6084/m9.figshare.23614473.

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

## Acknowledgements

The original reserach work was supported by the National Natural Science Foundation of China (No. 72174196 and No. 71874193 to J.-L.F.), Open Fund of State Key Laboratory of Coal Resources and Safe Mining (No. SKLCRSM21KFA05 to J.-L.F.), and the Fundamental Research Funds for the Central Universities (No. 2022JCCXNY02 to J.-L.F.). We also thank the contributions from Wenlong Su, Wenlong Zhou, Xinmeng Guan, Yujiao Xian, Jiayu Li, and Zixia Ding on the data collection and analysis discussion.

## Author contributions

X.Z. and J.L.F. designed the research. Z.L., J.L.F., and X.H. performed the integrated model simulation and data compiling. J.L.F. wrote the article with major contributions provided by Z.L., X.Z., K.L., X.L., J.W., K.H., and B.S.

## Competing interests

The authors declare no competing interests.
