## [Peer Review File · Nature Communications]

A net-zero emissions strategy for China's power sector using carbon capture utilization and storageREVIEWER COMMENTS

Reviewer #1 (Remarks to the Author):

This paper presents an optimization approach and solution for the power system of China. Methodologically it is rather conventional - especially noting that the optimization is conducted via brute-force iteration rather than some more refined optimization approach. It finds that significant amount of Carbon Capture Utilization and Storage is needed to support the electricity system in deep decarbonization. The treatment of source-sink pairing and inter-province dispatch is interesting. The observations on the challenges posed by the geographic distribution of RE resources are critically important. The study on resilience to extreme weather events is an important contribution. However there are several major points in the analysis that are questionable.

Initially, the authors consider a system with only short-term energy storage, which naturally leads to energy shortage. Coal+CCS is then investigated as a means to avoid shortage and boost reliability. However, it is not clear if coal+CCS is ever investigated in comparison to long-duration energy storage, which is expected to be significantly cheaper by 2050. The authors' conclusion 1, that CCS would reduce energy shortage compared to all FF system is not clearly justified. An all-FF system can supply energy with no shortage if enough storage is included; the question is how much storage is required and what it will cost. The question that should be answered is whether RE(+storage)+CCUS can provide energy with no major shortages at a lower cost than RE+storage. The limiting of storage to 12 hr inevitably leads to shortage in an all-RE system - some amount of multi-day storage will always be necessary in such systems to account for periods of low renewable production. The authors should quantify the cost of the 0-FF system that reaches <0.1% shortage (line 237), and state their cost assumptions clearly. In general cost assumptions are not made clear, especially for energy storage technologies.

Overall, the methodological choices (especially constraints, economic cost assumptions, and lack of inclusion of Hydrogen as a long-term storage option) may predetermine this finding, rendering the paper biased. Unless, this major issue is addressed, after a major revision - the paper's value would be limited.

Below are sequential comments for consideration:

Pg. 1 Ln 31 Abstract - reference to "power shortage" is unclear. Electricity grids must match demand and supply at any time - shortage implies load-shedding (black-outs). The way that this is used here should be explained.

Pg. 3 Ln 66 "or high voltage" - missing transmission lines which should be added.

Pg. 3 Ln 78 - It is true that many IAMs include CCUS but the reasons for this may be structural modeling constraints that are implicit and unrealistic cost assumptions regarding RE integration (cf. Kaya, A., Csala, D. & Sgouridis, S. Constant elasticity of substitution functions for energy modeling in general equilibrium integrated assessment models: a critical review and recommendations. *Climatic Change* 43, 225–14 (2017)).

Pg. 3 Ln 79 - the sentence that CCUS "avoids carbon lock-in" seems a little out of place. CCUS by definition is a result of "carbon lock-in". Need to rethink/rephrase.

Pg. 3 Ln 93-95 the same observation is made as in Ln 78. One should consider systemic biases in modeling. Since the IAM models have underlying structural constraints that prevent them from opting to 100% RE systems, your following statement (that few studies compared both) is correct. There is nevertheless an interesting comparison on Energy Return basis available that provides strong arguments against the use of CCUS for power systems (cf. 1.Sgouridis, S., Carbajales-Dale, M., Csala,

D., Chiesa, M. & Bardi, U. Comparative net energy analysis of renewable electricity and carbon capture and storage. *Nature Energy* 4, 456–465 (2019).)

Pg. 6 Ln 155-158 - some explanation on how the RE potentials have been calculated is needed and future demand by province has been calculated. What do they mean in terms of installed capacities?

Pg. 6 Ln 159 - 164. The concept of "shortage" is introduced. It seems to refer to RE power supply vs demand by province but it is not properly defined. Please define carefully and explain the importance of this metric.

Pg. 8 Fig. 1 - It may be a good idea to create an aggregated figure for the entire China region where RE resources are also aggregated. This would provide a sense of how the system would potentially perform over the whole country.

Pg. 9 Ln 210-213. The storage options are discussed for the first time. Batteries (why not Li-Ion that represent 90% of China's actual currently installed grid battery storage?), compressed air and pumped hydro are claimed to be lower TRL which is not supported. Yes, the battery tech discussed is but there is battery tech that is commercial today with successful project. Also, it seems that hydrogen (a key option for weekly to seasonal storage) is not considered at all which needs to be explained. Given the benefits of adding even 12 hours of storage, this addition would have closed the remaining gap.

Pg. 11 Fig. 2 and Ln 245- 247. Given the lack of Hydrogen inclusion in the analysis as noted in the previous comment, the claim that it is essentially impractical to operate an 100% RE system in China is invalid. It is clear that some kind of longer-term buffer is needed. By excluding this option (hydrogen and operating the pumped hydro only for 12 hour storage), the authors are forcing the conclusion that CCUS (the only other alternative) is needed. Unless, the hydrogen option is included and discussed, the paper's conclusions will not have any practical validity.

Pg. 11 Ln 53 and broadly. At this section, the CCUS option is discussed for the first time From the later discussion, it appears that the utilization component is EOR - which extends oil production. This should be discussed upfront - since EOR is not really a long-term carbon storage mechanism unless it is purposely operated for this (higher CO2 ratios - cf. Santos, R., Sgouridis, S. & Alhajaj, A. Potential of CO2-enhanced oil recovery coupled with carbon capture and storage in mitigating greenhouse gas emissions in the UAE. *Int J Greenh Gas Con* 111, 103485 (2021).) . Either way only a portion of the carbon can be considered captured as it is offset by the additional oil production. This is a significant caveat that should be discussed.

Pg. 20 Ln 476. The methodological assumption for estimating wind potential by a standard 2MW turbine could be accepted. Nevertheless, given that current wind systems have much higher capacities (with off-shore exceeding 10MW), the financial implications of the choice should be discussed specifically (and how it is corrected).

Pg. 21 Ln 495 -526. The solar estimation approach seems to have two main gaps:

- * It does not consider bi-facial systems (current state of the art - adds min 5-10% yield)
- * It does not consider single-axis tracking (current state of the art for utility scale projects that provide significant benefits by adding min 10-15%yield available in morning/afternoon periods)

Also, the cost assumptions of PV seem higher than expected. The table 6 supplementary is not clear but it seems that a price of 5c/kWh are being used for PV, with only a 37% reduction by 2050. This would make 2050 PV costs higher than current costs for high-insolation climates (<3c/kWh) Please revise/explain accordingly.

Pg. 22 Ln 586 - the role of the reference province (Anhui) is unclear. Why is it needed if all provinces have reference hourly values?

Pg. 25 Ln 642 - the dispatch rules of the simulation model seem reasonable with the notable exception of not including dispatchable hydrogen as noted earlier! Still, a more detailed explanation of the model should be provided (key equations and mathematical approach).

Reviewer #2 (Remarks to the Author):

This paper develops an optimal solution to have a reliable and resilient power sector in China with near-zero carbon emissions. Here are my concerns.

1. Reliability and resilience are two important terms in this paper, they should be clearly defined and explained, especially at the beginning of this study.
2. In abstract, the authors argue that the reliability and resilience of different electric power system architecture have rarely been assessed. However, there are lots of studies on the reliability and resilience of electric power systems. The authors should refine and reorganize innovations and contributions.
3. The reliability of the power system also needs to consider the real-time power balance among regions, including the following aspects, such as the analysis on steady state of the power system (when each node in the power system constructed in the article loses 10% of power generation or load, does the system have enough margin to automatically restore the steady state?The discussion about the oscillation period, in brief, the Lyapunov's observability and controllability proof need to be added in the model.
4. In the model and simulation of the impact of extreme weather events, the manuscript only simulates the loss of above 220kV lines within the province. However, the loss of inter-provincial lines was also serious during the 2008 snow disaster, which should also be discussed in the manuscript.
5. The scenario setting is currently inconsistent with the reality of China's power grid, because existing literature shows that since the snowstorm disaster in China in 2008, DC ice melting devices have been installed on transmission lines above 110kV, and it is supposed that ice and snow disasters will not cause large-scale disconnection accidents. Since 2022, heat wave and drought have become a more extreme climate problem that plagues China's power grid: the load on the power grid increases sharply, heat wave causes less wind, and drought causes less water, resulting in an imbalance between power supply and demand. In this paper, you'd better focus on the impact of heat wave and drought on grid resilience
6. In Line 735, KV should be revised as kV

Reviewer #3 (Remarks to the Author):

Thank you very much for inviting me to review the manuscript "An optimal solution to have a reliable and resilient power sector in China with near-zero carbon emissions". Strategies to have a reliable and resilient low-carbon power system are nowadays a hot topic whose impact on ensuring power system security and addressing climate change is remarkable. The manuscript focused on this real and interesting theme, which is sure of significance for power sector planning and climate goal achievement. In particular, the authors proposed a uniquely innovative modelling approach that integrated six interlinked modules (i.e., electricity demand prediction, variable renewable power potential assessment, source-sink optimal matching, power system simulation, cost-competitive analysis and weather extremes impact simulation) and developed a new assessment approach that can quantify the reliability and resilience of different power system by coupling the high share of renewable power system with abated fossil fuel with CCUS technology, which contributes to a notable advance in integrated assessment modelling knowledge. By simulating numerous power system infrastructure scenarios, the authors provided a complete and detailed study for revealing the optimal

solution to have a reliable and resilient power sector in China under the context of carbon neutrality.

Moreover, the modelling approach and analysis framework can be applied to all countries. Therefore, I think the study was conducted systematically under a rational work plan and presented meaningful results. I believe it can be accepted for publication in Nature Communications, and in principle, I recommend only four items could be addressed, as follows.

1. The introduction is well-written and easy to understand. The bibliographic research is complete and well-analyzed. The aims and methods are clearly described and supported by sufficient theoretical background. The current structure of the paper does not require changes.

2. Figure 1 is outstanding. However, the legend parts are a bit small for the reader. Especially if the reader has some age, you could move this legend to the left side of Figure 1g or describe them in a separate space.

3. The paper presented and explained all the key findings. The results and discussions were clearly written and easy to follow, and I could understand everything presented. There is one item that could be considered to enhance the contribution of this paper, i.e., one or two sentences about the findings of the optimal layout of coal-fired and natural gas power plants retrofitted with CCUS could be added in policy implications in the Discussion and Conclusions section.

4. The supplementary material is useful because it provides more insight into the results and case study sections. The authors are suggested to address one minor item in the supplementary figure, namely value one could be marked on the left vertical axis of Figure S3b, to better differentiate the provinces with power shortage from that with power surplus.

Responses to Reviewers' Comments

To Reviewer #1:

1. This paper presents an optimization approach and solution for the power system of China. Methodologically it is rather conventional - especially noting that the optimization is conducted via brute-force iteration rather than some more refined optimization approach. It finds that significant amount of Carbon Capture Utilization and Storage is needed to support the electricity system in deep decarbonization.

Authors' Response: Thank you for your suggestion. We agree with your point that from a modeling perspective, the cost-optimized power system model could be some more refined than the iterative-based approach we currently use in this study. However, considering our research object and in particular the model's operating speed with hourly level optimization, we believe that the iterative-based approach is more suitable for this study. The specific reasons are as follows.

To make the model can be solved in sensible computational time, there are some cost-optimized power system models that consider a country as a whole for the sake of simplifying the model without considering regional and inter-provincial power dispatch ¹. For instance, Daggash and Mac Dowell ² present an power systems optimization model for national-scale power supply capacity expansion applied to the UK without considering inter-regional power transmission. But these models differ significantly from reality and may lead to biased results ³. For another part of the cost-optimized power system models with dispatching considered, most of them have adopted methods of clustering to substantially reduce the model run time. For instance, Zhuo, Du ⁴ used the K-medoids algorithm to cluster the year into 12 typical days; Riera, Lima ⁵ used K-means clustering to aggregate 4194 days (~11.5 years) into 7 typical days per year; Heuberger, Rubin ⁶ also used K-means clustering to aggregate 8760 hours into 11 and 21 categories, respectively.

However, the clustering approach will reduce the model's accuracy for the following various reasons. First, variable renewable resource potential is susceptible to weather variability and varies greatly from season to season, month to month, and day to day ^{7, 8, 9}, and even hourly power generation potential varies significantly within the same cluster ⁴, especially in the moment of sudden occurrence of climatic disaster ^{10, 11}. The clustering approach will omit the details of fluctuation pattern in renewable resource potential over time to a large extent. Second, considering that the fluctuation pattern of electricity consumption differs from that of renewable resource potential, e.g., electricity consumption performs differently on working and non-working days ¹² but renewable power output does not, clustering renewable resource potential and electricity consumption together would result in a clustering bias. Third, the fluctuation patterns of different renewable resource potential are distinct, thus clustering wind power and solar PV together would neglect their inner fluctuating natures. Finally, even within the same study, the clustering method and the number of clusters used will significantly impact the results ^{6, 13}. Overall, by using a clustering method, the hourly cost-optimized power system models diminish model accuracy and omit the volatility information of renewable resource potential, resulting in an underestimation of the total cost of the power system (2.5%-44%) ^{12, 14, 15}.

In addition, the cost-optimized power system models usually set a strong constraint to avoid power shortage, typically by imposing a fixed and fairly high penalty cost of load shedding when power demand cannot be met ^{4, 6, 16}. These settings force the power system not to produce power shortages, which is not in accordance with reality in some cases, challenging to accurately simulate the power shortage of the power system. For example, a power supply shortage may occur during certain occasions, particularly while meeting summer peaks. Besides, massive power shortages during extreme climate events are more likely to occur because the power systems, especially those with a high proportion of renewable energy, are always vulnerable to climate disasters. For example, on December 23-25, 2022, the United States was affected by a winter storm that caused massive power outages, affecting more than 6.35 million people ¹⁷. Therefore, considering our study aims to measure the power system reliability and resilience by using the power shortage indicators under normal and disaster conditions, the penalty-based cost-optimized power system model may be not suitable.

For these above reasons, we have adopted the iterative-based model that is more suitable to investigate main topics of this study, i.e., the reliability and resilience of the power system, both of which could be realistically measured by hourly power shortages under the same modeling framework.

In the revised version, we have added the problems with the current research approach and hope to address them in future studies in the *Limitations and further perspectives* section in *Method* as follows.

“.....First, although we developed an iteration-based approach in this study by simulating a large number of cases in 2050 and obtaining the optimal power structure through cost calculation and comparisons instead of using a linear programming model with the objective of minimal cost that always aggregates the annual hours into ten or more clusters ^{12, 45, 97, 98}. However it did not capture the temporal evolution characteristics of the power system configuration and did not consider more extra cases other than the simulated 10,450 cases considering the computational runtime constraints.”

In the revised version, we have explained the reasons for choosing the iterative-based model instead of the cost-optimized power system model in the *Considerations for the optimal near-zero power system simulation model* section in *Supplementary Information* as follows.

“.....However, an hourly cost-optimized power system model that takes into account inter-provincial power dispatch is very time consuming to solve. To make the model can be solved in sensible computational time, some of the cost-optimized power system models consider a country as a whole and do not consider regional and inter-provincial power dispatch to simplify the model, which may lead to bias in the results. Another clustering-based cost-optimized intertemporal power system model that consider dispatch generally suffers from omitting fluctuation details of variable renewable resource potential ⁷, clustering bias of mixing electricity consumption and renewable energy generation potential ⁸, and inability to accurately assess power shortages especially under extreme climatic events ^{9, 10}, e.g., clustering results are difficult to capture the near real-time changes in meteorological factors at a specific hour during a climatic disaster, resulting in power shortages that cannot be accurately assessed. Therefore, considering the limitations of the above models and aiming at a simultaneous assessment of the power system’s reliability and resilience, this study adopts an iterative-based approach to construct a inter-provincial power system simulation model based on near real-time hourly meteorological data

and calculate the hourly power shortages during normal year and the period of extreme climatic events (snowstorms, sandstorms, heat waves, and droughts), respectively”

2. The treatment of source-sink pairing and inter-province dispatch is interesting. The observations on the challenges posed by the geographic distribution of RE resources are critically important. The study on resilience to extreme weather events is an important contribution. However there are a several major points in the analysis that are questionable.

Authors' Response: Thank you for your positive comments. We have fully responded to your concerns and revised the manuscript. Please see the subsequent one-by-one responses for details.

3. Initially, the authors consider a system with only short-term energy storage, which naturally leads to energy shortage. Coal+CCS is then investigated as a means to avoid shortage and boost reliability. However, it is not clear if coal+CCS if ever investigated in comparison to long-duration energy storage, which is expected to be significantly cheaper by 2050.

Authors' Response: Thank you for your valuable comments. We agree with your point that long-term energy storage technologies (e.g., hydrogen) may also play an important role in avoiding power shortage and boosting power supply reliability. In this revision, we have additionally run a set of scenarios for zero fossil fuel power system with long-term hydrogen storage following your suggestions and compare the results with the fossil fuel power system with CCUS.

The results show that from the power shortage perspective, the implementation of long-term hydrogen storage in a zero fossil fuel power system (i.e., zero fossil fuel with long-term hydrogen storage scenario) can also achieve a rather low power shortage comparable to or even stronger than the fossil fuel with CCS scenario. For instance, with over 4 times dispatch capacity and over 6 hours short-term energy storage, the zero fossil fuel with long-term hydrogen storage scenario could achieve a relatively lower power shortage (<0.03%) than the 16% fossil fuel with CCS scenario with the same dispatch capacity and short-term storage hours (<0.09%). However, from the cost perspective, in order to achieve a 0.1% power shortage (corresponding to China's power system reliability standard), the minimum costs of the power system with the zero fossil fuel with long-term hydrogen storage will be 19.3% (LCOE 3.72%) higher than the power system with 16% abated fossil fuel, which are \$820 billion (LCOE 47.15 USD/MWh) and \$662 billion (LCOE 45.64 USD/MWh), respectively. This shows that the power system involving a certain ratio of fossil fuel with CCS can facilities its reliability standard at a relatively lower cost than complete zero fossil fuel with long-term hydrogen storage. Nevertheless, if the power system reliability standard was further enhanced, the long-term hydrogen storage would play a more important role than abated fossil fuel with CCS. For example, for the 0.01% power shortage, the least LCOE of the power system with zero fossil fuel with long-term hydrogen storage will become 47.19 USD/MWh, which is 2.1% lower than that of the power system with 16% fossil fuel with CCS at the same power shortage (48.21 USD/MWh). Therefore, the long-term hydrogen storage technology is a visible alternative for significantly improving power system reliability to a much higher power system reliability standard than current, and we have highlighted this conclusion in the main text.

In the revised version, we have added the descriptions about the results of the zero fossil fuel power system with long-term hydrogen storage scenario in the ***Power system reliability under high renewables with abated fossil fuel*** section in ***Results*** as follows.

“Considering that long-term energy storage technologies (e.g., hydrogen energy storage, thermal energy storage) may play an essential role in sustaining electricity supply reliability like the role of fossil fuel power with CCUS. We exclusively run a set of scenarios for a zero fossil fuel power system with long-term hydrogen storage for comparison. The results show that, a minimum combination of dispatch capacity and short-term energy storage is required to ensure a rather low power shortage rate under zero fossil power system with long-term hydrogen storage as with all other scenarios. For instance, meeting the 0.1% national entire power shortage rate with the least cost necessitates the construction of 6 hours of short-term energy storage and 4 times of reference dispatch capacity (Supplementary Figure S6a), corresponding to the systemic cost of \$820 billion and the levelized cost of 47.15 USD/MWh, which are respectively 19.3% and 3.2% higher than the scenario of high renewable with 16% abated fossil fuel power generation (Supplementary Figure S7). However, if the national electricity supply reliability standard was further enhanced (e.g., 0.03%), long-term energy storage would play a more important role than abated fossil fuel with CCUS, as shown by a larger area towards the right-upper corner representing the same power shortage level in Supplementary Figure S6a than S6b; or even have a lower levelized cost to meet a higher power shortage standard, e.g., with at least 47.19 USD/MWh and 48.21 USD/MWh for 0.01% national entire power shortage rate under zero fossil fuel with long-term energy storage and high renewable with 16% abated fossil fuel, respectively. Therefore, long-term energy storage technology is a wise option to improve the power system supply reliability when faced with more stringent national shortage rate standard in the future.”

Figure S6. Power shortage and cost comparison under 16% abated fossil fuel and zero fossil fuel with long-term energy storage scenarios. (a and b are the percentages of power shortage for various combinations of dispatch capacity and energy storage hours. A warmer color indicates a more severe power shortage, while a cooler color indicates a less severe shortage. The lines represent the combination of dispatch capacity and energy storage hours for a specific level of power shortage. The red triangle represents the minimum cost required to meet the supply reliability criteria for the given set of scenarios. c represents a cost comparison under different scenarios in which the cost of variable renewable energy (wind power and solar PV) and hydrogen (hydrogen production, storage, and power generation) decreases. d represents the power shortage rate gap between the zero fossil fuel with long-term energy storage scenario and the 16% abated fossil fuel scenario at various combinations of energy storage hours and dispatch capacity.)

Figure S7. Comparison of the total cost and the LCOE of power system under 16% abated fossil fuel and zero fossil fuel with or without long-term energy storage scenarios. (**a** and **b** represent the total cost and the LCOE of power system, respectively.)

In the revised version, we have added the descriptions about inclusion mechanism of the zero fossil fuel power system with long-term hydrogen storage scenario in the *Assumptions of the optimal near-zero power system simulation model* section in *Method* as follows.

“3. The energy storage is assumed to be classified as short-term (within 24 hours) and long-term (without time constraint). Due to higher cost of long-term energy storage, priority is given to short-term energy storage discharging while performing electricity supply.

4. Considering the different cost of various electricity supply, the priorities for the use of electricity supply in each province are: real-time hourly electricity supply from local power generation > real-time hourly dispatch electricity supply via power generation > local short-term energy storage discharging > hourly dispatch via short-term energy storage discharging > local long-term energy storage discharging > hourly dispatch via long-term energy storage discharging.

.....

Inclusion mechanism of Long-term energy storage in the near-zero power system simulation model. Considering the future application prospects hydrogen energy storage⁸⁵, we adopt hydrogen energy storage as the form of long-term energy storage in this study. Following previous prediction of hydrogen consumption⁸⁶, we assume 15% of the end-use energy demand in 2050 (90 EJ⁸⁷) from hydrogen, 70% of which is provided by electrolytic water⁸⁶. The hydrogen production is tended to boost the electricity demand of each province by the same amplification factor. Since the hydrogen energy can be utilized via various ways, it is assumed up to 5% of the national hydrogen energy can be used in the electricity system as long-term electricity storage⁸⁸, and they are allocated to different provinces according to the degrees of power shortage in each province. Like the short-term energy storage, local long-term energy storage discharging and hourly dispatch via long-term energy storage discharging are included in the electricity supply system, with prioritizing to local long-term energy storage discharging.”

4. The authors’ conclusion 1, that CCS would reduce energy shortage compared to 0-FF system is not clearly justified. An 0-FF system can supply energy with no shortage if enough storage is included; the question is how much storage is required and what it will cost. The question that should be answered is whether RE(+storage)+CCUS can provide energy with no major shortages at a lower

cost than RE+storage. The limiting of storage to 12 hr inevitably leads to shortage in an all-RE system – some amount of multi-day storage will always be necessary in such systems to account for periods of low renewable production. The authors should quantify the cost of the 0-FF system that reaches <0.1% shortage (line 237), and state their cost assumptions clearly. In general cost assumptions are not made clear, especially for energy storage technologies.

Authors' Response: Thank you for your valuable comments. We agree with your point that with enough energy storage and adequate dispatch capacity, a zero fossil fuel system will not cause power shortages. Following your suggestion, we have rewrote the conclusion 1 in Introduction section as follows: *A high proportion renewable combined with fossil fuel power generation with CCUS in 2050 would obviously offset the requirement for dispatch capacity and short-term storage, resulting in a lower cost to achieve a certain electricity shortage rate (or power system reliability) compared with zero fossil fuel power generation system, e.g., to achieve the least national entire power shortage rate (the ratio of unmet ideal demand to ideal demand) of 0.07% with zero fossil fuel power generation system at 5 times the reference dispatch and 12 hours energy storage, 20% abated fossil fuel power generation with CCUS only require 3.5-fold dispatch with 8-hour storage, corresponding to 3.0% decrease in levelized cost of energy (LCOE) relative to zero fossil fuel power generation system.*

In addition, we have added the cost comparisons between zero fossil and representative abated fossil fuel power generation scenarios in the results such as *“The lowest national entire power shortage rate of 0.07% could be obtained, with relevant levelized electricity supply costs, power transmission costs, and energy storage costs being 47.24 USD/MWh, \$64.7, and \$71.0 billion (Supplementary Table S3).” “In addition, to approximate the best power system reliability under the zero fossil scenario (i.e., 0.07% electricity shortage with five-fold dispatch and 12-hours electricity storage), for the 5%, 10%, and 20% abated fossil fuel scenarios, corresponding to 1.1%, 2.5%, and 2.8% decrease in LOCE, respectively.” The compositions and sources of these cost are provided in Supplementary Table S3 and Table S5, and corresponding assumptions are provided in the revised sections of Method and Supplementary Methods, with some examples as follows, “To evaluate the economics of the power system, we calculate the cost of the overall power system and its components for all scenarios and select the optimal scenario characterized by the lowest total cost and total power shortage less than 0.1% (i.e., ensuring the general level of electricity supply reliability with 99.9% in Chinese cities). The total cost of the power system includes the cost of non-fossil fuel power generation, the cost of abated fossil fuel power generation with CCUS, the cost of short-term energy storage, the cost of hydrogen energy, and the cost of electricity dispatch (all costs in this study have been adjusted to 2020 constant price)” “The Levelized Cost of Energy (LCOE) of non-fossil fuel power generation is calculated considering technological progress” “As existing coal-fired and gas-fired power plants are considered as CCUS retrofitting candidate in the power system, the construction cost of these plants is not included into the LCOE calculation.” “Based on the technical maturity, development prospect and experts survey of each energy storage technology, we adopt Li-ion battery energy storage to represent short-term energy storage technology and hydrogen energy storage to represent long-term energy storage in the calculation of power system cost. The LCOE of energy storage (short-term and long-term) is calculated considering technological progress.”*

Moreover, we have added the hydrogen as a long-term energy storage approach into the model, and additionally run 475 scenarios to capture the power shortage and systematic cost changes under

zero fossil fuel power generation with long-term energy storage. “The results show that, a minimum combination of dispatch capacity and short-term energy storage is required to ensure a rather low power shortage rate under zero fossil power system with long-term hydrogen storage as with all other scenarios. For instance, meeting the 0.1% national entire power shortage rate with the least cost necessitates the construction of 6 hours of short-term energy storage and 4 times of reference dispatch capacity (Supplementary Figure S6a), corresponding to the systemic cost of \$820 billion and the levelized cost of 47.15 USD/MWh, which are respectively 19.3% and 3.2% higher than the scenario of high renewable with 16% abated fossil fuel power generation (Supplementary Figure S7).”

In the revised version, we have added more descriptions about the zero fossil fuel power system in the **Unmet electricity demand in a zero fossil fuel power system in 2050 China** section in **Results** as follows.

“

..... The lowest national entire power shortage rate of 0.07% could be obtained when the maximum short-term electricity storage and power transmission are fully utilized concurrently (Figure 2a), but this may incur very high economic costs due to the huge capital investment cost of transmission and storage infrastructures, with relevant levelized electricity supply costs, power transmission costs, and energy storage costs being 47.24 USD/MWh, \$64.7, and \$71.0 billion (Supplementary Table S3).

..... In addition, to approximate the best power system reliability under the zero fossil scenario (i.e., 0.07% electricity shortage with five-fold dispatch and 12-hours electricity storage), only 4.5-fold dispatch with 10-hour storage, 3.5-fold dispatch with 8-hour storage, and 3.5-fold dispatch with 8-hour storage are required for the 5%, 10%, and 20% abated fossil fuel scenarios, corresponding to 1.1%, 2.5%, and 2.8% decrease in LOCE, respectively.....”

In the revised version, we have added the power system costing method considering hydrogen energy in the **Cost-competitive analysis of the near-zero power system simulation model** section in **Method** as follows.

“**Cost-competitive analysis of the near-zero power system simulation model.** To evaluate the economics of the power system, we calculate the cost of the overall power system and its components for all scenarios and select the optimal scenario characterized by the lowest total cost and total power shortage less than 0.1% (i.e., ensuring the general level of electricity supply reliability with 99.9% in Chinese cities). The total cost of the power system includes the cost of non-fossil fuel power generation, the cost of abated fossil fuel power generation with CCUS, the cost of short-term energy storage, the cost of hydrogen energy, and the cost of electricity dispatch (all costs in this study have been adjusted to 2020 constant price), as shown in Equation (45).

$$\begin{aligned}
 COST = & LCOE^{nc} \cdot EC^{nc} + LCOE^{hp} \cdot EC^{hp} + LCOE^{pv} \cdot EC^{pv} + LCOE^{on-wp} \cdot EC^{on-wp} \\
 & + LCOE^{off-wp} \cdot EC^{off-wp} + LCOE^{cpcss} \cdot EC^{cp} + LCOE^{ngccs} \cdot EC^{ng} + LCOE^{es} \cdot EC^{es} \\
 & + (H2^{uc} \cdot (1 - PEC^{H2}) + H2^s) \cdot H2^c + LCOE^{H2} \cdot EC^{H2} + \sum_I COST^{UD} \cdot LC_I \cdot Cap_I^D
 \end{aligned} \tag{1}$$

where, $COST$ is the total cost of power system (at 2020 constant price) (USD). $LCOE^{nc}$, $LCOE^{hp}$, $LCOE^{pv}$, $LCOE^{on-wp}$, $LCOE^{off-wp}$, $LCOE^{cpcss}$, $LCOE^{ngccs}$, and $LCOE^{es}$ are the levelized cost of energy (LCOE) of nuclear power, hydropower, solar PV, onshore wind, offshore wind, coal fuel with CCUS, natural gas fuel with CCUS, and short-term energy storage in 2050 (USD/kWh), respectively;

EC^{nc} , EC^{hp} , EC^{pv} , EC^{on-wp} , EC^{off-wp} , EC^{cp} , EC^{ng} , EC^{es} , and EC^{H2} are the total electricity consumption of nuclear power, hydropower, solar PV, onshore wind, offshore wind, coal fuel with CCUS, natural gas fuel with CCUS, short-term energy storage, and hydrogen in 2050 (kWh), respectively (in the long-term energy storage scenario, the additional consumption of additional variable renewable electricity due to the production of hydrogen is captured by EC^{pv} , EC^{on-wp} , and EC^{off-wp}); $H2^{uc}$ is the unit production cost of hydrogen in 2050 (USD/kg). PEC^{H2} is the proportion of electricity costs in hydrogen production (%). $H2^s$ is the cost of hydrogen storage in 2050 (USD/kg). $H2^c$ is the consumption of hydrogen in 2050 (kg). $LCOE^{H2}$ is the LCOE of electricity generated from hydrogen in addition to the cost of fuel in 2050 (USD/kWh). I is the candidate dispatch route in this study, a total of 85. $COST^{UD}$ is the unit dispatch cost (USD/km-GW). LC_I is the length of power transmission route I (km), and Cap_I^D is the maximum hourly utilization capacity of route I (GW). Detailed cost data are show in Supplementary Table S5. The cost calculation of non-fossil fuel power generation, abated fossil fuel power generation, short-term energy storage, hydrogen energy and electricity dispatch are shown in the Supplementary Method. The LCOE of near zero carbon power system is obtained by dividing the total levelized cost of the power system by the electricity consumption in 2050.”

In the revised version, we have added relevant cost descriptions in the **Cost calculation for power generation technologies** section in **Supplementary Information** as follows.

“**The cost of non-fossil fuel power generation.** The Levelized Cost of Energy (LCOE) of non-fossil fuel power generation is calculated considering technological progress, as shown in Equation (S5).

$$LCOE_k = LCOE'_k \cdot (1 - CR_k) \quad (S2)$$

where, $LCOE_k$ is the LCOE of zero fossil fuel power generation technologies k in 2050 (at 2020 constant price) (USD/kWh), where $k=1$ is onshore wind power, $k=2$ is offshore wind power, $k=3$ is solar PV power, $k=4$ is nuclear power, and $k=5$ is hydropower. $LCOE'_k$ is the LCOE of zero fossil fuel power generation technologies k in 2020 (USD/kWh), respectively. CR_k is the LCOE reduction of power generation technologies k in 2050. The detailed values and sources can be found in Supplementary Table S5.

The cost of abated fossil fuel power generation with CCUS. Coal-fired and gas-fired power plants are assumed to operate continuously after the CCUS retrofit. As existing coal-fired and gas-fired power plants are considered as CCUS retrofitting candidate in the power system, the construction cost of these plants is not included into the LCOE calculation. The LCOE of coal-fired power and nature gas-fired power generation with CCUS are calculated by Equation (S6).

$$LCOE_{k'}^{fl-ccs} = \frac{\sum_p (O \& M_{k'} \cdot Cap_p + h_p \cdot Cap_p \cdot \sigma_{k'} \cdot P_{k'} + (CCS^c + CCS^t \cdot L_{k'} + CCS^s) \cdot CER_p)}{\sum_p Q_p} \quad (S3)$$

where, $LCOE_{k'}^{fl-ccs}$ is the LCOE of fossil fuel power generation with CCUS (USD/kWh), where $k'=1$ is coal-fired power generation with CCUS, $k'=2$ is nature gas-fired power generation

with CCUS. $O \& M_k$ is the annual operation and maintenance cost of fossil fuel power generation plants (USD/kW). Cap_p is the installed capacity of fossil fuel power plant p (kW). h_p is the annual operation time of fossil fuel power plant p (h). σ_k is the fossil fuel consumed of fossil fuel power generation plant (t/kWh or m^3/kWh). P_k is the fossil fuel price (USD/t or USD/ m^3). CCS^c is the unit CO_2 capture cost in 2050 (USD/t). CCS^t is the unit CO_2 transport cost in 2050 (USD/t-km). L_k is the weighted average source-sink distance at the current fossil fuel ratio (km), and is taken from the result of the source-sink matching model. CCS^s is the unit CO_2 storage cost in 2050 (USD/t). CER_p is the annual CO_2 capture from fossil fuel power plant p (t), and Q_p is the annual electricity generation of fossil fuel power plant p (kWh). The detailed values can be found in Supplementary Table S5.

The cost of energy storage. Based on the technical maturity, development prospect and experts survey of each energy storage technology, we adopt Li-ion battery energy storage to represent short-term energy storage technology and hydrogen energy storage to represent long-term energy storage in the calculation of power system cost. The LCOE of energy storage (short-term and long-term) is calculated considering technological progress, as shown in Equations (S7)-(S8).

$$LCOE^{es} = LCOE^{es'} \cdot (1 - CR^{es}) \quad (S4)$$

$$H2^s = H2^{s'} \cdot (1 - CR^{H2S}) \quad (S5)$$

where, $LCOE^{es}$ is the LCOE of short-term energy storage in 2050 (at 2020 constant price) (USD/kWh). $LCOE^{es'}$ is the LCOE of short-term energy storage in 2020 (USD/kWh). CR^{es} is the LCOE reduction rate of short-term energy storage in 2050. $H2^s$ is the cost of hydrogen storage in 2050 (at 2020 constant price) (USD/kg). $H2^{s'}$ is the cost of hydrogen storage in 2015 (at 2020 constant price) (USD/kg). CR^{H2S} is the decrease rate in hydrogen cost from 2015 to 2050, taken as 0.175. In addition, the cost of hydrogen production from renewable energy can be referred to Equation (45) in the Method section, and as for hydrogen combustion power generation, we only consider the operation and maintenance cost, assuming that it can use the infrastructure of existing fossil fuel power plants. The detailed values and sources can be found in Supplementary Table S5.”

In the revised version, we have added Table S3 to show the power system costs under zero fossil fuel scenario and 16% abated fossil fuel scenario in **Supplementary Information** as follows.

“Table S3. Power system costs under the zero fossil fuel scenario and the 16% abated fossil fuel scenario

Total cost (billion USD)		Dispatch capacity									
		3		3.5		4		4.5		5	
Energy	storage	Zero fossil fuel	16% abated fossil fuel	Zero fossil fuel	16% abated fossil fuel	Zero fossil fuel	16% abated fossil fuel	Zero fossil fuel	16% abated fossil fuel	Zero fossil fuel	16% abated fossil fuel
5	TT	--	--	--	--	--	--	--	--	--	--
	DS&ES	--	--	--	--	--	--	--	--	--	--
6	TT	--	--	--	--	--	669.1	--	673.1	--	676.9
	DS&ES	--	--	--	--	--	44.2 47.7	--	49.2 46.6	--	53.7 45.8
7	TT	--	--	--	665.7	--	669.4	682.5	673.3	686.0	677.1
	DS&ES	--	--	--	39.0 49.5	--	44.2 47.9	49.9 72.3	49.1 46.8	54.7 70.9	53.7 45.9
8	TT	--	662.4	--	665.7	679.2	669.4	682.5	673.3	686.0	677.2
	DS&ES	--	33.7 51.9	--	39.0 49.6	44.8 74.3	44.2 48.0	49.9 72.3	49.1 46.8	54.7 70.9	53.7 46.0
9	TT	--	662.5	--	665.8	679.2	669.4	682.5	673.3	686.0	677.2
	DS&ES	--	33.7 51.9	--	39.0 49.7	44.8 74.3	44.2 48.0	49.9 72.4	49.1 46.9	54.7 70.9	53.7 46.0
10	TT	--	662.5	--	665.8	679.2	669.4	682.5	673.4	686.0	677.2
	DS&ES	--	33.7 51.9	--	39.0 49.7	44.8 74.3	44.2 48.0	49.9 72.4	49.1 46.9	54.7 70.9	53.7 46.1
11	TT	--	662.5	--	665.8	679.2	669.5	682.5	673.4	686.0	677.2
	DS&ES	--	33.7 52.0	--	39.0 49.7	44.8 74.3	44.2 48.1	49.9 72.4	49.1 46.9	54.7 71.0	53.7 46.1
12	TT	--	662.5	--	665.8	679.2	669.4	682.5	673.4	686.0	677.2

DS&ES -- -- 33.7 52.0 -- -- 39.0 49.7 44.8 74.3 44.1 48.1 49.9 72.4 49.1 46.9 54.7 71.0 53.7 46.1

Note: TT represents the total cost in near-zero power system, DS&ES represents the cost of dispatch capacity and energy storage in near-zero power system, and "--" represents the power shortage of the power system in this scenario exceeds 0.1%.

5. Overall, the methodological choices (especially constraints, economic cost assumptions, and lack of inclusion of Hydrogen as a long-term storage option) may predetermine this finding, rendering the paper biased. Unless, this major issue is addressed, after a major revision – the paper’s value would be limited.

Authors’ Response: Thank you for your valuable suggestions. All these above-mentioned concerns have been carefully addressed in the revised version to ensure objectivity and accuracy of our results. In terms of constraints, we have specified essential constraints in model construction and calculation sections. First, when estimating the variable renewable resource potential, we took full account of spatial geographic constraints such as land use, elevation slope, etc. by using high-resolution grid data. Second, we set the maximum installed capacity constraints for firm non-fossil fuel power. Third, we considered the minimum installed capacity and minimum remaining lifetime for existing fossil fuel power retrofitted with CCUS, and the maximum source-sink matching distance for the CCUS source-sink matching optimization model, and ensured that the cumulative sequestration of each storage site did not exceed its maximum storage potential, and that the annual injection of each storage site did not exceed its maximum annual injection capacity. Fourth, when developing the power system model, we considered several key constraints including the short-term energy storage hours (0-24 hours), the long-term hydrogen energy storage capacity cap (hydrogen as long-term energy storage accounts for 5% of hydrogen demand in all energy sectors¹⁸), and the dispatch flow directions and capacities (a basic capacity of China’s 50 existing inter-provincial dispatch lines and 35 new UHV lines was assumed in this study, while varying from 1-10 times), and the electricity demand (i.e., conventional electricity demand and 15% additional end-use energy demand from hydrogen¹⁹ in long-term energy storage scenarios). Importantly, we have specifically illustrated one overall demand-supply constraint and four groups of electricity supply constraints for the relevant modules of the optimal power system simulation model. Fifth, for power system resilience assessment during extreme weather events, we considered several key constraints including real-time hourly meteorological conditions, as well as renewable energy and fossil fuel installed capacities. All of the above constraints have been added or further emphasized in the revised version.

Moreover, we have added or emphasized the detailed cost assumptions in model construction and calculation sections. First, the theoretical power generation potential and installed capacity potential obtained from the National Renewable Energy Center²⁰ and the National Meteorological Administration²¹ were used as a reference to calculate the hourly resource potential assessment of wind power and solar PV. These reference potentials were estimated with consideration of the technical and economic feasibility of respective renewable resource, reflecting our cost consideration in this high-resolution assessment.

Second, for the CCUS source-sink matching optimization model, we used the minimized transport equivalent (i.e., the CO₂ transport distance multiplied by annual CO₂ storage for each power plant) as one of the objective functions to obtain optimal fossil fuel power generation plant candidates associated with their layout matching with saline aquifer and oil storage sites.

Third, this study involved several crucial assumptions for cost considerations in the process of iteration in the optimal near-zero power system simulation model. For instance, (1) the firm non-fossil fuel power and fossil fuel power had a fixed start-up cost and provided steady-state power supply, whereas variable renewable energy power was intermittent and volatile, thus we ranked their

generation order after steady-state power to reduce the overall power system cost; (2) we prioritized real-time hourly local power generation in terms of cost, followed by power dispatch, energy storage discharging, and dispatch via energy storage discharging; (3) the provinces with the highest renewable energy power generation potential were prioritized for energy storage discharge and subsequent dispatch due to low power generation cost, in order to fully utilize the power output potential, improve utilization hours, and increase power generation in provinces with better resource conditions; (4) energy storage in this study was classified into two categories: short-term energy storage within 24 hours and long-term energy storage, and we prioritized short-term energy storage discharge considering a higher cost of long-term energy storage.

Fourth, in our cost comparison of 10,450 scenarios, we calculated the cost of each type of power generation technology (the cost of non-fossil fuel power generation, the cost of abated fossil fuel power generation), energy storage (the cost of short-term and long-term energy storage), and dispatch infrastructure in 2050, which was adjusted to 2020 constant price to ensure comparability). During the calculation, (1) we considered the future decrease for the cost of all generation technologies, energy storage, and power dispatch caused by technological advances; (2) we assumed abated fossil fuel power generation refers to CCUS retrofitted existing fossil fuel power plants so that only the operation and maintenance (O&M) cost and the CCUS retrofit cost were included when evaluating abated fossil fuel power generation cost; (3) we calculated the costs of renewable energy hydrogen production, hydrogen storage, and hydrogen fuel power generation when evaluating hydrogen energy cost; (4) we considered 85 dispatch lines and calculated their dispatch cost of each dispatch line based on its unique capacity and length, taking the technology learning rate into account.

Overall, after assessing the total power system cost for each scenario, the power system structure with the least cost combination of abated fossil fuel shares, energy storage hours, and dispatch capacity could be determined.

In addition, this study has introduced long-term hydrogen energy storage into the power system model as one set of core scenarios and compared it with the abated fossil fuel power generation with CCUS, as responded in Comment 3 and Comment 4. The results show that zero fossil fuel power generation system with long-term energy storage could achieve the 0.01% power shortage rate at a higher cost than 16% abated fossil fuel power system. When the reliability standard of the power system supply is further enhanced (e.g., 99.97%), however, zero fossil fuel power generation system with long-term energy storage could even perform a lower cost.

In the revised version, we have added the descriptions about the results of the zero fossil fuel power system with long-term hydrogen storage scenario in the ***Power system reliability under high renewables with abated fossil fuel*** section in ***Results*** as follows.

“Considering that long-term energy storage technologies (e.g., hydrogen energy storage, thermal energy storage) may play an essential role in sustaining electricity supply reliability like the role of fossil fuel power with CCUS. We exclusively run a set of scenarios for a zero fossil fuel power system with long-term hydrogen storage for comparison. The results show that, a minimum combination of dispatch capacity and short-term energy storage is required to ensure a rather low power shortage rate under zero fossil power system with long-term hydrogen storage as with all other scenarios. For instance, meeting the 0.1% national entire power shortage rate with the least cost necessitates the construction of

6 hours of short-term energy storage and 4 times of reference dispatch capacity (Supplementary Figure S6a), corresponding to the systemic cost of \$820 billion and the levelized cost of 47.15 USD/MWh, which are respectively 19.3% and 3.2% higher than the scenario of high renewable with 16% abated fossil fuel power generation (Supplementary Figure S7). However, if the national electricity supply reliability standard was further enhanced (e.g., 0.03%), long-term energy storage would play a more important role than abated fossil fuel with CCUS, as shown by a larger area towards the right-upper corner representing the same power shortage level in Supplementary Figure S6a than S6b; or even have a lower levelized cost to meet a higher power shortage standard, e.g., with at least 47.19 USD/MWh and 48.21 USD/MWh for 0.01% national entire power shortage rate under zero fossil fuel with long-term energy storage and high renewable with 16% abated fossil fuel, respectively. Therefore, long-term energy storage technology is a wise option to improve the power system supply reliability when faced with more stringent national shortage rate standard in the future.”

In the revised version, we have added relevant premise assumptions of the CCUS source-sink matching model in the **CCUS source-sink matching model for fossil fuel power generation plants** section in **Method** as follows.

“1. To ensure the economics of CCUS retrofitting fossil fuel power plants, this study assumes that fossil fuel power plants with installed capacities greater than 300 MW⁸² and remaining lifetime longer than 15 years can be retrofitted with CCUS projects.

.....

The CCUS source-sink matching model was solved using Python and ArcGIS software, where the objective functions are to maximize CO₂ storage and minimize CO₂ transport equivalent, as shown in Equations (26).

$$\begin{cases} \max TS = \sum_p \sum_{s \in S_x} (X_{p,s} \cdot S_{p,s}) \\ \min TT = \sum_p \sum_{s \in S_x} (X_{p,s} \cdot D_{p,s} \cdot S_{p,s}) \end{cases} \quad (26)$$

where, TS and TT denote total CO₂ storage (t) and total CO₂ transport equivalent (t-km) attributed to all possible source-sink matching results. S_x is four various sequestration options, where S_1 is the set of onshore oil field, S_2 is the set of offshore oil field, S_3 is the set of onshore saline aquifer, S_4 is the set of offshore saline aquifer. $X_{p,s}$ is a binary variable with $X_{p,s} = 1$ indicating power plant p and storage site s are matched successfully, otherwise, $X_{p,s} = 0$. $S_{p,s}$ is the annual CO₂ storage amount of power plant p when power plant p and storage site s are matched successfully, and $D_{p,s}$ is the transport distance from power plant p and storage site s .”

In the revised version, we have added relevant premise assumptions and rules of the power system in the **Assumptions of the optimal near-zero power system simulation model** section in **Method** as follows.

“1. Due to the difficulty of obtaining grid transmission lines within a province and simplifying the model to make the optimization issue manageable, the power system simulation is assumed to only consider inter-provincial electricity dispatch, i.e., electricity within each province can be dispatched flexibly so that each province is considered as a single node. Meanwhile, the same type of power generation units within a province is considered as a single unit.

2. Considering the grid’s preference for various electricity supply source and the cost of power generation technology, the power system is assumed to prioritize the use of steady-state power sources,

and thus the priorities for the adoption of power generation technologies are: nuclear power > hydropower > abated fossil fuel power > variable renewable energy power (including onshore wind power, offshore wind power, and solar PV).

3. The energy storage is assumed to be classified as short-term (within 24 hours) and long-term (without time constraint). Due to higher cost of long-term energy storage, priority is given to short-term energy storage discharging while performing electricity supply.

4. Considering the different cost of various electricity supply, the priorities for the use of electricity supply in each province are: real-time hourly electricity supply from local power generation > real-time hourly dispatch electricity supply via power generation > local short-term energy storage discharging > hourly dispatch via short-term energy storage discharging > local long-term energy storage discharging > hourly dispatch via long-term energy storage discharging.

5. In order to reduce power shortage, real-time hourly electricity dispatch is prioritized for supplying the province with the most severe power shortage where local real-time hourly electricity supply cannot meet the province's electricity demand and dispatch lines are available.

6. In order to improve utilization hours and increase power generation in provinces with better resource conditions, provinces with the highest remaining variable renewable energy power generation potential after meeting their local real-time hourly electricity supply and real-time hourly electricity dispatch are prioritized for energy storage discharging and subsequent dispatch.

7. In terms of short-term energy storage charging, only variable renewable energy storage is examined, and different storage hours are assumed as the upper limit for continuous energy storage charging in each province, without considering the hours constraint of energy storage discharging.

Inclusion mechanism of Long-term energy storage in the near-zero power system simulation model. Considering the future application prospects hydrogen energy storage⁸⁵, we adopt hydrogen energy storage as the form of long-term energy storage in this study. Following previous prediction of hydrogen consumption⁸⁶, we assume 15% of the end-use energy demand in 2050 (90 EJ⁸⁷) from hydrogen, 70% of which is provided by electrolytic water⁸⁶. The hydrogen production is tended to boost the electricity demand of each province by the same amplification factor. Since the hydrogen energy can be utilized via various ways, it is assumed up to 5% of the national hydrogen energy can be used in the electricity system as long-term electricity storage⁸⁸, and they are allocated to different provinces according to the degrees of power shortage in each province. Like the short-term energy storage, local long-term energy storage discharging and hourly dispatch via long-term energy storage discharging are included in the electricity supply system, with prioritizing to local long-term energy storage discharging.

.....

Constraints of the optimal near-zero power system simulation model. The overall constraint of optimal near-zero power system is to ensure that the hourly electricity supply is less than the electricity demand, as shown in Equations (34).

$$ES_{n,t} \leq IED_{n,t} \quad (34)$$

where, $ES_{n,t}$ is total electricity supply at hour t in province n , including four sources of electricity supply. Based on the hourly electricity supply sources, the optimal near-zero power simulation model is divided into four modules: local real-time hourly electricity supply via power generation module, real-time hourly dispatch electricity supply via power generation module, local energy storage discharging

electricity supply module, and hourly dispatch via energy storage discharging electricity supply module; note that energy storage in the above modules does not include the long-term energy storage.”

In the revised version, we have added the power system costing method considering hydrogen energy in the **Cost-competitive analysis of the near-zero power system simulation model** section in **Method** as follows.

“**Cost-competitive analysis of the near-zero power system simulation model.** To evaluate the economics of the power system, we calculate the cost of the overall power system and its components for all scenarios and select the optimal scenario characterized by the lowest total cost and total power shortage less than 0.1% (i.e., ensuring the general level of electricity supply reliability with 99.9% in Chinese cities). The total cost of the power system includes the cost of non-fossil fuel power generation, the cost of abated fossil fuel power generation with CCUS, the cost of short-term energy storage, the cost of hydrogen energy, and the cost of electricity dispatch (all costs in this study have been adjusted to 2020 constant price), as shown in Equation (45).

$$\begin{aligned}
COST = & LCOE^{nc} \cdot EC^{nc} + LCOE^{hp} \cdot EC^{hp} + LCOE^{pv} \cdot EC^{pv} + LCOE^{on-wp} \cdot EC^{on-wp} \\
& + LCOE^{off-wp} \cdot EC^{off-wp} + LCOE^{cpccs} \cdot EC^{cp} + LCOE^{ngccs} \cdot EC^{ng} + LCOE^{es} \cdot EC^{es} \\
& + (H2^{uc} \cdot (1 - PEC^{H2}) + H2^s) \cdot H2^c + LCOE^{H2} \cdot EC^{H2} + \sum_I COST^{UD} \cdot LC_I \cdot Cap_I^D
\end{aligned} \quad (45)$$

where, $COST$ is the total cost of power system (at 2020 constant price) (USD). $LCOE^{nc}$, $LCOE^{hp}$,

$LCOE^{pv}$, $LCOE^{on-wp}$, $LCOE^{off-wp}$, $LCOE^{cpccs}$, $LCOE^{ngccs}$, and $LCOE^{es}$ are the levelized cost of energy (LCOE) of nuclear power, hydropower, solar PV, onshore wind, offshore wind, coal fuel with CCUS, natural gas fuel with CCUS, and short-term energy storage in 2050 (USD/kWh), respectively;

EC^{nc} , EC^{hp} , EC^{pv} , EC^{on-wp} , EC^{off-wp} , EC^{cp} , EC^{ng} , EC^{es} , and EC^{H2} are the total electricity consumption of nuclear power, hydropower, solar PV, onshore wind, offshore wind, coal fuel with CCUS, natural gas fuel with CCUS, short-term energy storage, and hydrogen in 2050 (kWh), respectively (in the long-term energy storage scenario, the additional consumption of additional variable renewable electricity due to the production of hydrogen is captured by EC^{pv} , EC^{on-wp} , and EC^{off-wp});

$H2^{uc}$ is the unit production cost of hydrogen in 2050 (USD/kg). PEC^{H2} is the proportion of electricity costs in hydrogen production (%). $H2^s$ is the cost of hydrogen storage in 2050 (USD/kg).

$H2^c$ is the consumption of hydrogen in 2050 (kg). $LCOE^{H2}$ is the LCOE of electricity generated from hydrogen in addition to the cost of fuel in 2050 (USD/kWh). I is the candidate dispatch route in this study,

a total of 85. $COST^{UD}$ is the unit dispatch cost (USD/km-GW). LC_I is the length of power transmission route I (km), and Cap_I^D is the maximum hourly utilization capacity of route I (GW).

Detailed cost data are show in Supplementary Table S5. The cost calculation of non-fossil fuel power generation, abated fossil fuel power generation, short-term energy storage, hydrogen energy and electricity dispatch are shown in the Supplementary Method. The LCOE of near zero carbon power system is obtained by dividing the total levelized cost of the power system by the electricity consumption

in 2050.”

In the revised version, we have added relevant cost descriptions in the **Cost calculation for power generation technologies** section in **Supplementary Methods** as follows.

“The cost of non-fossil fuel power generation. The Levelized Cost of Energy (LCOE) of non-fossil fuel power generation is calculated considering technological progress, as shown in Equation (S5).

$$LCOE_k = LCOE'_k \cdot (1 - CR_k) \quad (S5)$$

where, $LCOE_k$ is the LCOE of zero fossil fuel power generation technologies k in 2050 (at 2020 constant price) (USD/kWh), where $k = 1$ is onshore wind power, $k = 2$ is offshore wind power, $k = 3$ is solar PV power, $k = 4$ is nuclear power, and $k = 5$ is hydropower. $LCOE'_k$ is the LCOE of zero fossil fuel power generation technologies k in 2020 (USD/kWh), respectively. CR_k is the LCOE reduction rate of power generation technologies k in 2050. The detailed values and sources can be found in Supplementary Table S5.

The cost of abated fossil fuel power generation with CCUS. Coal-fired and gas-fired power plants are assumed to operate continuously after the CCUS retrofit. As existing coal-fired and gas-fired power plants are considered as CCUS retrofitting candidate in the power system, the construction cost of these plants is not included into the LCOE calculation. The LCOE of coal-fired power and nature gas-fired power generation with CCUS are calculated by Equation (S6).

$$LCOE_{k'}^{fl-ccs} = \frac{\sum_p (O \& M_{k'} \cdot Cap_p + h_p \cdot Cap_p \cdot \sigma_{k'} \cdot P_{k'} + (CCS^c + CCS^t \cdot L_{k'} + CCS^s) \cdot CER_p)}{\sum_p Q_p} \quad (S6)$$

where, $LCOE_{k'}^{fl-ccs}$ is the LCOE of fossil fuel power generation with CCUS (USD/kWh), where $k' = 1$ is coal-fired power generation with CCUS, $k' = 2$ is nature gas-fired power generation with CCUS. $O \& M_{k'}$ is the annual operation and maintenance cost of fossil fuel power generation plants (USD/kW). Cap_p is the installed capacity of fossil fuel power plant p (kW). h_p is the annual operation time of fossil fuel power plant p (h). $\sigma_{k'}$ is the fossil fuel consumed of fossil fuel power generation plant (t/kWh or m^3/kWh). $P_{k'}$ is the fossil fuel price (USD/t or USD/ m^3). CCS^c is the unit CO_2 capture cost in 2050 (USD/t). CCS^t is the unit CO_2 transport cost in 2050 (USD/t·km). $L_{k'}$ is the weighted average source-sink distance at the current fossil fuel ratio (km), and is taken from the result of the source-sink matching model. CCS^s is the unit CO_2 storage cost in 2050 (USD/t). CER_p is the annual CO_2 capture from fossil fuel power plant p (t), and Q_p is the annual electricity generation of fossil fuel power plant p (kWh). The detailed values can be found in Supplementary Table S5.

The cost of energy storage. Based on the technical maturity, development prospect and experts survey of each energy storage technology, we adopt Li-ion battery energy storage to represent short-term energy storage technology and hydrogen energy storage to represent long-term energy storage in the calculation of power system cost. The LCOE of energy storage (short-term and long-term) is calculated considering technological progress, as shown in Equations (S7)-(S8).

$$LCOE^{es} = LCOE^{es'} \cdot (1 - CR^{es}) \quad (S7)$$

$$H2^s = H2^{s'} \cdot (1 - CR^{H2^s}) \quad (S8)$$

where, $LCOE^{es}$ is the LCOE of short-term energy storage in 2050 (at 2020 constant price) (USD/kWh). $LCOE^{es'}$ is the LCOE of short-term energy storage in 2020 (USD/kWh). CR^{es} is the LCOE reduction rate of short-term energy storage in 2050. $H2^s$ is the cost of hydrogen storage in 2050 (at 2020 constant price) (USD/kg). $H2^{s'}$ is the cost of hydrogen storage in 2015 (at 2020 constant price) (USD/kg). CR^{H2S} is the decrease rate in hydrogen cost from 2015 to 2050, taken as 0.175. In addition, the cost of hydrogen production from renewable energy can be referred to Equation (45) in the Method section, and as for hydrogen combustion power generation, we only consider the operation and maintenance cost, assuming that it can use the infrastructure of existing fossil fuel power plants. The detailed values and sources can be found in Supplementary Table S5.”

Below are sequential comments for consideration:

6. Pg. 1 Ln 31 Abstract - reference to “power shortage” is unclear. Electricity grids must match demand and supply at any time - shortage implies load-shedding (black-outs). The way that this is used here should be explained.

Authors’ Response: Thank you for your careful reminding. It is true that the electricity grid should always match power supply and demand; and when the electricity demand cannot be met, a load shedding or blackout occurs. In order to ensure an accurate meaning, we have defined the concept of power shortage rate as the share of unmet ideal electricity demand in 2050, which could be measured by the unmet ideal demand divided by the total ideal demand of that year. In addition, depending on the subject level and time scale, the concept of power shortage rate includes (1) national entire power shortage rate, defined as the cumulative gap between the provincial overall hourly power supply not meeting its corresponding ideal hourly electricity demand divided by the national overall ideal electricity demand; (2) provincial entire power shortage rate, defined as the cumulative gap for a specific province between the overall hourly power supply not meeting its corresponding ideal hourly electricity demand divided by the provincial overall ideal electricity demand; (3) national hourly power shortage rate, defined as the gap between the national hourly power supply not meeting its corresponding ideal hourly electricity demand divided by the national ideal hourly electricity demand; (4) provincial hourly power shortage rate, defined as the gap for a specific province between the hourly power supply not meeting its corresponding ideal hourly electricity demand divided by the provincial ideal hourly electricity demand. Here, in the abstract, it refers to the national entire power shortage rate.

In the revised version, we have added the definition of power shortage in **Abstract** as follows.

“.....Results indicate that allowing 20% abated fossil power in the overall power system could ease the national entire power shortage rate (the share of unmet ideal electricity demand) by up to 9.0 percentages in 2050 compared with zero fossil system (from 21.8% to 12.8%).....”

In the revised version, we have also added the definition of power shortage in **Introduction** as follows.

“.....e.g., to achieve the least national entire power shortage rate (the ratio of unmet ideal demand to ideal demand) of 0.07% with zero fossil fuel power generation system at 5 times the reference dispatch and 12 hours energy storage, 20% abated fossil fuel power generation with CCUS only require 3.5-fold dispatch with 8-hour storage, corresponding to 3.0% decrease in levelized cost of energy (LCOE) relative to zero fossil fuel power generation system.....”

In the revised version, we have added a **Power shortage rate definition** section including equations for the four types of power shortage rates in **Method** as follows.

“Power shortage rate definition. This study defined four categories of power shortage rates including national entire power shortage rate, provincial entire power shortage rate, national hourly power shortage rate, and provincial hourly power shortage rate as follows.

First, the national entire power shortage rate represents the cumulative gap between the provincial hourly electricity supply not meeting its ideal hourly electricity demand divided by the national overall ideal electricity demand, as shown in Equation (28).

$$EPS^N = \frac{\sum_n \sum_t (IED_{n,t} - ES_{n,t})}{\sum_n \sum_t IED_{n,t}} \quad (28)$$

where, EPS^N represents the national entire power shortage rate. $ES_{n,t}$ is the electricity supply at hour t in province n , including local real-time hourly electricity supply via power generation, real-time hourly dispatch electricity supply via power generation, local energy storage discharging electricity supply, and hourly dispatch via energy storage discharging electricity supply as shown in Equation (33), $IED_{n,t}$ is the ideal electricity demand in 2050 at hour t in province n , as shown in Equation (24).

Second, the provincial entire power shortage rate represents the cumulative gap for a specific province between the hourly electricity supply not meeting its ideal hourly electricity demand divided by the provincial overall ideal electricity demand, as shown in Equation (29).

$$EPS_n^P = \frac{\sum_t (IED_{n,t} - ES_{n,t})}{\sum_t IED_{n,t}} \quad (29)$$

where, EPS_n^P represents the entire power shortage rate in province n .

Third, the national hourly power shortage rate represents the gap between the national hourly electricity supply not meeting its ideal hourly electricity demand divided by the national ideal hourly electricity demand, as shown in Equation (30).

$$HPS_t^N = \frac{\sum_n (IED_{n,t} - ES_{n,t})}{\sum_n IED_{n,t}} \quad (30)$$

where, HPS_t^N represents the national hourly power shortage rate at hour t .

Finally, the provincial hourly power shortage rate represents the gap for a specific province between the hourly electricity supply not meeting its ideal hourly electricity demand divided by the provincial ideal hourly electricity demand, as shown in Equation (31).

$$HPS_{n,t}^P = \frac{IED_{n,t} - ES_{n,t}}{IED_{n,t}} \quad (31)$$

where, $HPS_{n,t}^P$ represents the hourly power shortage rate at hour t in province n .”

7. Pg. 3 Ln 66 “or high voltage” - missing transmission lines which should be added.

Authors' Response: Thank you for your careful reminder. According to your suggestion, we have changed “or high voltage” to “or high voltage transmission lines”.

In this revised version, we have modified this sentence in **Introduction** as follows.

“.....abated fossil fuels with CCUS in high renewable power system could partially replace variable renewable energy and lower the associated need for construction of electricity storage or high voltage transmission lines.....”

8. Pg. 3 Ln 78 - It is true that many IAMs include CCUS but the reasons for this may be structural modeling constraints that are implicit and unrealistic cost assumptions regarding RE integration (cf. Kaya, A., Csala, D. & Sgouridis, S. Constant elasticity of substitution functions for energy modeling in general equilibrium integrated assessment models: a critical review and recommendations. *Climatic Change* 43, 225–14 (2017)).

Authors' Response: Thank you for your valuable comments. We agree with you and the views expressed in this reference. As mentioned in this reference, using constant elasticity of substitution (CES) functions in general equilibrium integrated assessment models (GE-IAMs) for the substitution of technical factor inputs (e.g., replacing fossil fuels) fails to match historically observed patterns in energy transition dynamics. And this method of substitution is also very sensitive to the structure of CES implementation (nesting) and parameter choice. For example, many top-down general equilibrium models do not consider energy storage technology when depicting the CES function of the power sector, and therefore treat intermittent power generation technologies (e.g., wind power and solar PV) as an imperfect substitute for other power generation technologies, which distorts the costs of wind power and solar PV to some extent as you commented. Indeed, this assumption in GE-IAMs will result in an overestimation of the share of CCS-related generation technologies. However, these are problems that will exist in the top-down models based on general equilibrium theory. While, for bottom-up IAMs or power system optimization models, which are generally based on linear programming in operational research, technology choices therein are completely determined by the cost competition between various technologies and do not involve the substitution between factor inputs. Therefore, there is no distortion of the cost of variable renewable generation technologies in these bottom-up models, which allows for the realization of a 100% renewable electricity system. Nevertheless, there are still many bottom-up models that introduce CCS-related technologies due to their cost effectiveness, such as GCAM, TIMES, US-REGEN^{22, 23, 24}.

In addition, we believe that CCS-related technology should be considered in China's power system model, not because of model assumptions, but considering the actual situation in China. First, China has a large number of young coal-fired power plants, with over 80% operating for less than 15 years²⁵. If CCS-related technologies are not considered, these coal-fired units will be phased out before their lifetimes under the strict CO₂ emission mitigation targets, resulting in huge stranded assets. Second, the large-scale phase-out of coal-fired power plants will also lead to a drop in employment in coal-fired power generation and coal mining industries²⁶, causing certain social employment issues. Therefore, it is critical to incorporate CCS-related technologies for a just transformation of China's power system under the carbon neutral target.

In the revised version, we have revised relevant descriptions about IAMs in **Introduction** as follows.

“.....although these are caused in part by IAMs’ specific modelling assumptions (e.g., the general equilibrium theory based IAMs)³⁷.....”

In the revised version, we have added the reasons for choosing bottom-up model based on linear programming in the **Considerations for the optimal near-zero power system simulation model** section in **Supplementary Methods** as follows.

“Considerations for the optimal near-zero power system simulation model. Considering the general equilibrium-based top-down model may distort the costs of intermittent renewable power when modelling the power system, which is limited by the model assumption of a constant elasticity of substitution production function⁶, the cost-optimized bottom-up model based on linear programming is more appropriate for the scientific questions in this study.....”

9. Pg. 3 Ln 79 - the sentence that CCUS “avoids carbon lock-in” seems a little out of place. CCUS by definition is a result of “carbon lock-in”. Need to rethink/rephrase.

Authors’ Response: Thank you for your careful check. We agree with you that CCUS technology cannot avoid the fact of carbon lock-in itself. Instead, once equipped with existing facilities, CCUS technology could provide a solution for carbon lock-in of fossil fuel power plants.

In this revised version, we have modified “avoids carbon lock-in” to “provides a viable solution for carbon lock-in” in this sentence in **Introduction** as follows.

“.....as the CCUS option tends to offer lower costs in reducing carbon emissions than nuclear and renewable ones in these scenarios^{15, 25, 31, 32} and provides a viable solution for carbon lock-in of fossil fuel energy infrastructure^{8, 33, 34}.....”

10. Pg. 3 Ln 93-95 the same observation is made as in Ln 78. One should consider systemic biases in modeling. Since the IAM models have underlying structural constraints that prevent them from opting to 100% RE systems, your following statement (that few studies compared both) is correct. There is nevertheless an interesting comparison on Energy Return basis available that provides strong arguments against the use of CCUS for power systems (cf. 1.Sgouridis, S., Carbajales-Dale, M., Csala, D., Chiesa, M. & Bardi, U. Comparative net energy analysis of renewable electricity and carbon capture and storage. Nature Energy 4, 456–465 (2019).)

Authors’ Response: Thanks for the comment and the valuable literature offering us, and we have read your recommended literature carefully. By comparing the energy return on energy invested (EROEI) of fossil-fuel-based power plants with CCS and dispatchable scalable renewable electricity with storage, this literature concludes that renewables plus storage provide a more energetically effective approach to mitigating climate change than constructing CCS fossil-fuel power stations, thus indicating that CCS technology is not a key technology choice for climate mitigation²⁷. The methodology and conclusions of this literature are clear and reliable, but the following points should be noticed.

First, it is important to note that the fossil-fuel-based power plants with CCS considered in this literature are newly built, whereas in our study, considering China's reality that a large number of young coal-fired power plants with over 80% being operational for less than 15 years²⁵, CCS is more likely to retrofit existing coal-fired power plants to achieve deep decarbonization, which would result in significant capital cost savings from less stranded assets compared with newly built CCS facilities (as the response to Comment 8). Second, this literature focuses on two single types of generation technologies, with little attention given to the subsequent reliability and resilience of the whole power system. Third, although this literature treats renewable electricity as dispatchable and stable power source by incorporating energy storage, the impact of high percentage renewables on the whole power system is more than just power intermittency. For example, a high proportion of renewable electricity will lower power system inertia^{28,29}, which will have a direct effect on the rate of change of frequency (RoCoF) and the minimum frequency when power system breakdowns occur⁴.

In the revised version, we have added the reasons for choosing bottom-up model based on linear programming in the *Considerations for the optimal near-zero power system simulation model* section in *Supplementary Methods* as follows.

“Considerations for the optimal near-zero power system simulation model. Considering the general equilibrium-based top-down model may distort the costs of intermittent renewable power when modelling the power system, which is limited by the model assumption of a constant elasticity of substitution production function⁶, the cost-optimized bottom-up model based on linear programming is more appropriate for the scientific questions in this study.....”

11. Pg. 6 Ln 155-158 - some explanation on how the RE potentials have been calculated is needed and future demand by province has been calculated. What do they mean in terms of installed capacities?

Authors' Response: Thank you for your helpful suggestions and questions. In this study, we have included two different definitions for “variable RE potentials”, i.e., variable renewable energy installed capacity potential and variable renewable energy power generation potential, where the variable renewable energy installed capacity potential was determined by the maximum power output in all hours of the year, and the variable renewable energy power generation potential was calculated by adding up all hours of power output of the year. Specifically, to calculate the variable renewable energy potential in each province, we first obtained hourly wind speed (1980-2019) from the MERRA-2 database, and derived hourly temperatures and solar radiation intensity data at a specific tilt angle (2010-2019) using the Meteonorm software for 31 provinces in China. Combining these hourly meteorological data, the real-time hourly power output of onshore wind power, offshore wind power, and solar PV in each province could be calculated by using the generating power curves of wind turbines^{20,21} and HOMER model^{30,31,32}, and then calibrated by the official statistics from the Renewable Energy Data Book 2019 and the National Meteorological Administration^{20,21}. Then, the maximum power output in all hours of the year was selected as the variable renewable energy installed capacity potential in each province, and all hours of power output of the year were summed up to obtain the variable renewable energy power generation potential in each province. For other renewable energy (e.g., hydro, biomass), we estimated the two types of potentials by predicted annual installed capacity and power generation in 2050.

In terms of future electricity demand in 2050 for each province, we constructed a fixed-effects multiple regression model based on electricity consumption per capita and various independent variables data for 30 provinces from 1995 to 2019. By using the estimated regression model, the total electricity demand in 2050 for each province was further predicted based on the model's future prediction of the independent variable (including GDP per capita, value of heating degree days, value of cooling degree days, value added of secondary industry to GDP, and electricity price index), combined with China's future population prediction.

However, data availability issues troubled further forecasting of real-time hourly electricity demand by province. To overcome this difficulty, we first collected the actual hourly electricity consumption of 8760 hours in Anhui Province in 2019, the monthly electricity consumption of 30 provinces in 2019, and the typical hourly electricity load on workdays and non-workdays of 30 provinces in 2019³³. We then calculated the electricity consumption of 8760 hours for 30 provinces of China in 2019 by using the electricity consumption curve of Anhui Province and combining it with the typical hourly electricity load of 30 provinces. Finally, by using the amplification factor obtained for total provincial electricity demand in 2050 relative to 2019, the real-time hourly electricity demand in 2050 could be estimated for each province. The calculations process and equations above have been added or reorganized in the revised manuscript to make them clearer shown as follows.

In this revised version, we have modified relevant descriptions about variable renewable energy potentials in the *Unmet electricity demand in a zero fossil fuel power system in 2050 China* section in **Results** as follows.

“Unmet electricity demand in a zero fossil fuel power system in 2050 China

By 2050, China's non-fossil energy (onshore wind, offshore wind, solar PV, hydropower and nuclear) power generation potential (equal to the sum of corresponding hourly maximum power output potential) reaches 90,076 billion kWh, of which variable renewables (solar and wind in this study) account for 96%.....”

“.....The model configures the future power system compositions based on the estimated hour-by-hour power output potential of different power source types.....”

In the revised version, we have added relevant calculation and description about variable renewable energy potentials in the *Wind power potential assessment, and Solar PV potential assessment* sections in **Method** as follows.

“The wind power installed capacity potential in each province is determined by the maximum power output of a year, and the wind power generation potential in each province is calculated by the total cumulative power output of each hour in the year, as shown in Equations (3)-(4).

$$WP_n^{IC} = \max WP_{n,t} \quad (3)$$

$$G_n^W = \sum_t WP_{n,t} \quad (4)$$

where, WP_n^{IC} denotes the potential of wind power installed capacity of province n (MW), G_n^W denotes the annual potential of wind power generation of province n (MWh).

.....

Like wind power, the solar PV installed capacity potential in each province is determined by the

maximum power output of the year, and the solar PV generation potential in each province is calculated by the total power output of the year, as shown in Equations (10)-(13).

$$PV_n^{IC1} = \max PV_{n,t}^{V1} \quad (10)$$

$$PV_n^{IC2} = \max PV_{n,t}^{V2} \quad (11)$$

$$G_n^{V1} = \sum_t PV_{n,t}^{V1} \quad (12)$$

$$G_n^{V2} = \sum_t PV_{n,t}^{V2} \quad (13)$$

where, PV_n^{IC1} and PV_n^{IC2} respectively represent the potential of solar PV installed capacity in province n by the two methods (W), G_n^{V1} and G_n^{V2} respectively represent the annual potential of solar PV generation in province n by the two methods (Wh)."

In the revised version, we have added the formula for calculating future hourly electricity demand in the **Provincial real-time hourly electricity demand predictions** section in **Method** as follows.

“Provincial real-time hourly electricity demand predictions. Compared with previous literature on electricity demand predictions for Chinese power system models^{43, 56, 77, 78}, we use a more accurate and refined method to predict the hourly electricity demand of each province in China in 2050 as follows.

First, according to the monthly electricity consumption and the typical hourly electricity load on workdays and non-workdays in each province in 2019⁷⁹, the corresponding hourly electricity load on workdays and non-workdays for each month are calculated by Equations (20)-(21). Second, considering the number of workdays and non-workdays in each month, the average hourly baseline electricity load at same hour for each month is calculated by Equation (22). Third, due to data availability, Anhui province was selected as the reference province, and its hourly actual electricity consumption in 2019 was used to calculate the proportion of electricity load for each hour of each day of each month, compared to the electricity load for the same hour of the same month, as shown in Equation (23). Then the hourly electricity demand for each province for the whole year is calculated referring to the hourly electricity demand variation proportion of Anhui province by Equation (24). Finally, the real-time hourly electricity demand for each province in 2050 is predicted using the multipliers of the annual provincial electricity demand relative to 2019. The specific equations are as follows.

The hourly electricity load of workdays and non-workdays for each month of each province is shown in Equations (20)-(21).

$$EC_{n,m,t}^w = TEL_{n,t}^w \cdot \frac{EC_{n,m}}{\sum_t (TEL_{n,t}^w \cdot D_m^w + TEL_{n,t}^{nw} \cdot D_m^{nw})} \quad (20)$$

$$EC_{n,m,t}^{nw} = TEL_{n,t}^{nw} \cdot \frac{EC_{n,m}}{\sum_t (TEL_{n,t}^w \cdot D_m^w + TEL_{n,t}^{nw} \cdot D_m^{nw})} \quad (21)$$

where, $EC_{n,m,t}^w$ is the electricity consumption at hour t of workdays for month m of province n (MWh), $TEL_{n,t}^w$ is the typical electricity load at hour t of workdays of province n (MW), $EC_{n,m}$ is the electricity consumption for month m of province n in 2019 (MWh), D_m^w is the number of workdays in month m ,

D_m^{nw} is the number of non-workdays in month m , $EC_{n,m,t}^{nw}$ is the electricity load at hour t of non-workdays for month m of province n (MWh), $TEL_{n,t}^{nw}$ is the typical electricity load at hour t of non-workdays of province n (MW).

The average hourly baseline electricity load for each province is shown in Equation (22).

$$AEC_{n,m,t} = \frac{EC_{n,m,t}^w \cdot D_m^w + EC_{n,m,t}^{nw} \cdot D_m^{nw}}{D_m^{all}} \quad (22)$$

where, $AEL_{n,m,t}$ represents the average electricity consumption at hour t in month m of province n (MWh).

The variation ratio of actual hourly electricity consumption to average electricity consumption in Anhui province is shown in Equation (23).

$$PP_{m,d,t} = EC_{m,d,t} / \left(\frac{\sum_d EC_{m,d,t}}{D_m^{all}} \right) \quad (23)$$

where, $PP_{m,d,t}$ is the Anhui province's variation proportion of actual electricity consumption in 2019 to the average electricity demand at hour t on day d in month m , $EC_{m,d,t}$ represents the actual electricity consumption in 2019 of the Anhui province at hour t on day d in month m (MWh).

The hourly electricity demand for the other province is shown in Equation (24).

$$ED_{n,m,d,t} = AEC_{n,m,t} \cdot PP_{m,d,t} \quad (24)$$

where, $ED_{n,m,d,t}$ represents the electricity demand of the province n at hour t on day d in month m (MWh).

Finally, the real-time hourly electricity demand in 2050 for each province can be estimated using the multiplier derived from provincial electricity demand in 2050 relative to 2019 estimated by econometric models."

12. Pg. 6 Ln 159 - 164. The concept of "shortage" is introduced. It seems to refer to RE power supply vs demand by province but it is not properly defined. Please define carefully and explain the importance of this metric.

Authors' Response: Thank you for your careful reminding. We are sorry for not clearly interpreting the concept of "shortage" in the original version. Power shortage is an important metric for assessing the reliability and resilience of China's power system proposed in this study, which is used to reflect the relationship between electricity supply and ideal electricity demand. Thanks to your reminding, we recognize that the "shortage" in this sentence is not exactly the same as our definition of power shortage. Therefore, to avoid conceptual confusion, we have avoided using the concept of "shortage" here and use a direct expression to represent this ratio.

In addition, in order to distinguish all the power shortages described in this article, we classify the power shortage rates into four types according to the subject level and time scale, including (1) national entire power shortage rate, defined as the cumulative gap between the provincial overall

hourly electricity supply not meeting its corresponding ideal hourly electricity demand divided by the national overall ideal electricity demand; (2) provincial entire power shortage rate, defined as the cumulative gap for a specific province between the overall hourly electricity supply not meeting its corresponding ideal hourly electricity demand divided by the provincial overall ideal electricity demand; (3) national hourly power shortage rate, defined as the gap between the national hourly electricity supply not meeting its corresponding ideal hourly electricity demand divided by the national ideal hourly electricity demand; (4) provincial hourly power shortage rate, defined as the gap for a specific province between the hourly electricity supply not meeting its corresponding ideal hourly electricity demand divided by the provincial ideal hourly electricity demand. We also emphasize the importance of this metric in the definition of reliability and resilience of power system.

In this revised version, we have added the clear definition of power shortage rate in **Introduction** as follows.

“.....e.g., to achieve the least national entire power shortage rate (the ratio of unmet ideal demand to ideal demand)”

In the revised version, we have added the relationship between power shortage rate and reliability in the **The reliability and resilience of the power system** section in **Method** as follows.

“Both the reliability and resilience of the power system could be measured by the power shortage. Referring to previous research¹¹ and national standards for electricity supply (e.g., 99.9% for cities in China)¹⁸, reliability is defined as the ability of all generating units connected to the grid to meet the electricity demand during normal years (i.e., without extreme weather events), quantified by one minus the power shortage rate. Resilience mainly measures the power system’s ability to withstand power shortages and restore electricity supply in a timely manner during extreme weather events^{21, 22}. Here, the degrees of power shortage in affected areas during extreme weather periods are used to indicate the power system resilience, such as power shortage hours, the highest power shortage rate, and national entire power shortage.....”

In the revised version, we have added the following equations to calculate the cumulative unmet electricity demand by resource potential in the **The share of cumulative unmet electricity demand via hourly non-fossil resource power generation potential.** section in **Method** as follows.

“**The share of cumulative unmet electricity demand via hourly non-fossil resource power generation potential.** This indicator represents the cumulative unmet ideal electricity demand by province relative to provincial overall non-fossil fuel energy resource potential divided by the national overall ideal electricity demand, as shown in Equation (25).

$$UEDRP^N = \frac{\sum_n \left(\max \left(\sum_t IED_{n,t} - \sum_t (WP_{n,t} + PV_{n,t} + HP_{n,t} + NC_{n,t}), 0 \right) \right)}{\sum_n \sum_t IED_{n,t}} \quad (6)$$

where, $UEDRP^N$ represents the share of cumulative unmet electricity demand via hourly non-fossil resource power generation potential, $IED_{n,t}$ is the ideal electricity demand in 2050 at hour t in province n , $WP_{n,t}$ is the wind power output potential at hour t in province n , $PV_{n,t}$ is the solar PV

power output potential at hour t in province n , $HP_{n,t}$ is the hydropower output potential at hour t in province n , $NC_{n,t}$ is the nuclear power output potential at hour t in province n .”

In the revised version, we have added the following equations to calculate the power shortage in the **Power shortage rate definition** section in **Method** as follows.

“**Power shortage rate definition.** This study defined four categories of power shortage rates including national entire power shortage rate, provincial entire power shortage rate, national hourly power shortage rate, and provincial hourly power shortage rate as follows.

First, the national entire power shortage rate represents the cumulative gap between the provincial hourly electricity supply not meeting its ideal hourly electricity demand divided by the national overall ideal electricity demand, as shown in Equation **Error! Reference source not found.**

$$EPS^N = \frac{\sum_n \sum_t (IED_{n,t} - ES_{n,t})}{\sum_n \sum_t IED_{n,t}} \quad (7)$$

where, EPS^N represents the national entire power shortage rate. $ES_{n,t}$ is the electricity supply at hour t in province n , including local real-time hourly electricity supply via power generation, real-time hourly dispatch electricity supply via power generation, local energy storage discharging electricity supply, and hourly dispatch via energy storage discharging electricity supply as shown in Equation **Error! Reference source not found.**, $IED_{n,t}$ is the ideal electricity demand in 2050 at hour t in province n , as shown in Equation (24).

Second, the provincial entire power shortage rate represents the cumulative gap for a specific province between the hourly electricity supply not meeting its ideal hourly electricity demand divided by the provincial overall ideal electricity demand, as shown in Equation **Error! Reference source not found.**

$$EPS_n^P = \frac{\sum_t (IED_{n,t} - ES_{n,t})}{\sum_t IED_{n,t}} \quad (8)$$

where, EPS_n^P represents the entire power shortage rate in province n .

Third, the national hourly power shortage rate represents the gap between the national hourly electricity supply not meeting its ideal hourly electricity demand divided by the national ideal hourly electricity demand, as shown in Equation **Error! Reference source not found.**

$$HPS_t^N = \frac{\sum_n (IED_{n,t} - ES_{n,t})}{\sum_n IED_{n,t}} \quad (9)$$

where, HPS_t^N represents the national hourly power shortage rate at hour t .

Finally, the provincial hourly power shortage rate represents the gap for a specific province between the hourly electricity supply not meeting its ideal hourly electricity demand divided by the provincial ideal hourly electricity demand, as shown in Equation **Error! Reference source not found.**

$$HPS_{n,t}^P = \frac{IED_{n,t} - ES_{n,t}}{IED_{n,t}} \quad (10)$$

where, $HPS_{n,t}^P$ represents the hourly power shortage rate at hour t in province n .

13. Pg. 8 Fig. 1 - It may be a good idea to create an aggregated figure for the entire China region where RE resources are also aggregated. This would provide a sense of how the system would potentially perform over the whole country.

Authors' Response: Thank you for your helpful suggestions. In this revised version, we have incorporated the national daily and hourly variations of wind power and solar PV resource potential and projected electricity demand in 2050 as a subfigure in Fig. 1. Meanwhile, the calculation of all indicators (e.g., power generation, electricity demand) in Fig. 1 has been presented in detail in methods.

In the revised version, we have added Fig. 1i in Fig.1 and relevant descriptions in the ***Unmet electricity demand in a zero fossil fuel power system in 2050 China*** section in ***Results*** as follows.

“By 2050, China’s non-fossil energy (onshore wind, offshore wind, solar PV, hydropower and nuclear) power generation potential (equal to the sum of corresponding hourly maximum power output potential) reaches 90,076 billion kWh, of which variable renewables (solar and wind in this study) account for 96%, or 6.2 times the total projected electricity demand (shown by Equation (16)) for that year (Figure 1i, Supplementary Figure S3a).”

Fig. 1 Daily and hourly variability of wind power and solar PV generation potential and predicted electricity demand in 2050. (a-i represent the North Coast, Northeast, East Coast, Beijing-Tianjin, South Coast, Northwest, Central, Southwest, and the whole mainland China (Nation), respectively. The cyan and orange curves in each panel represent wind power and solar PV generation potential, and the green

curves in each panel represent predicted electricity demand for each region in 2050, respectively. The left-to-right column for each region depicts daily variability, hourly variability in summertime (June, July, and August) and wintertime (December, January, and February) for power generation potential and predicted electricity demand, respectively. The lines represent the mean values, the dark shading represents the inner 50% of the observations (25th to 75th percentile) and the light shading represents the outer 50% of the observations (0th to 100th percentile) of the daily average value of that date in each year of the relevant observations, i.e., 1980-2019 for wind power and 2010-2019 for solar PV.”

14. Pg. 9 Ln 210-213. The storage options are discussed for the first time. Batteries (why not Li-Ion that represent 90% of China’s actual currently installed grid battery storage?), compressed air and pumped hydro are claimed to be lower TRL which is not supported. Yes, the battery tech discussed is but there is battery tech that is commercial today with successful project. Also, it seems that hydrogen (a key option for weekly to seasonal storage) is not considered at all which needs to be explained. Given the benefits of adding even 12 hours of storage, this addition would have closed the remaining gap.

Authors’ Response: Thank you for your valuable comments. Sorry for the inappropriate descriptions about the storage options. We agree that several energy storage approaches previously described are relatively mature already (Supplementary Information Table S2). For example, pumped hydroelectric storage is considered as the most mature energy storage technology in China, while compressed air energy storage and Li-Ion batteries technology are also relatively mature compared to other energy storage technologies³⁴. We have revised the corresponding sentence and included the hydrogen storage.

Moreover, following your suggestions, we have incorporated hydrogen energy as a long-term energy storage alternative, as responded to the Comment 3 and 4. And results show that considering long-term energy storage in the power system will effectively reduce power shortages. For instance, when the energy storage hours are 6 hours and the dispatch capacity is 4 times, zero fossil fuel power generation with long-term energy storage scenario could achieve 0.03% power shortage, while the zero fossil fuel scenario without long-term energy storage would cause 1.5% power shortage rate.

In this revised version, we have modified relevant description about the energy storage in the **Unmet electricity demand in a zero fossil fuel power system in 2050 China** section in **Results** as follows.

“.....On the other hand, short-term or long-term electricity storage (e.g., using low-cost flow batteries, Li-ion batteries, compressed air energy storage, pumped hydroelectric storage, and hydrogen energy storage^{10, 19}), particularly in renewable-rich areas, could stabilize intermittent local wind power and solar PV, despite most of them being at a lower technology readiness level⁵⁶, relatively higher cost (Supplementary Table S2), geographically limited, or having less installed capacity advancement than other low-carbon technologies such as nuclear and energy efficiency.....

Table S2. Current maturity and cost of various types of energy storage technologies in China^{1, 2, 3, 4, 5}

Types of energy storage	Technical classification	Technology maturity	Cost
--------------------------	---------------------	------

Thermal energy storage	Cold and energy storage	H	H
	Thermal energy storage	H	M
	Pumped hydroelectric storage	EH	L
Mechanical energy storage	Compressed air energy storage	H	M
	Flywheel energy storage	M	H
Electromagnetic energy storage	Ultracapacitor energy storage	M	H
	Superconductor energy storage	L	EH
Electrochemical energy storage	Li-ion battery	H	M
	Sodium-ion battery	M	H
	Lead-acid battery	H	M
Chemical energy storage	Liquid flow battery	M	H
	Hydrogen energy storage	M	EH

Note: The maturity and cost intensity of the technology includes four levels: Low (L), Moderate (M), High (H) and Extremely high (EH).”

In this revised version, we have added relevant cost descriptions in the **Cost calculation for power generation technologies** section in **Supplementary Information** as follows.

“**The cost of energy storage.** Based on the technical maturity, development prospect and experts survey of each energy storage technology, we adopt Li-ion battery energy storage to represent short-term energy storage technology and hydrogen energy storage to represent long-term energy storage in the calculation of power system cost. The LCOE of energy storage (short-term and long-term) is calculated considering technological progress, as shown in Equations (S7)-(S8).

$$LCOE^{es} = LCOE^{es'} \cdot (1 - CR^{es}) \quad (S11)$$

$$H2^s = H2^{s'} \cdot (1 - CR^{H2S}) \quad (S12)$$

where, $LCOE^{es}$ is the LCOE of short-term energy storage in 2050 (at 2020 constant price) (USD/kWh). $LCOE^{es'}$ is the LCOE of short-term energy storage in 2020 (USD/kWh). CR^{es} is the LCOE reduction rate of short-term energy storage in 2050. $H2^s$ is the cost of hydrogen storage in 2050 (at 2020 constant price) (USD/kg). $H2^{s'}$ is the cost of hydrogen storage in 2015 (at 2020 constant price) (USD/kg). CR^{H2S} is the decrease rate in hydrogen cost from 2015 to 2050, taken as 0.175. In addition, the cost of hydrogen production from renewable energy can be referred to Equation (45) in the Method

section, and as for hydrogen combustion power generation, we only consider the operation and maintenance cost, assuming that it can use the infrastructure of existing fossil fuel power plants.”

15. Pg. 11 Fig. 2 and Ln 245- 247. Given the lack of Hydrogen inclusion in the analysis as noted in the previous comment, the claim that it is essentially impractical to operate an 100% RE system in China is invalid. It is clear that some kind of longer-term buffer is needed. By excluding this option (hydrogen and operating the pumped hydro only for 12 hour storage), the authors are forcing the conclusion that CCUS (the only other alternative) is needed. Unless, the hydrogen option is included and discussed, the paper’s conclusions will not have any practical validity.

Authors’ Response: Thank you for your valuable comment. We agree with your point that an 100% renewable energy power system may obtain a satisfactory power reliability (a rather low power shortage) when enough energy storage and power dispatch are included. Therefore, we have revised the wording of this sentence by adding the prerequisite, i.e., “.....*Overall, a fully non-fossil power system will have difficulty achieving satisfactory power reliability in 2050 unless short-term energy storage and dispatch infrastructure is well developed.....*”. In addition, it should be noted that 0-24 storage hours are considered as the typical short-term duration in this study. As energy storage hours are less effective in reducing power shortages within 13-24 hours, we mainly report the results of energy storage hours within 12 hours, and we have explained this point in the note of Figure 2. For energy storage technologies, we also agree that the duration of pumped hydroelectric storage can exceed 12 and 24 hours, but too long storage time will lead to lower utilization efficiency and higher cost. Therefore, considering the technology characteristics, geographic constraints and development prospects of each type of energy storage, the short-term energy storage technology and long-term energy storage technology in the model of this study refer to the Li-ion battery energy storage and hydrogen energy storage, respectively.

As replied to Comment 3 and 4, we have introduced long-term hydrogen option in the power system simulation model and additionally ran a set of scenarios for zero fossil fuel power system with long-term hydrogen storage to compare the results with the abated fossil fuel scenario in this revision. The results show that implementing long-term hydrogen storage in a zero fossil fuel power system can also achieve a lower power shortage comparable to the abated fossil fuel scenario, which even has an advantage in achieving higher power system reliability (<0.03%). In terms of the costs, to achieve a 0.1% power shortage, the minimum costs of the zero fossil fuel power system with long-term hydrogen storage will be 19.3% (LCOE 3.72%) higher than the 16% abated fossil fuel power system, which are \$820 billion (LCOE 47.15 USD/MWh) and \$662 billion (LCOE 45.64 USD/MWh), respectively. However, the long-term hydrogen storage will be more cost effective than the abated fossil fuel if the power supply reliability standard is further improved. For example, to achieve a 0.01% power shortage, the LCOE of the zero fossil fuel power system with long-term hydrogen storage is 47.19 USD/MWh, which is lower than that of the 16% fossil fuel power system with CCS (48.21 USD/MWh). Therefore, the long-term hydrogen storage technology is a visible alternative for enhancing power system reliability, especially when the power shortage standard improves in the future.

In this way, the revised version has included more considerations about the effect of long-term hydrogen storage on the power system, which does not deny our conclusion that a certain ratio of

fossil fuel with CCS can facilitate the power system's 99.9% reliability at a lowest cost compared with zero fossil fuel power system.

In this revised version, we have modified this sentence in the “**Unmet electricity demand in a zero fossil fuel power system in 2050 China**” section in **Results** as follows.

“.....Overall, a fully non-fossil power system will have difficulty achieving satisfactory power reliability in 2050 unless short-term energy storage and dispatch infrastructure is well developed (as indicated by the limited area of colder color representing less than 0.1% national entire power shortage rate shown in Figure 2a) or long-term storage is included (see analysis afterward).”

In the revised version, we have added more descriptions about the zero fossil fuel power system with long-term hydrogen storage in the **Power system reliability under high renewables with abated fossil fuel** section in **Results** as follows.

“Considering that long-term energy storage technologies (e.g., hydrogen energy storage, thermal energy storage) may play an essential role in sustaining electricity supply reliability like the role of fossil fuel power with CCUS. We exclusively run a set of scenarios for a zero fossil fuel power system with long-term hydrogen storage for comparison. The results show that, a minimum combination of dispatch capacity and short-term energy storage is required to ensure a rather low power shortage rate under zero fossil power system with long-term hydrogen storage as with all other scenarios. For instance, meeting the 0.1% national entire power shortage rate with the least cost necessitates the construction of 6 hours of short-term energy storage and 4 times of reference dispatch capacity (Supplementary Figure S6a), corresponding to the systemic cost of \$820 billion and the levelized cost of 47.15 USD/MWh, which are respectively 19.3% and 3.2% higher than the scenario of high renewable with 16% abated fossil fuel power generation (Supplementary Figure S7). However, if the national electricity supply reliability standard was further enhanced (e.g., 0.03%), long-term energy storage would play a more important role than abated fossil fuel with CCUS, as shown by a larger area towards the right-upper corner representing the same power shortage level in Supplementary Figure S6a than S6b; or even have a lower levelized cost to meet a higher power shortage standard, e.g., with at least 47.19 USD/MWh and 48.21 USD/MWh for 0.01% national entire power shortage rate under zero fossil fuel with long-term energy storage and high renewable with 16% abated fossil fuel, respectively. Therefore, long-term energy storage technology is a wise option to improve the power system supply reliability when faced with more stringent national shortage rate standard in the future.”

In the revised version, we have added relevant cost descriptions in the **Cost calculation for power generation technologies** section in **Supplementary Information** as follows.

“The cost of energy storage. Based on the technical maturity, development prospect and experts survey of each energy storage technology, we adopt Li-ion battery energy storage to represent short-term energy storage technology and hydrogen energy storage to represent long-term energy storage in the calculation of power system cost.”

16. Pg. 11 Ln 53 and broadly. At this section, the CCUS option is discussed for the first time. From the later discussion, it appears that the utilization component is EOR - which extends oil production. This should be discussed upfront - since EOR is not really a long-term carbon storage mechanism

unless it is purposely operated for this (higher CO₂ ratios - cf. Santos, R., Sgouridis, S. & Alhajaj, A. Potential of CO₂-enhanced oil recovery coupled with carbon capture and storage in mitigating greenhouse gas emissions in the UAE. *Int J Greenh Gas Con* 111, 103485 (2021).). Either way only a portion of the carbon can be considered captured as it is offset by the additional oil production. This is a significant caveat that should be discussed.

Authors' Response: Thank you for your valuable comment. We totally agree with you that unless purposely operated for higher CO₂ storage ratios, EOR is not really a long-term carbon storage mechanism mainly due to its relative limited CO₂ storage rate, CO₂ emissions from EOR operation, and the additional usage of enhanced oil. Indeed, in this study, we have adopted the CCU option of EOR as a possible supplement to CO₂ sequestration in addition to the saline aquifer CO₂ storage. On the one hand, such consideration is based on the assumption that EOR in this study could be identified as a purposely operated CO₂ sequestration, like the EOR practices in the UAE³⁵. Given that there are many large oil fields in China, we assume advanced EOR measures can be taken to improve the CO₂ storage ratios, such as CO₂ contained and running in a closed loop in EOR activities. In this case, CO₂-EOR could present a relatively low carbon footprint with more than 95% of anthropogenic CO₂ used in EOR that can be permanently stored in oil reservoirs³⁵.

On the other hand, since the uncertainties in the oil use structure (e.g., as energy or non-energy) and various application prospects of oil-related CO₂ emissions reduction measures in the future, it is difficult to account for the indirect CO₂ emissions caused by the enhanced oil production in our study year (2050). Therefore, we assumed a limited research boundary for CO₂ emissions and mitigation accounting related to EOR.

Last but not least, as a CO₂ utilization option, EOR could provide substantial oil benefits so as to stimulate its application in oil enterprises, which is crucial to advance the CCUS deployment in China. Actually, CO₂-EOR projects have dominated China's current CCUS deployment over the past years, with 11 EOR-related projects (1.5 Mt CO₂/a for CO₂ capture capacity) in total 40 CCUS projects (3.0 Mt CO₂/a) by 2021³⁶. For example, China's first million-tonne CCUS project, the Qilu Petrochemical - Shengli Oilfield CCUS Project, demonstrated the significance of EOR for early CCUS retrofit opportunities in China. As a result, we believe that it is essential to incorporate EOR into the candidate CO₂ sequestration options in this study.

In this revised version, we further defined the EOR sequestration option in the ***CCUS source-sink matching model for fossil fuel power generation plants*** section in ***Method*** as follows.

"5. Geological reservoirs associated with deep saline aquifers and oil fields are considered as candidate storage sites; note that here the oil fields are assumed to be identified as a purposely operated CO₂ sequestration option with matching priority due to the benefits of enhanced oil recovery (EOR)."

In this revised version, we further defined the CO₂ emission boundary in the ***Optimal matching model between biomass resources and CCUS-qualified power plants*** section in ***Method*** as follows.

"CO₂ emissions accounting boundary. Here, we set the boundary to the direct CO₂ emissions reduction related to all types of low carbon technologies involved in the whole power system. Specifically, the indirect CO₂ emissions from wind power, solar PV, hydropower, and nuclear power are not considered, and only the remaining CO₂ emissions that cannot be captured by CCUS retrofitting coal-fired and gas-fired power are calculated (10% of the total emissions). As a result, we introduce the negative emission through coal and biomass co-firing system coupled with CCS to achieve complete net zero emissions for

China's power system in the same emissions accounting framework.”

In this revised version, we have added specific descriptions about EOR-related emissions in the **Limitations and further perspectives** section in **Method** as follows.

“.....Finally, due to large uncertainties in future emission factors of oil usage, indirect emissions from additional oil production via the EOR process are not considered within our study boundary.....”

17. Pg. 20 Ln 476. The methodological assumption for estimating wind potential by a standard 2MW turbine could be accepted. Nevertheless, given that current wind systems have much higher capacities (with off-shore exceeding 10MW), the financial implications of the choice should be discussed specifically (and how it is corrected).

Authors' Response: Thank you for your helpful advice. We also note that a new single standard wind turbine will have a higher installed capacity than that assumed in our study. For instance, the average capacity of newly-added single-unit for onshore and offshore wind power in China in 2021 is 3.1 and 5.6 MW, respectively, with only a few offshore wind power exceeding 10 MW recently started. And for China's historically existing wind power by the end of 2021, the average single-unit capacity for onshore and offshore wind power is 2.29 and 5.49 MW, with a single-unit capacity of 1.5-3 MW accounting for 79% of the total onshore capacity and a single-unit capacity of 4-6 MW accounting for 54.3% of the total offshore capacity³⁷. We also summarize previous studies on the future estimates of wind power potential employing onshore and offshore turbines, and find that their rated capacities are ranging from 1.5-3.4 and 5-8 MW^{21, 38, 39}, respectively.

Considering the above consideration and your suggestion, in this revised version, we have chosen the medium value of the rated capacity of onshore wind as 2 MW, and revised the rated capacity of offshore wind as 6.45 MW. After correcting the installed capacity of offshore wind power, we have recalculated the real-time generation potential of offshore wind power and re-run the power system model. Finally, all the relevant data and results have been updated in the main text and figures.

In the revised version, we have modified the related methodological description in the **Wind power potential assessment** section in **Method** as follows.

“Given that earlier studies investigated the wind power potential employing onshore and offshore turbines with rated capacities ranging from 1.5-3.4 and 5-8 MW^{47, 65, 66}, as well as the newly-added single-unit installed capacity, we have set the rated capacity of onshore wind as 2 MW turbines at 90m hub height, and the rated capacity of offshore wind as 6.45 MW turbines at 108m hub height to represent the average level of wind power in China.....”

18. Pg. 21 Ln 495 -526. The solar estimation approach seems to have two main gaps:

- * It does not consider bi-facial systems (current state of the art - adds min 5-10% yield)
- * It does not consider single-axis tracking (current state of the art for utility scale projects that provide significant benefits by adding min 10-15%yield available in morning/afternoon periods)

Authors' Response: Thank you for carefully reminding. We acknowledge that the two aspects you mentioned are very important for solar PV power potential estimation; however, based on the

following two considerations, we have investigated photovoltaic panels using a single fixed-tilt mechanism in this study. First, although a bi-facial system may increase annual energy generation by 2.8%-11.9%⁴⁰, mono-facial modules are more affordable because they are easier to install, lighter than bi-facial panels, and can operate on all surfaces without requiring a reflective surface⁴¹. Second, although tracking systems can increase annual energy generation by up to 27% for single-axis trackers and 45% for dual-axis trackers, they require additional capital costs for procurement and installation, more land area to avoid shading, and higher maintenance costs due to moving parts and actuation systems⁴². Meanwhile, we also note that the single fixed-tilt systems assumption has been widely used in assessing solar PV potential^{30, 43, 44}. For example, Chen, Lu⁴³ selected an optimal tilt, altitude angle, azimuth angle, and spacing, which could help to maximize incoming solar radiation and optimize the sizing of a solar PV farm.

In the revised version, we have modified the related methodological description in the *Solar PV potential assessment* section in *Method* as follows.

“.....Here we chose photovoltaic panels with a single fixed-tilt system rather than the more advanced bi-facial or single-axis tracking systems. Although bi-facial systems and single-axis tracking systems theoretically can track the sun completely for higher solar energy output⁶⁸, single fixed-tilt systems are more widely used, simpler, cheaper, and have lower maintenance requirements⁶⁹. In this system, adopting an optimal tilt and azimuth angle can help the photovoltaic array receive more solar radiation and increase the overall power generation of the solar PV system⁷⁰.”

19. Also, the cost assumptions of PV seem higher than expected. The table 6 supplementary is not clear but it seems that a price of 5c/kWh are being used for PV, with only a 37% reduction by 2050. This would make 2050 PV costs higher than current costs for high-insolation climates (<3c/kWh) Please revise/explain accordingly.

Authors' Response: Thank you for your helpful comment. Indeed, we assumed the levelized cost of solar PV in 2020 is 0.051USD/kWh (5c/kWh) and a reduction of 37% by 2050 in this study. By this assumption, the LCOE of in 2050 is calculated as 0.032USD/kWh, which is likely to be higher than that for high-insolation climates (<0.03USD/kWh) as you mentioned. However, it should be noted that the high-insolation climate is not appropriate to represent the basic weather condition of a country, as it only occurs at a special period in some specific areas. And the cost reduction rate of 37% for solar PV by 2050 is a reference to the medium scenario cost evaluated by Chen, Liu³, which we expect to be representative of an average cost in a regular climate.

Following your suggestion, we further survey on previous literature and found that previous studies showed a wide range of predictions for future solar PV costs. Specifically, some provided optimistic estimations of future solar PV costs. For example, NREL⁴⁵ predicted a 46%-67% drop in the LCOE of solar PV by 2050 compared to 2020; Graham, Hayward⁴⁶ predicted a 55%-60% drop in the solar PV investment cost by 2050 compared to 2020. Meanwhile, some indicated that the drop in solar PV costs may not be as significant as the above estimations. For example, IRENA⁴⁷ suggested that the greatest LCOE of solar PV in 2050 will remain at 0.05 USD/kWh (at 2018 constant price). Therefore, following Chen, Liu³, we expected that the LCOE of solar PV in 2050

would be 37% lower than in 2020, which could be considered a comparable average level of previous research results.

In addition, to make the results more robust, we performed a sensitivity analysis of the variation in solar PV cost in 2050 in revised version. The results show that the total cost of a zero fossil fuel power system would decrease by 4.8% (increase by 3.8%) with the higher (lower) cost reduction rate of solar PV (i.e., 48% or 28%) evaluated by Chen, Liu³ under the high (low) scenario. When the cost of the near-zero power system reaches the minimum, the share of abated fossil fuel would change from 16% to 12% (20%).

In the revised version, we have modified relevant cost descriptions of solar PV in the **Table S5** in **Supplementary Information** as follows.

“Note: For the cost reduction of onshore wind power, offshore wind power, and solar PV, we used the medium scenario of all cost reduction scenarios in the corresponding references^{27,28}.”

In the revised version, we have added a section of **Impact of renewable energy costs on the power system** in **Supplementary Discussions** as follows.

“Impact of renewable energy costs on the power system

Impact of solar PV costs on the power system. *The sensitivity analysis of the change in solar PV cost in 2050 showed that even with a lower (higher) cost of solar PV (i.e., 48% or 28%) estimated by Chen, Liu²⁸, the total cost of a zero fossil power system would decrease by 4.8% (increase by 3.8%).*

Impact of wind power costs on the power system. *The sensitivity analysis of the change in onshore and offshore wind power cost in 2050 showed that even with a lower (higher) cost of onshore and offshore wind power (i.e., 54% and 64% or 8% and 26%) estimated by Wiser, Rand²⁷, the total cost of a zero fossil power system would decrease by 8.2% (increase by 11.9%).*

Impact of variable renewable energy and hydrogen energy storage costs on the power system. *Furthermore, we further analyzed the comparison of zero fossil fuel with long-term energy storage scenarios and 16% abated fossil fuel scenario costs and found that the total power system costs for the zero fossil fuel with long-term energy storage scenario would be comparable when variable renewable energy and long-term energy storage (hydrogen energy storage) costs are reduced by 20% and 25%, respectively (Supplementary Figure S6c).”*

20. Pg. 22 Ln 586 - the role of the reference province (Anhui) is unclear. Why is it needed if all provinces have reference hourly values?

Authors’ Response: Thank you for your careful reminding. We are sorry for not interpreting “the role of reference province (Anhui)” accurately in the original version. We totally agree that obtaining sufficient statistics from all provinces would be more informative and valuable. However, due to data availability, we only acquire the real-time hourly electricity consumption in Anhui province for the whole year (8760 hours) in 2019. Based on this accessible information, we have produced relatively accurate estimates for other provinces. Specifically, the relevant calculation method has been demonstrated in response to Comment 12, and in summary, hourly electricity demand in 2050 for other provinces is estimated based on the variation of hourly electricity consumption in reference

province (Anhui) in 2019, the monthly electricity consumption in each province in 2019, and the typical hourly electricity load on workdays and non-workdays in each province in 2019, as well as the amplification factor derived from econometric prediction.

In addition, owing to the difficulty in collecting real-time hourly electricity consumption data in China, some studies have adopted a simplified method that relies solely on total electricity consumption, as well as typical hourly electricity load on workdays and non-workdays to obtain provincial hourly electricity demand^{4, 48, 49}. In contrast to earlier research, our data acquisition process for each province, which is derived from the real-time hourly electricity consumption in a reference province, should be considered more sophisticated and accurate.

In the revised version, we have added relevant description of choosing Anhui as the reference province in the *Provincial real-time hourly electricity demand predictions* section in *Method* as follows.

“Provincial real-time hourly electricity demand predictions. Compared with previous literature on electricity demand predictions for Chinese power system models^{43, 56, 77, 78}, we use a more accurate and refined method to predict the hourly electricity demand of each province in China in 2050 as follows.

First, according to the monthly electricity consumption and the typical hourly electricity load on workdays and non-workdays in each province in 2019⁷⁹, the corresponding hourly electricity load on workdays and non-workdays for each month are calculated by Equations (20)-(21). Second, considering the number of workdays and non-workdays in each month, the average hourly baseline electricity load at same hour for each month is calculated by Equation (22). Third, due to data availability, Anhui province was selected as the reference province, and its hourly actual electricity consumption in 2019 was used to calculate the proportion of electricity load for each hour of each day of each month, compared to the electricity load for the same hour of the same month, as shown in Equation (23). Then the hourly electricity demand for each province for the whole year is calculated referring to the hourly electricity demand variation proportion of Anhui province by Equation (24). Finally, the real-time hourly electricity demand for each province in 2050 is predicted using the multipliers of the annual provincial electricity demand relative to 2019. The specific equations are as follows.

The hourly electricity load of workdays and non-workdays for each month of each province is shown in Equations (20)-(21).

$$EC_{n,m,t}^w = TEL_{n,t}^w \cdot \frac{EC_{n,m}}{\sum_t (TEL_{n,t}^w \cdot D_m^w + TEL_{n,t}^{nw} \cdot D_m^{nw})} \quad (13)$$

$$EC_{n,m,t}^{nw} = TEL_{n,t}^{nw} \cdot \frac{EC_{n,m}}{\sum_t (TEL_{n,t}^w \cdot D_m^w + TEL_{n,t}^{nw} \cdot D_m^{nw})} \quad (14)$$

where, $EC_{n,m,t}^w$ is the electricity consumption at hour t of workdays for month m of province n (MWh), $TEL_{n,t}^w$ is the typical electricity load at hour t of workdays of province n (MW), $EC_{n,m}$ is the electricity consumption for month m of province n in 2019 (MWh), D_m^w is the number of workdays in month m , D_m^{nw} is the number of non-workdays in month m , $EC_{n,m,t}^{nw}$ is the electricity load at hour t of non-workdays for month m of province n (MWh), $TEL_{n,t}^{nw}$ is the typical electricity load at hour t of non-workdays of province n (MW).

The average hourly baseline electricity load for each province is shown in Equation (22).

$$AEC_{n,m,t} = \frac{EC_{n,m,t}^w \cdot D_m^w + EC_{n,m,t}^{mw} \cdot D_m^{mw}}{D_m^{all}} \quad (15)$$

where, $AEL_{n,m,t}$ represents the average electricity consumption at hour t in month m of province n (MWh).

The variation ratio of actual hourly electricity consumption to average electricity consumption in Anhui province is shown in Equation (23).

$$PP_{m,d,t} = EC_{m,d,t} / \left(\frac{\sum_d EC_{m,d,t}}{D_m^{all}} \right) \quad (16)$$

Where, $PP_{m,d,t}$ is the Anhui province's variation proportion of actual electricity consumption in 2019 to the average electricity demand at hour t on day d in month m , $EC_{m,d,t}$ represents the actual electricity consumption in 2019 of the Anhui province at hour t on day d in month m (MWh).

The hourly electricity demand for the other province is shown in Equation (24).

$$ED_{n,m,d,t} = AEC_{n,m,t} \cdot PP_{m,d,t} \quad (17)$$

where, $ED_{n,m,d,t}$ represents the electricity demand of the province n at hour t on day d in month m (MWh).

Finally, the real-time hourly electricity demand in 2050 for each province can be estimated using the multiplier derived from provincial electricity demand in 2050 relative to 2019 estimated by econometric models.”

21. Pg. 25 Ln 642 - the dispatch rules of the simulation model seem reasonable with the notable exception of not including dispatchable hydrogen as noted earlier! Still, a more detailed explanation of the model should be provided (key equations and mathematical approach).

Authors' Response: Thank you for your helpful suggestion. As shown in the response to Comments 3, 4 and 5, we have introduced long-term hydrogen storage into the power system simulation model and ran an additional set of scenarios for comparison in this revised version. In the new model, both the social hydrogen production via water-electrolysis and hydrogen used as long-term storage via power generation are captured. Like the short-term electricity storage, the local electricity supply via hydrogen storage discharging and dispatch electricity supply via hydrogen storage discharging are arranged to fill electricity demand gap, respectively, with priority given to the local discharging supply. Due to the higher cost of hydrogen storage compared to short-term energy storage and problems with storage and transportation security of hydrogen, priority is given to short-term storage discharging when conducting power supply. And hydrogen storage facilities used for power generation are tended to be preferentially placed in provinces with large power shortages in order to make full use of hydrogen power generation.

In addition, to make the model clearer and more transparent, we have carefully revised the descriptions of the model. Some examples regarding the key equations and mathematical approach are as follows.

In the revised version, we have added objective functions and main **constraints** of the power system model in the *Optimal near-zero power system simulation model* section in *Method* as follows.

“Objective function of the optimal near-zero power system simulation model. The objective of the optimal near-zero power system simulation model is to ensure a minimum national shortage rate under different scenarios of abated fossil fuel shares, dispatch capacity and storage hours. The objective is shown in Equation (32).

$$\min NPS = \sum_n \sum_t (IED_{n,t} - ES_{n,t}) \quad (18)$$

where, NPS represents the national power shortage (MWh), and $ES_{n,t}$ including four sources of electricity supply, as shown in Equation (33).

$$ES_{n,t} = ES_{n,t}^l + ES_{n,t}^d + ES_{n,t}^s + ES_{n,t}^{sd} \quad (19)$$

where, $ES_{n,t}^l$ is the local real-time hourly electricity supply via local power generation at hour t in province n . $ES_{n,t}^d$ is the real-time hourly electricity supply from other provinces via power generation to province n at hour t through dispatch. $ES_{n,t}^s$ is the hourly electricity supply comes from energy storage discharging at hour t in province n . $ES_{n,t}^{sd}$ is the hourly electricity supply from other provinces to province n at hour t through dispatch via energy storage discharging.

Constraints of the optimal near-zero power system simulation model. The overall constraint of optimal near-zero power system is to ensure that the hourly electricity supply is less than the electricity demand, as shown in Equations (34).

$$ES_{n,t} \leq IED_{n,t} \quad (20)$$

where, $ES_{n,t}$ is total electricity supply at hour t in province n , including four sources of electricity supply. Based on the hourly electricity supply sources, the optimal near-zero power simulation model is divided into four modules: local real-time hourly electricity supply via power generation module, real-time hourly dispatch electricity supply via power generation module, local energy storage discharging electricity supply module, and hourly dispatch via energy storage discharging electricity supply module; note that energy storage in the above modules does not include the long-term energy storage. The main constraints of the four modules are as follows:

First, in the local real-time hourly electricity supply via power generation module, the local power generation is assumed to be prioritized to meet the local electricity demand, with main constraints including: local real-time hourly electricity supply via power generation in each province does not exceed its electricity demand, and it does not exceed the total power generation potential of local power generation technologies, as shown in Equations (35)-(36).

$$ES_{n,t}^l \leq IED_{n,t} \quad (21)$$

$$ES_{n,t}^l \leq \sum_z PGP_{z,n,t} \quad (22)$$

where, $PGP_{z,n,t}$ is the power generation potential of power generation technologies z at hour t in

province n , where $z=1$ is onshore wind power, $z=2$ is offshore wind power, $z=3$ is solar PV power, $z=4$ is nuclear power, $z=5$ is hydropower, $z=6$ is coal-fired power with CCUS, and $z=7$ is nature gas-fired power with CCUS.

Second, if the local real-time hourly electricity supply from power generation is insufficient to meet the local real-time hourly electricity demand, the real-time hourly dispatch electricity supply via power generation module will be required, with main constraints including: the dispatched electricity should not exceed the local unmet electricity demand by local power generation; the maximum electricity dispatch of each route is constrained by its designed dispatch capacity; and the maximum dispatched electricity does not exceed the upper limit of that available from the outflow province, as shown in Equations (37)-(39).

$$ES_{n,t}^d \leq IED_{n,t} - ES_{n,t}^l \quad (23)$$

$$ES_{n,t}^d \leq \sum_{n'} X_{n',n} \cdot DC_{n',n} \quad (24)$$

$$ES_{n,t}^d \leq \sum_{n'} X_{n',n} \cdot \left(\sum_z PGP_{z,n',t} - IED_{n',t} - EDS_{n',n,t}^d \right) \quad (25)$$

where, $X_{n',n}$ is a binary variable and takes 1 if there is a dispatch line from dispatched outflow province n' to dispatched inflow province n , otherwise it takes 0. $DC_{n',n}$ is the maximum dispatch capacity from dispatched outflow province n' to dispatched inflow province n . $PGP_{z,n',t}$ is the power generation potential of power generation technologies z at hour t in province n' . $IED_{n',t}$ is the ideal electricity demand in 2050 at hour t in province n' . $EDS_{n',n,t}^d$ is the accumulation of electricity dispatch via power generation from province n' to other provinces that was given priority than province n at hour t (if province n is the most electricity-deficient province and all other provinces are given priority to supply electricity to province n , then $EDS_{n',n,t}^d$ is taken as 0).

Third, if the local real-time hourly electricity supply from power generation and real-time hourly dispatch electricity supply from other provinces' power generation still cannot meet the local electricity demand, the local energy storage discharging electricity supply module will be required, with main constraints including: the local energy storage discharging electricity supply is less than the remaining electricity demand after local hourly power generation and dispatch electricity supply, and is less than the amount of energy storage charging minus the amount of electricity that has been discharged, as shown in Equations (40)-(41).

$$ES_{n,t}^s \leq IED_{n,t} - ES_{n,t}^l - ES_{n,t}^d \quad (26)$$

$$ES_{n,t}^s \leq \sum_{h=t-h_0-H+1}^{t-h_0} (RWP_{n,h} + RPV_{n,h}) - \sum_{h=t-h_0+1}^{t-1} (ES_{n,h}^s + EDS_{n,h}^{sd}) \quad (27)$$

where, h is the auxiliary variable related to t to simulate the process of energy storage charging and discharging. h_0 is the number of hours from the end of energy storage charging to hour t . H is the maximum number of energy storage hours, which varies from 1 to 24 hours according to different scenarios. $RWP_{n,h}$ is the remaining power generation potential of wind power after local real-time hourly electricity supply and real-time hourly dispatch electricity supply at hour h in province n . $RPV_{n,h}$ is the remaining power generation potential of solar PV after local real-time hourly electricity

supply and real-time hourly dispatch electricity supply at hour h in province n . $ES_{n,h}^s$ is the local energy storage discharging electricity supply at hour h in province n . $EDS_{n,h}^{sd}$ is the electricity dispatch through energy storage discharging from province n at hour h .

Fourth, if all the above electricity supply sources are unable to meet the local electricity demand, the dispatch via energy storage discharging electricity supply module will be required, with main constraints including: the dispatch via energy storage discharging electricity supply does not exceed the local remaining electricity demand, does not exceed the remaining dispatch capacity of each dispatch line, and does not exceed the amount of energy storage charging minus the amount of electricity that has been discharge in electricity outflow provinces, as shown in Equations (42)-(44).

$$ES_{n,t}^{sd} \leq IED_{n,t} - ES_{n,t}^l - ES_{n,t}^d - ES_{n,t}^s \quad (28)$$

$$ES_{n,t}^{sd} \leq \sum_{n'} X_{n',n} \cdot (DC_{n',n} - ES_{n',n,t}^d) \quad (29)$$

$$ES_{n,t}^{sd} \leq \sum_{n'} X_{n',n} \cdot \left(\sum_{h=t-h_0-H+1}^{t-h_0} (RWP_{n',h} + RPV_{n',h}) - \sum_{h=t-h_0+1}^{t-1} (ES_{n',h}^s + EDS_{n',h}^{sd}) - (ES_{n',t}^s + EDS_{n',n,t}^{sd,o}) \right) \quad (30)$$

where, $ES_{n',n,t}^d$ is the real-time hourly electricity supply dispatch via power generation from province n' to province n at hour t . $RWP_{n',h}$ is the remaining power generation potential of wind power after local real-time hourly electricity supply and real-time hourly dispatch electricity supply at hour h in province n' . $RPV_{n',h}$ is the remaining power generation potential of solar PV after local real-time hourly electricity supply and real-time hourly dispatch electricity supply at hour h in province n' . $ES_{n',h}^s$ is the local energy storage discharging electricity supply at hour h in province n' . $EDS_{n',h}^{sd}$ is the electricity dispatch via energy storage discharging from province n' at hour h . $ES_{n',t}^s$ is the local energy storage discharging electricity supply at hour t in province n' . $EDS_{n',n,t}^{sd,o}$ is the accumulation of electricity dispatch via energy storage discharging that has been dispatched from province n' in addition to province n at hour t . Specific calculations are shown in Supplementary Method."

In the revised version, we have explained model methodology and the calculation of the main parameters in the **Considerations for the optimal near-zero power system simulation model** section in **Supplementary Information** as follows.

".....aiming at a simultaneous assessment of the power system's reliability and resilience, this study adopts an iterative-based approach to construct an inter-provincial power system simulation model based on near real-time hourly meteorological data and calculate the hourly power shortages during normal year and the period of extreme climatic events (snowstorms, sandstorms, heat waves, and droughts), respectively."

"**Calculated amount of electricity supply from various sources.** Based on the constraints of the optimal near-zero power system simulation model, the local real-time hourly electricity supply via power generation, real-time hourly dispatch electricity supply via power generation, local energy storage discharging electricity supply, and electricity dispatch via energy storage discharging electricity supply can be calculated by the following Equations (S1)-(S4).

$$ES_{n,t}^l = \min \left(IED_{n,t}, \sum_z PGP_{z,n,t} \right) \quad (S31)$$

where, $ES_{n,t}^l$ is local the real-time hourly electricity supply via power generation at hour t in province n , as shown in Equation (33) in Method section. $IED_{n,t}$ is the ideal electricity demand in 2050 at hour t in province n , as shown in Equation (32) in Method section. $PGP_{z,n,t}$ is the power generation potential of power generation technologies z at hour t in province n , where $z = 1$ is onshore wind power; $z = 2$ is offshore wind power; $z = 3$ is solar PV power; $z = 4$ is nuclear power; $z = 5$ is hydropower; $z = 6$ is coal-fired power with CCUS, and $z = 7$ is nature gas-fired power with CCUS, as shown in Equation (35) in Method section.

$$ES_{n,t}^d = \min \left(\begin{array}{l} IED_{n,t} - ES_{n,t}^l, \sum_{n'} X_{n',n} \cdot DC_{n',n}, \\ \sum_{n'} X_{n',n} \cdot \left(\sum_z PGP_{z,n',t} - IED_{n',t} - EDS_{n',n,t}^d \right) \end{array} \right) \quad (S32)$$

where, $ES_{n,t}^d$ is the real-time hourly electricity supply dispatch via power generation from other provinces at hour t in province n , as shown in Equation (33) in Method section. $X_{n',n}$ is a binary variable and takes 1 if there is a dispatch line from dispatched outflow province n' to dispatched inflow province n , otherwise it takes 0, as shown in Equation (36) in Method section. $DC_{n',n}$ is the maximum dispatch capacity from dispatched outflow province n' to dispatched inflow province n , as shown in Equation (36) in Method section. $PGP_{z,n',t}$ is the power generation potential of power generation technologies z at hour t in province n' , as shown in Equation (36) in Method section. $IED_{n',t}$ is the ideal electricity demand in 2050 at hour t in province n' , as shown in Equation (36) in Method section. $EDS_{n',n,t}^d$ is the accumulation of electricity dispatch via power generation that has been dispatched away from province n' before the priority of province n at hour t (if province n is the most electricity-deficient province and all other provinces are given priority to supply electricity to province n , then $EDS_{n',n,t}^d$ is taken as 0), as shown in Equation (36) in Method section.

$$ES_{n,t}^s = \min \left(IED_{n,t} - ES_{n,t}^l - ES_{n,t}^d, \sum_{h=t-h_0-H+1}^{t-h_0} (RWP_{n,h} + RPV_{n,h}) - \sum_{h=t-h_0+1}^{t-1} (ES_{n,h}^s + EDS_{n,h}^{sd}) \right) \quad (S33)$$

where, $ES_{n,t}^s$ is the real-time hourly electricity supply comes from energy storage discharging at hour t in province n , as shown in Equation (33) in Method section. h is the auxiliary variable related to t to simulate the process of energy storage charging and discharging, as shown in Equation (37) in Method section. h_0 is the number of hours from the end of energy storage charging to hour t , as shown in Equation (37) in Method section. H is the maximum number of energy storage hours, which varies from 1 to 24 hours according to different scenarios, as shown in Equation (37) in Method section. $RWP_{n,h}$ is the remaining power generation potential of wind power after local real-time hourly electricity supply and real-time hourly dispatch electricity supply at hour h in province n , as shown in Equation (37) in Method section. $RPV_{n,h}$ is the remaining power generation potential of solar PV after local real-time hourly electricity supply and real-time hourly dispatch electricity supply at hour h in province n , as shown in Equation (37) in Method section. $ES_{n,h}^s$ is the local energy storage discharging electricity supply at hour h in province n , as shown in Equation (37) in Method section. $EDS_{n,h}^{sd}$ is the electricity dispatch through energy storage discharging from province n at hour h , as shown in Equation (37) in Method section.

$$ES_{n,t}^{sd} = \min \left(\begin{array}{l} IED_{n,t} - ES_{n,t}^l - ES_{n,t}^d - ES_{n,t}^s, \sum_n X_{n',n} \cdot (DC_{n',n} - ES_{n',n,t}^d), \\ \sum_{n'} X_{n',n} \cdot \left(\begin{array}{l} \sum_{h=t-h_0-H+1}^{t-h_0} (RWP_{n',h} + RPV_{n',h}) - \\ \sum_{h=t-h_0+1}^{t-1} (ES_{n',h}^s + EDS_{n',h}^{sd}) - (ES_{n',t}^s + EDS_{n',n,t}^{sd,o}) \end{array} \right) \end{array} \right) \quad (S34)$$

where, $ES_{n,t}^{sd}$ is the hourly electricity supply from other provinces to province n at hour t through dispatch after energy storage discharging, as shown in Equation (33) in Method section. $ES_{n',n,t}^d$ is the real-time hourly electricity supply dispatch via power generation from province n' to province n at hour t , as shown in Equation (38) in Method section. $RWP_{n',h}$ is the remaining power generation potential of wind power after local real-time hourly electricity supply and real-time hourly dispatch electricity supply at hour h in province n' , as shown in Equation (38) in Method section. $RPV_{n',h}$ is the remaining power generation potential of solar PV after local real-time hourly electricity supply and real-time hourly dispatch electricity supply at hour h in province n' , as shown in Equation (38) in Method section. $ES_{n',h}^s$ is the local energy storage discharging electricity supply at hour h in province n' , as shown in Equation (38) in Method section. $EDS_{n',h}^{sd}$ is the electricity dispatch via energy storage discharging from province n' at hour h , as shown in Equation (38) in Method section. $ES_{n',t}^s$ is the local energy storage discharging electricity supply at hour t in province n' , as shown in Equation (38) in Method section. $EDS_{n',n,t}^{sd,o}$ is the accumulation of electricity dispatch via energy storage discharging that has been dispatched away from province n' in addition to province n at hour t , as shown in Equation (38) in Method section.

To Reviewer #2:

This paper develops an optimal solution to have a reliable and resilient power sector in China with near-zero carbon emissions. Here are my concerns.

1. Reliability and resilience are two important terms in this paper, they should be clearly defined and explained, especially at the beginning of this study.

Authors' Response: Thank you for your helpful suggestions. We agree with your points that the power system reliability and resilience are critical analytical metrics in this paper. In this revised revision, these metrics have been clearly defined with accurate expressions.

First, reliability is used to reflect the ability of all grid-connected generating units that could meet electricity demand in a normal year⁵⁰, which is quantified by one minus the power shortage rate. For example, in order to avoid large-scale power shortages, this study uses the power shortage rate of less than 0.1% as a reliability indicator in power system cost analysis by referring to the present power supply reliability of a typical Chinese city (99.9%)⁵¹. Second, resilience is used to reflect the ability of power system to withstand power shortages and restore power supply in a timely manner during extreme weather events^{10, 52}, using the power shortage rate difference before and after the disasters as a resilience indicator.

In the revised version, we have added the explanation of reliability and resilience when they first appear in **Introduction** as follows.

“.....otherwise suffering from low reliability (defined as the degree to meet the ideal electricity demand under normal circumstances, with 99.9% as the current standard for Chinese cities¹⁸)^{5, 11, 14, 19}.”

“.....threatening the resilience (defined as the degree to meet the ideal electricity demand under weather events^{21, 22}, see Method).”

In the revised version, we also have added the definitions of resilience and reliability in **the reliability and resilience in power system** section in **Method** as follows.

“The reliability and resilience of the power system

Both the reliability and resilience of the power system could be measured by the power shortage. Referring to previous research¹¹ and national standards for electricity supply (e.g., 99.9% for cities in China)¹⁸, reliability is defined as the ability of all generating units connected to the grid to meet the electricity demand during normal years (i.e., without extreme weather events), quantified by one minus the power shortage rate. Resilience mainly measures the power system’s ability to withstand power shortages and restore electricity supply in a timely manner during extreme weather events^{21, 22}. Here, the degrees of power shortage in affected areas during extreme weather periods are used to indicate the power system resilience, such as power shortage hours, the highest power shortage rate, and national entire power shortage.....”

2. In abstract, the authors argue that the reliability and resilience of different electric power system architecture have rarely been assessed. However, there are lots of studies on the reliability and resilience of electric power systems. The authors should refine and reorganize innovations and contributions.

Authors’ Response: Thank you for your helpful suggestion. Sorry for the inappropriate expression in the abstract of the original version. We agree that there have been many assessments on power system reliability^{3, 4, 5} and resilience^{11, 53, 54}. For instance, regarding the reliability, Zhuo, Du⁴ evaluated the change in the system cost of electricity supply under the premise of meeting renewable energy development goals and ensuring power system reliability, and found that the electricity supply cost would increase by 9.6 CNY¢/kWh from 2020 to 2050 to achieve a carbon-neutral power system; regarding the resilience, Perera, Nik¹¹ considered the impact of various climate scenarios on the Swedish power system to test its resilience and concluded that extreme climate events would reduce Swedish power supply reliability by 16%.

However, most of these studies are from a single reliability or resilience perspective, or individual technology comparison perspective rather than within the same high-resolution power system model framework, and very few explore the optimal reliable and resilient power system through specific comparison between 100% renewable power and high share renewable with abated fossil fuel power generation with CCUS. In addition, following your suggestion, we have

rewritten the corresponding expression in Abstract and Introduction to refine our innovations and contributions in this study.

In the revised version, we have refined and reorganized relevant innovations and contributions in **Abstract** as follows.

“Decarbonized power systems are critical to mitigate global climate change, yet the method to achieve a reliable and resilient near-zero power system has rarely been investigated.”

In the revised version, we have refined and analyzed the relevant literature and contributions in **Introduction** as follows.

“However, few studies have compared the overall cost-effectiveness performance of a 100% renewable system versus a high share of renewables combined with abated fossil fuel power generation under the same system modelling framework, so as to obtain a reliable and resilient near-zero power system, except for some analysis from a single reliability^{12, 43} or resilience^{44, 45} perspective, or a comparative perspective on two individual technologies rather than within the same power system model framework⁴⁶.”

3. The reliability of the power system also needs to consider the real-time power balance among regions, including the following aspects, such as the analysis on steady state of the power system (when each node in the power system constructed in the article loses 10% of power generation or load, does the system have enough margin to automatically restore the steady state? The discussion about the oscillation period, in brief, the Lyapunov's observability and controllability proof need to be added in the model.

Authors' Response: Thank you for your careful reminder. In the revised version, according to the definition of Lyapunov stability, we have discussed more about real-time power balance among regions to examine the power system reliability. According to the electricity supply business rules issued by the Chinese National Energy Administration (CNEA), the frequency rating of China's power grid is around 50 Hz⁵⁵, and the permitted frequency variation should not exceed 1.0 Hz under abnormal conditions. Thus, we assume that the power system is at a steady state when the national real-time hourly power shortage rate is less than 2% (i.e., 1/50).

For the steady state of optimal power system in normal weather, how the power system can be restored to a steady state from various power shortage rates over 2% is shown in Supplementary Figure S13. In terms of observability of Lyapunov stability, 54 of 8760 hours own the national real-time hourly power shortage rate larger than 2%, with 2-3% for 36 hours, 3-4% for 11 hours, 4-5% for 6 hours, and more than 5% for 1 hour, and with 0, 32, 4, and 18 hours of national hourly power shortage rate greater than 2% in spring, summer, autumn, and winter, respectively (Supplementary Figure S13). In terms of controllability of Lyapunov stability, the fluctuation period for power shortage rate more than 2% is generally between 6 pm and 7 am. The power system can be restored to a steady state in a maximum of 7 hours when the fluctuation period is from 6 pm to 11 pm, whereas it can be restored in a maximum of 5 hours when the fluctuation period is from 0 am to 6 am (Supplementary Figure S13). This indicates that under normal weather conditions, the power output variation has little and short impacts on the electricity supply, which enables the optimal

power system easily to return to a stable state.

For the steady state of optimal power system under weather events, the power shortage rates during extreme weather events (including snowstorms, sandstorms, droughts, and heat waves) are shown in Figure 4 and Supplementary Figure S9; and the results indicate that extreme weather events have a large impact on the power system stability, mostly in the form of higher power shortage rate than the threshold value of 2%. In terms of observability of Lyapunov stability, there are 466, 17, 36, and 0 hours (101, 3, 5, and 0 hours) of power shortage rate higher than 2% (10%) in the 16% abated fossil fuel scenario during events of snowstorms, sandstorms, droughts, and heat waves, respectively, while 514, 29, 73 and 3 hours (396, 8, 18 and 0 hours) of power shortage rate higher than 2% (10%) in the zero fossil fuel scenario. In terms of controllability of Lyapunov stability, the power shortage rate can return to less than 2% (10%) within 440, 14, 8 and 0 hours (15, 3, 3, and 0 hours) in the 16% abated fossil fuel scenario during events of snowstorms, sandstorms, droughts, and heat waves, respectively, and within 503, 20, 10, and 2 hours (90, 5, 5, and 0 hours) in the zero fossil fuel scenario (Figure 4 and Supplementary Figure S9). Nevertheless, following various recovery measures taken by the government in reality after the weather events, we assume that the variable renewable power and grid infrastructure will gradually return to normal or be repaired, indicating the power system would have an automatic recovery from weather events.

Overall, we assume Lyapunov stability in our model, i.e., for normal and extreme weather, the government has sufficient capacity to make variable renewable power and grid infrastructure gradually restore steady state to ensure electricity supply security. Also, the results show that under normal or extreme weather, the power shortage rate can return to steady state in a limited time even if it is subject to fluctuations of more than 2%, indicating that the power system in this study can meet the Lyapunov's observability and controllability requirements.

In the revised version, we first have added relevant descriptions about the Lyapunov's observability and controllability in the ***Reliability and resilience of the power system*** and ***Limitations and further perspectives*** sections in ***Method*** as follows.

“The reliability and resilience of the power system.

..... Considering that zero-fossil fuel power generation and destroyed electricity dispatch infrastructure can be recovered or rebuilt artificially during or after a climatic disaster (e.g., snowstorms), we assume the affected power system could gradually restore to normal electricity supply, indicating that the power system has enough margin to automatically restore steady state. The detailed demonstration of Lyapunov's observability and controllability for the model is shown in Supplementary Information.”

“Limitations and further perspectives.

..... Seventh, this study did not specify the constraints on the steady state recovery and oscillation period in the power system model, instead analyzing the Lyapunov's rule from the power shortage perspective.”

Second, in the revised version, we have added relevant descriptions about the Lyapunov's observability and controllability in the ***Steady state consideration of the power system*** section in ***Supplementary Discussions*** as follows.

“Steady state consideration of the power system

According to the electricity supply business rules issued by the Chinese National Energy Administration, the frequency rating of China’s power grid is around 50 Hz ⁶⁴, and the permitted frequency variation should not exceed 1.0 Hz under abnormal conditions. Thus, we assume that the power system is steady state when the national real-time hourly power shortage rate is less than 2% (i.e., 1/50).

Steady state of optimal power system under normal weather. How the power system can be restored to a steady state under various power shortage rate over 2% is shown in Supplementary Figure S13. In terms of observability of Lyapunov stability, 54 of 8760 hours own the national real-time hourly power shortage rate larger than 2%, with 2-3% for 36 hours, 3-4% for 11 hours, 4-5% for 6 hours, and more than 5% for 1 hour, and with 0, 32, 4, and 18 hours of national hourly power shortage rate greater than 2% in spring, summer, autumn, and winter, respectively (Supplementary Figure S13). In terms of controllability of Lyapunov stability, the fluctuation period for power shortage rate more than 2% is generally between 6 pm and 7 am. The power system can be restored to a steady state in a maximum of 7 hours when the fluctuation period is from 6 pm to 11 pm, whereas it can be restored in a maximum of 5 hours when the fluctuation period is from 0 am to 6 am (Supplementary Figure S13). This indicates that under normal weather conditions, the power output variation has little and short impacts on the electricity supply, which enables the optimal power system easily to return to a stable state.

Steady state of optimal power system under weather events. The power shortage rates during extreme weather events (including snowstorms, sandstorms, droughts, and heat waves) are shown in Figure 4 and Supplementary Figure S9; and the results indicate that extreme weather events have a large impact on the power system stability, mostly in the form of higher power shortage rate than the threshold value of 2%. In terms of observability of Lyapunov stability, there are 466, 17, 36, and 0 hours (101, 3, 5, and 0 hours) of power shortage rate higher than 2% (10%) in the 16% abated fossil fuel scenario during events of snowstorms, sandstorms, droughts, and heat waves, respectively, while 514, 29, 73 and 3 hours (396, 8,18 and 0 hours) of power shortage rate higher than 2% (10%) in the zero fossil fuel scenario. In terms of controllability of Lyapunov stability, the power shortage rate can return to less than 2% (10%) within 440, 14, 8 and 0 hours (15, 3, 3, and 0 hours) in the 16% abated fossil fuel scenario during events of snowstorms, sandstorms, droughts, and heat waves, respectively, and within 503, 20, 10, and 2 hours (90, 5, 5, and 0 hours) in the zero fossil fuel scenario (Figure 4 and Supplementary Figure 9). Nevertheless, following various recovery measures taken by the government in reality after the weather events, we assume variable renewable power and grid infrastructure will gradually return to normal or be repaired, indicating the power system would have an automatic recovery from weather events.

Overall, we assume Lyapunov stability in our model, i.e., for normal and extreme weather, the government has sufficient capacity to make variable renewable power and grid infrastructure gradually restore steady state to ensure electricity supply security. Also, the results show that under normal or extreme weather, the power shortage rate can return to steady state in a limited time even if it is subject to fluctuations of more than 2%, indicating that the power system in this study can meet the Lyapunov’s observability and controllability requirements.

Figure S13. Hourly change in the national four-season power shortage rate under the 16% abated fossil fuel scenario. (a, b, c, and d represent hourly change in power shortage rate in spring (March, April, May), summer (June, July, August), autumn (September, October, November) and winter (December, January, February), respectively. The dotted line represents the hours when the hourly power shortage rate exceeds 2%”

4. In the model and simulation of the impact of extreme weather events, the manuscript only simulates the loss of above 220kV lines within the province. However, the loss of inter-provincial lines was also serious during the 2008 snow disaster, which should also be discussed in the manuscript.

Authors' Response: Thank you for your helpful suggestion. We agree that inter-provincial line losses were substantial during China's 2008 snowstorm. Here, we are sorry for some ambiguous expression in the previous statement, and we would like to clarify that we aimed to simulate the inter-provincial line losses in our analysis, rather than those losses within the province. This is because the model boundary of this study is limited to inter-provincial level, which treats the province as a whole, assuming the power within the province is free to be dispatched. We have already emphasized this point in the power system model assumptions and acknowledged this limitation in Method section.

In the revised version, we have modified relevant descriptions in the *Modeling and simulation of the impact of extreme weather events on the power system* section in *Method* as follows.

“Combined with the actual loss of transmission lines over 220 kV in Jiangxi, Guizhou, and Hunan during the 2008 snowstorm ⁹⁴, up to 60.5% of inter-provincial lines related to the affected provinces were assumed to be damaged.”

In the revised version, we have added relevant premise assumptions in the ***Assumptions of the optimal near-zero power system simulation model*** section in ***Method*** as follows.

“1. Due to the difficulty of obtaining grid transmission lines within a province and simplifying the model to make the optimization issue manageable, the power system simulation is assumed to only consider inter-provincial electricity dispatch, i.e., electricity within each province can be dispatched flexibly so that each province is considered as a single node. Meanwhile, the same type of power generation units within a province is considered as a single unit.”

In the revised version, we have added relevant premise assumptions in the ***Limitations and further perspectives*** section in ***Method*** as follows.

“..... Fourth, to refine the model composition, we examine each province as a single node and assume that the electricity in the province can be freely dispatched.”

5. The scenario setting is currently inconsistent with the reality of China's power grid, because existing literature shows that since the snowstorm disaster in China in 2008, DC ice melting devices have been installed on transmission lines above 110kV, and it is supposed that ice and snow disasters will not cause large-scale disconnection accidents. Since 2022, heat wave and drought have become a more extreme climate problem that plagues China's power grid: the load on the power grid increases sharply, heat wave causes less wind, and drought causes less water, resulting in an imbalance between power supply and demand. In this paper, you'd better focus on the impact of heat wave and drought on grid resilience.

Authors' Response: Thank you for your helpful suggestion and comments. We agree with you that direct-current (DC) ice melting devices have been widely employed in China since the 2008 snowstorm, and melting ice with electricity is becoming a feasible technique to reduce transmission line damage caused by excessive ice cover. However, considering the application conditions of DC ice melting devices and the complex causes of transmission line damage, we still assume that snowstorms are likely to have a significant influence on transmission lines. The reasons are as follows.

First, the DC ice melting devices are mainly applied in 110–500 kV lines ⁵⁶, while the inter-provincial dispatching lines assumed in this study are mostly Ultra-High Voltage (UHV) and Extra-High Voltage (EHV) transmission lines, thus it is difficult for DC ice melting devices to melt ice for UHV and EHV transmission lines when large-scale storms hit the grid. Second, despite the widespread use of DC ice melting devices, there are still many cases of damaging transmission lines and affecting power systems by snowstorms in China and other countries or regions in recent years ^{57, 58, 59, 60}, e.g., in Texas in 2021, winter snowstorms led to more than 10 million people without power during peak hours, and power outages lasted for several days in some areas, thus causing a cumulative economic loss of \$130 billion ⁵⁸. Third, during severe snowstorms, transmission lines

are easy to be affected by fallen trees, collapsed towers, and other damaged transmission equipment⁶¹, at which time transmission line failures cannot be avoided even with the installation of DC ice melting devices. In addition, considering the actual disaster impact of China's 2008 snowstorm, and related studies indicate that the frequency and magnitude of extreme weather events are likely to increase in the future⁶², we assume that snowstorms will still have a significant impact on transmission lines.

Following your suggestions, in addition to snowstorms and sandstorms, we have also added droughts and heat waves to simulate their impacts on the future near-zero power systems. First, we have investigated the drought and heat wave events that have occurred in China since the twenty-first century, then we selected the most severe drought and heat wave disasters that occurred in 2022 as the subjects of our investigation. Second, we have analyzed the mechanisms of drought and heat wave impacts on the power system, For example, 1) droughts may cause a decrease in water level, and affect hydroelectric power generation, as well as reduce photovoltaic efficiency, and result in an increase in electricity demand because they are usually accompanied by high temperatures; and 2) heat waves may cause less wind, which affects wind power generation, and the high temperatures caused by heat waves would reduce PV efficiency and increase electricity demand. Finally, we have simulated the effects of drought and heat wave on the power system based on the impact mechanism analysis and real-time meteorological data at the time of historical disasters.

In the revised version, we have added the reasons about the snowstorm damage to the dispatch line in the ***Modeling and simulation of the impact of extreme weather events on the power system*** section in ***Method*** as follows.

“..... Third, the snowstorm will seriously damage the power transmission infrastructures and reduce their dispatch capacities. Although direct-current (DC) melting devices have been widely installed to power grid after 2008 snowstorms, the destruction of snowstorms on dispatch still cannot be ignored due to the limited application conditions of DC melting devices⁹¹, the complex and various causes for transmission line damage⁹², and the likely enhanced frequency and magnitude of extreme weather in the future⁹³.”

In the revised version, we have modified relevant descriptions about additional droughts and heat waves in the ***Comparison of resilience to extreme climatic events under two power system configurations*** section in ***Results*** as follows.

“In comparison, sandstorms and droughts are likely to have less significant impacts on the power system in terms of both occurrences of adversely affected hours over 10% power shortage rate (8 and 18 hours) and entire power shortages (0.9 and 2.1 TWh) for affected provinces (i.e., regional power shortages), but a high share of renewables combined with abated fossil fuel power system is more resilient than a zero fossil fuel power system (Figure 4c and 4e). For instance, the affected areas under sandstorms and droughts events would have experienced 5 and 13 more hours with hourly power shortage rate over 10% under zero fossil fuel power system than the high renewable with abated fossil fuel power system, and the abated fossil fuel with CCUS scenario is able to reduce the power shortages by 56% and 57% during the events (respectively from 0.9 to 0.4 TWh, and from 2.1 to 0.9 TWh) (Figure 4d, 4f and Supplementary Figure S8d,S8 f).

..... At provincial level, Xinjiang, Gansu, and Ningxia under sandstorm and Sichuan under drought

have less power shortages to a larger extent in the high renewables combined with abated fossil fuel power than the zero fossil fuel power system, with cumulative power shortage rates decreasing from 6.61%, 6.47%, 41.93%, and 5.4% to 0%, 4.57%, 19.0%, and 3.0%, respectively (Supplementary Figure S8c, S8e). Hot waves have the least impacts on electricity supply system either under zero fossil fuel power generation or abated fossil fuel power generation with CCUS scenario (Supplementary Figure S8g, S8h and S9).

..... Importantly, incorporating abated fossil fuel power generation could significantly alleviate extreme power shortage at certain moment during weather occurrences. For instance, the greatest hourly power shortage rates with zero fossil fuel power system under snowstorms (43.9%), sandstorms (28.8%) and droughts (38.3%) would be reduced by 33.2, 14.9 and 38.3 percentages under 16% abated fossil fuel power generation system (Figure 4a, 4c and 4e).

..... Unlike snowstorms and sandstorms, high proportion renewables with abated fossil fuel power generation under droughts generate slight power shortage and only have a power shortage rate over 1% during midnight 0-3 am and 18-24 pm, with the maximum value up to 3.6% during 21-24 pm (Figure 4f). These can be confirmed by a similar performance observed from the fossil free case (Supplementary Figure S9d).

Fig. 4 Comparison of power shortage rates over event time in the affected provinces under snowstorm, sandstorm, and drought. (a and b represent the real-time and aggregated 3-hour periods under the snowstorm events, respectively, c and d represent sandstorm events, e and f represent drought events; note that b, f and d denote power shortage under the optimal power system only. Event times and intensities and affected provinces are sourced from actual disasters in China, i.e., January 14 to February 4, 2008 for snowstorms, March 15 to March 17, 2021 for sandstorms, and August 12 to August 27, 2022

for droughts.)”

Figure S8. Electricity structure and the impact of disasters on the power shortage in major affected provinces under two scenarios. (a, c, e, and g represent the electricity consumption structure of specific provinces during snowstorm events, sandstorm events, drought events, and heat wave events, respectively; b, d, f, and h represent the power shortage at aggregated 3-hour intervals in zero fossil fuel during snowstorm events, sandstorm events, drought events, and heat wave events, respectively.)

Figure S9. Comparison of power shortage in affected provinces during heat wave events. (a and b represent the real-time hourly and aggregated 3-hour periods during heat wave events, respectively. The event occurrence time, duration, intensity, and affected provinces are sourced from actual heat wave disasters in China from June 1 to August 31, 2022.)”

In this revised version, we have modified relevant descriptions about additional droughts and heat waves in the **Modeling and simulation of the impact of extreme weather events on the power system** section in **Method** as follows.

“Drought. The 2022 drought impacts simulation incorporates six most severely affected provinces (including Sichuan, Chongqing, Hubei, Hunan, Jiangxi, and Anhui, as shown in Supplementary Figure S12b) associated with their climatic conditions (drought level and temperature). And droughts often have three typical effects on the near-zero power system (see Supplementary Methods for details).

First, droughts cause significant evaporation of water, resulting in insufficient reservoir storage and losses in hydropower generation. Second, high temperatures during droughts can increase societal demand for refrigeration, leading to higher electricity demand; in addition, droughts also cause excessive temperature of photovoltaic panels, thus reducing solar PV generation efficiency⁹⁶.

Heat wave. The 2022 heat wave impacts simulation incorporates 14 most severely affected provinces (including Hunan, Zhejiang, Chongqing, Jiangxi, Jiangsu, Anhui, Shanghai, Guangdong, Sichuan, Xinjiang, Henan, Hubei, Fujian, and Hainan, as shown in Supplementary Figure S12b) associated with their climatic conditions (hourly temperature and wind speed during the heat wave). And heat waves often have three typical effects on the near-zero power system (see Supplementary Methods for details).

First, high temperatures during heat waves can raise the electricity consumption of refrigeration equipment (such as air conditioners), thus increasing societal electricity demand. Second, high temperatures during heat waves result in a reduction in solar PV generation efficiency, due to excessive temperature of photovoltaic panels⁹⁶. Third, when heat waves occur, the air warms strongly and the

pressure gradient is small, thus weakening the wind speed and reducing wind power output.”

In the revised version, we have added the impact mechanism of additional droughts and heat waves on the power system in **Supplementary Information** as follows.

“Impact mechanism of droughts. Droughts often have three effects on the near-zero power system. First, droughts cause significant evaporation of water, resulting in insufficient reservoir storage and reductions in hydropower output. The specific process by which droughts affect hydropower can be expressed by Equation (S19).

$$HP_{n,t}^{dr'} = HP_{n,t}^{dr} \cdot EHP_{n,t}^{dr} \quad (S19)$$

where, $HP_{n,t}^{dr'}$ is the actual hydropower output at hour t in province n during the drought. $HP_{n,t}^{dr}$ is the power output potential at hour t in province n during the drought, and $EHP_{n,t}^{dr}$ is the change coefficient of output efficiency of hydropower generation at hour t in province n during the drought relative to normal weather, the values can be found in Supplementary Table S9.

Second, droughts are typically accompanied by hot weather; high temperatures during drought periods can raise societal demand for refrigeration (e.g., air conditioners), thus increasing the electricity demand. The specific process by which droughts affect electricity demand can be expressed by Equation (S20).

$$ED_{n,t}^{dr} = IED_{n,t} \cdot (1 + \Delta T_{n,t}^{dr} \cdot I^{ED}) \quad (S20)$$

where, $ED_{n,t}^{dr}$ is the hourly electricity demand at hour t in province n during the drought. $\Delta T_{n,t}^{dr}$ is the temperature increment at hour t in province n during the drought compared to the year selected in the study (2016), and I^{ED} is the percentage increase in electricity demand per 1°C higher temperature rise, taken as 2.3%⁵⁰.

Third, the high temperature of PV panels caused by hot weather reduces solar PV power output efficiency, as shown in Equation (5) in Method section. The solar PV and wind power output during drought periods is calculated by Equations (S21)-(S22).

$$PV_{n,t}^{dr'} = \min(PV_{n,t}^{dr}, PV_n^{IC}) \quad (S21)$$

$$WP_{n,t}^{dr'} = \min(WP_{n,t}^{dr}, WP_n^{IC}) \quad (S22)$$

where, $PV_{n,t}^{dr'}$ and $WP_{n,t}^{dr'}$ are the actual solar PV and wind power output at hour t in province n during the drought respectively. $PV_{n,t}^{dr}$ and $WP_{n,t}^{dr}$ are the power output potential at hour t in province n at the corresponding radiation intensity and wind speed during the drought respectively. The impacts of droughts on hydropower are shown in Supplementary Table S9, and the starting and ending times of the provinces affected by droughts are shown in Supplementary Table S10.”

“Impact mechanism of heat waves. Heat waves often have three effects on the near-zero power system. First, heat waves have similar effects from the hot weather of droughts, i.e., high

temperatures during heat waves can raise societal electricity demand, e.g., refrigeration equipment. The specific process by which heat waves affect electricity demand can be expressed by Equation (S23).

$$ED_{n,t}^{hw} = ED_{n,t} \cdot (1 + \Delta T_{n,t}^{hw} \cdot I^{ED}) \quad (S23)$$

where, $ED_{n,t}^{hw}$ is the hourly electricity demand at hour t in province n during the heat wave, and $\Delta T_{n,t}^{hw}$ is the temperature increment at hour t in province n during the heat wave compared to the year selected in the study.

Second, high temperatures cause photovoltaic panels to overheat, resulting in lower solar PV output efficiency. The specific process by which heat waves affect solar PV can be expressed by Equation (S24).

$$PV_{n,t}^{hw'} = \min(PV_{n,t}^{hw}, PV_n^{IC}) \quad (S24)$$

where, $PV_{n,t}^{hw'}$ is the actual solar PV power output at hour t in province n during the heat wave, and $PV_{n,t}^{hw}$ is the solar PV power output potential at hour t in province n at the corresponding photovoltaic panels temperature during the heat wave.

Third, heat waves also weaken wind speeds and associated wind power output. The specific process by which heat waves affect wind power can be expressed by Equation (S25).

$$WP_{n,t}^{hw'} = \min(WP_{n,t}^{hw}, WP_n^{IC}) \quad (S25)$$

where, $WP_{n,t}^{hw'}$ is the actual wind power output during the heat wave, and $WP_{n,t}^{hw}$ is the power output potential of the province at the corresponding wind speed during the heat wave."

6. In Line 735, KV should be revised as kV.

Authors' Response: Thank you for your careful reminding. We have double-checked the entire paper and replaced all "KV" to "kV" in our revised manuscript.

To Reviewer #3:

Thank you very much for inviting me to review the manuscript "An optimal solution to have a reliable and resilient power sector in China with near-zero carbon emissions". Strategies to have a reliable and resilient low-carbon power system are nowadays a hot topic whose impact on ensuring power system security and addressing climate change is remarkable. The manuscript focused on this real and interesting theme, which is sure of significance for power sector planning and climate goal achievement. In particular, the authors proposed a uniquely innovative modelling approach that integrated six interlinked modules (i.e., electricity demand prediction, variable renewable power potential assessment, source-sink optimal matching, power system simulation, cost-competitive analysis and weather extremes impact simulation) and developed a new assessment approach that can quantify the reliability and resilience of different power system by coupling the high share of

renewable power system with abated fossil fuel with CCUS technology, which contributes to a notable advance in integrated assessment modelling knowledge. By simulating numerous power system infrastructure scenarios, the authors provided a complete and detailed study for revealing the optimal solution to have a reliable and resilient power sector in China under the context of carbon neutrality.

Authors' Response: Thank you for your positive comments. We have fully responded to your major concerns and revised the manuscript. Please see the subsequent one-by-one responses for details.

Moreover, the modelling approach and analysis framework can be applied to all countries. Therefore, I think the study was conducted systematically under a rational work plan and presented meaningful results. I believe it can be accepted for publication in Nature Communications, and in principle, I recommend only four items could be addressed, as follows.

1. The introduction is well-written and easy to understand. The bibliographic research is complete and well-analyzed. The aims and methods are clearly described and supported by sufficient theoretical background. The current structure of the paper does not require changes.

Authors' Response: We are grateful to the reviewer for his/her positive feedback.

2. Figure 1 is outstanding. However, the legend parts are a bit small for the reader. Especially if the reader has some age, you could move this legend to the left side of Figure 1g or describe them in a separate space.

Authors' Response: Thank you for your careful suggestion. We have adjusted the relevant figure and separated the legend. The specific modifications are shown as follows.

Fig. 1 Daily and hourly variability of wind power and solar PV generation potential and predicted electricity demand in 2050. (a-i represent the North Coast, Northeast, East Coast, Beijing-Tianjin, South Coast, Northwest, Central, Southwest, and the whole mainland China (Nation), respectively. The cyan and orange curves in each panel represent wind power and solar PV generation potential, and the green

curves in each panel represent predicted electricity demand for each region in 2050, respectively. The left-to-right column for each region depicts daily variability, hourly variability in summertime (June, July, and August) and wintertime (December, January, and February) for power generation potential and predicted electricity demand, respectively. The lines represent the mean values, the dark shading represents the inner 50% of the observations (25th to 75th percentile) and the light shading represents the outer 50% of the observations (0th to 100th percentile) of the daily average value of that date in each year of the relevant observations, i.e., 1980-2019 for wind power and 2010-2019 for solar PV.”

3. The paper presented and explained all the key findings. The results and discussions were clearly written and easy to follow, and I could understand everything presented. There is one item that could be considered to enhance the contribution of this paper, i.e., one or two sentences about the findings of the optimal layout of coal-fired and natural gas power plants retrofitted with CCUS could be added in policy implications in the Discussion and Conclusions section.

Authors’ Response: Thank you for your suggestion. We have raised the optimal layout conclusion of coal-fired and natural gas power plants retrofitted with CCUS.

In the revised version, we have modified relevant descriptions in *Discussions and conclusions* as follows.

“Following the valid experiences from selected developed countries (e.g., 45Q credit in the US, carbon taxes in Norway), incentive policies for CCUS as well as biomass and coal co-firing with CCUS need be enhanced in China to promote their deployment in fossil fuel power generation facilities.”

4. The supplementary material is useful because it provides more insight into the results and case study sections. The authors are suggested to address one minor item in the supplementary figure, namely value one could be marked on the left vertical axis of Figure S3b, to better differentiate the provinces with power shortage from that with power surplus.

Authors’ Response: Thank you for your helpful suggestion. The Figure S3 has been corrected. The specific modifications are shown as follows.

“**Figure S3.** Comparison between non-fossil fuel power generation potential and electricity demand by province in 2050. (a represents the distribution of non-fossil fuel power generation potential and electricity consumption by province, respectively; wind power and solar PV generation potential is determined by summing up the real-time hourly power output. b represents the values of two gap indicators between non-fossil power generation potential and electricity demand; red triangles represent the ratio of total non-fossil power generation potential to total electricity consumption in each province (all hours cumulative); purple squares represent the provincial entire power shortage rate when electricity supply relying solely on non-fossil fuel without dispatch and energy storage.)”

References

1. Heuberger CF, Rubin ES, Staffell I, Shah N, Mac Dowell N. Power capacity expansion planning considering endogenous technology cost learning. *Applied Energy* **204**, 831-845 (2017).
2. Daggash HA, Mac Dowell N. Structural Evolution of the UK Electricity System in a below 2°C World. *Joule* **3**, 1239-1251 (2019).
3. Chen X, *et al.* Pathway toward carbon-neutral electrical systems in China by mid-century with negative CO₂ abatement costs informed by high-resolution modeling. *Joule* **5**, 2715-2741 (2021).
4. Zhuo Z, *et al.* Cost increase in the electricity supply to achieve carbon neutrality in China. *Nature Communications* **13**, 3172 (2022).
5. Riera JA, Lima RM, Hoteit I, Knio O. Simulated co-optimization of renewable energy and desalination systems in Neom, Saudi Arabia. *Nature Communications* **13**, 3514 (2022).
6. Heuberger CF, Rubin ES, Staffell L, Shah N, Mac Dowell N. Power capacity expansion planning considering endogenous technology cost learning (vol 204, pg 831, 2017). *Applied Energy* **220**, 974-974 (2018).
7. Gonzalez JM, *et al.* Designing diversified renewable energy systems to balance multisector performance. *Nature Sustainability* (2023).
8. Gernaat DEHJ, de Boer HS, Daioglou V, Yalew SG, Müller C, van Vuuren DP. Climate change impacts on renewable energy supply. *Nature Climate Change* **11**, 119-125 (2021).
9. Joshi S, Mittal S, Holloway P, Shukla PR, Ó Gallachóir B, Glynn J. High resolution global spatiotemporal assessment of rooftop solar photovoltaics potential for renewable electricity generation. *Nature Communications* **12**, 5738 (2021).
10. Feng K, Ouyang M, Lin N. Tropical cyclone-blackout-heatwave compound hazard resilience in a changing climate. *Nature Communications* **13**, 4421 (2022).
11. Perera ATD, Nik VM, Chen D, Scartezzini J-L, Hong T. Quantifying the impacts of climate change and extreme climate events on energy systems. *Nature Energy* **5**, 150-159 (2020).
12. Kuepper LE, Teichgraeber H, Baumgärtner N, Bardow A, Brandt AR. Wind data introduce error in time-series reduction for capacity expansion modelling. *Energy* **256**, 124467 (2022).
13. Kotzur L, Markewitz P, Robinius M, Stolten D. Impact of different time series aggregation methods on optimal energy system design. *Renewable Energy* **117**, 474-487 (2018).
14. Frew BA, Jacobson MZ. Temporal and spatial tradeoffs in power system modeling with assumptions about storage: An application of the POWER model. *Energy* **117**, 198-213 (2016).
15. Klemm C, Wiese F, Vennemann P. Model-based run-time and memory reduction for a mixed-use

- multi-energy system model with high spatial resolution. *Applied Energy* **334**, 120574 (2023).
16. Webster M, Zhao B, Bukenberger J, Blumsack S. Transition to Low-Carbon Electric Power: Portfolios, Flexibility, and Option Value. *Environmental Science & Technology* **56**, 9583-9592 (2022).
 17. Major Power Outage Events - 2022 December winter storm. <https://poweroutage.us/about/majorevents> (PowerOutage.US, 2022).
 18. China's Hydrogen Energy and Fuel Cell Industry White Paper 2020. http://www.h2cn.org.cn/dynamics_detail/787.html (China Hydrogen Alliance, 2021).
 19. Opportunities for Hydrogen Production with CCUS in China. <https://www.iea.org/reports/opportunities-for-hydrogen-production-with-ccus-in-china> (International Energy Agency, 2022).
 20. Renewable Energy Data Sheet 2019. (China National Renewable Energy Center, 2019).
 21. Liu L, *et al.* Potential contributions of wind and solar power to China's carbon neutrality. *Resources, Conservation and Recycling* **180**, 106155 (2022).
 22. Bistline JET, Young DT. The role of natural gas in reaching net-zero emissions in the electric sector. *Nature Communications* **13**, 4743 (2022).
 23. Duan H, *et al.* Assessing China's efforts to pursue the 1.5 C warming limit. *Science* **372**, 378-385 (2021).
 24. Zhang S, Chen W. Assessing the energy transition in China towards carbon neutrality with a probabilistic framework. *Nature Communications* **13**, 87 (2022).
 25. Fan J-L, Xu M, Li F, Yang L, Zhang X. Carbon capture and storage (CCS) retrofit potential of coal-fired power plants in China: The technology lock-in and cost optimization perspective. *Applied Energy* **229**, 326-334 (2018).
 26. Mercure JF, *et al.* Macroeconomic impact of stranded fossil fuel assets. *Nature Climate Change* **8**, 588-593 (2018).
 27. Sgouridis S, Carbajales-Dale M, Csala D, Chiesa M, Bardi U. Comparative net energy analysis of renewable electricity and carbon capture and storage. *Nature Energy* **4**, 456-465 (2019).
 28. Fernández-Guillamón A, Gómez-Lázaro E, Muljadi E, Molina-García Á. Power systems with high renewable energy sources: A review of inertia and frequency control strategies over time. *Renewable and Sustainable Energy Reviews* **115**, 109369 (2019).
 29. Kontis EO, Pasiopoulou ID, Kirykos DA, Papadopoulos TA, Papagiannis GK. Estimation of power system inertia: A Comparative assessment of measurement-Based techniques. *Electric Power Systems Research* **196**, 107250 (2021).
 30. Li M, Virguez E, Shan R, Tian J, Gao S, Patiño-Echeverri D. High-resolution data shows China's wind and solar energy resources are enough to support a 2050 decarbonized electricity system. *Applied Energy* **306**, 117996 (2022).
 31. Ding S, Zeng P, Xing H, Yang J, Zhou Q. A medium and long term multi-objective optimal operation method for integrated wind/PV/hydro power. *Electric Power Science and Engineering* **35**, 17-25 (2019).
 32. PVWatts Version 5 Manual. (National Renewable Energy Lab. (NREL), 2014).
 33. Notice of the National Development and Reform Commission on the signing of medium- and long-term contracts for electric power in 2020. https://www.ndrc.gov.cn/xxgk/zcfb/tz/201912/t20191230_1216857_ext.html (National Development and Reform Commission, 2019).

34. In-depth report on the energy storage industry: the development of six types of energy storage and their economic assessment. <https://www.vzkoo.com/document/202205111b2af51c59e3228545236eab.html> (State Grid Yingda Group, 2022).
35. Santos R, Sgouridis S, Alhajaj A. Potential of CO₂-enhanced oil recovery coupled with carbon capture and storage in mitigating greenhouse gas emissions in the UAE. *International Journal of Greenhouse Gas Control* **111**, 103485 (2021).
36. China Carbon Capture, Utilization and Storage (CCUS) Annual Report (2021). (Chinese Academy of Environmental Planning; Institute of Rock and Soil Mechanics, Chinese Academy of Sciences; Management Centre of Agenda in the 21st Century, 2021).
37. 2021 China Wind Power Lifting Capacity Statistics Brief. (2022).
38. Lu T, Sherman P, Chen X, Chen S, Lu X, McElroy M. India's potential for integrating solar and on- and offshore wind power into its energy system. *Nature Communications* **11**, 4750 (2020).
39. Davidson Michael R, Zhang D, Xiong W, Zhang X, Karplus Valerie J. Modelling the potential for wind energy integration on China's coal-heavy electricity grid. *Nature Energy* **1**, 16086 (2016).
40. Hasan A, Dincer I. A new performance assessment methodology of bifacial photovoltaic solar panels for offshore applications. *Energy Conversion and Management* **220**, 112972 (2020).
41. DIFFERENCE BETWEEN BIFACIAL AND MONOFACIAL SOLAR PANELS: WHICH IS BETTER? <https://www.solarsquare.in/blog/monofacial-solar-panels/> (2022).
42. Utility-Scale Solar Photovoltaic Power Plants: A Project Developer's Guide. https://www.ifc.org/wps/wcm/connect/a1b3dbd3-983e-4ee3-a67b-cdc29ef900cb/IFC+Solar+Report_Web+_08+05.pdf?MOD=AJPERES&CVID=kZePDPG (2015).
43. Chen S, *et al.* The Potential of Photovoltaics to Power the Belt and Road Initiative. *Joule* **3**, 1895-1912 (2019).
44. Lu X, *et al.* Combined solar power and storage as cost-competitive and grid-compatible supply for China's future carbon-neutral electricity system. **118**, e2103471118 (2021).
45. 2022 electricity annual technology baseline data. <https://atb.nrel.gov/electricity/2022/data> (2022).
46. GenCost 2018: Updated projections of electricity generation technology costs. <https://publications.csiro.au/rpr/download?pid=csiro:EP189502&dsid=DS1> (2018).
47. Future of Solar Photovoltaic. https://www.irena.org/-/media/Files/IRENA/Agency/Publication/2019/Nov/IRENA_Future_of_Solar_PV_2019.pdf?rev=d2e0fb395422440bbeb74c69bbe2dc99 (2019).
48. He G, Lin J, Sifuentes F, Liu X, Abhyankar N, Phadke A. Rapid cost decrease of renewables and storage accelerates the decarbonization of China's power system. *Nature Communications* **11**, 2486 (2020).
49. He G, *et al.* SWITCH-China: A Systems Approach to Decarbonizing China's Power System. *Environmental Science & Technology* **50**, 5467-5473 (2016).
50. Tong D, *et al.* Geophysical constraints on the reliability of solar and wind power worldwide. *Nature communications* **12**, 6146 (2021).
51. "Ten Commitments" of Power Supply Service of State Grid Corporation of China. <http://www.sc.sgcc.com.cn/html/files/2022-03/25/20220325152015177346621.pdf> (State Grid Corporation of China, 2022).
52. Bie Z, Lin Y, Li G, Li F. Battling the extreme: A study on the power system resilience. *Proceedings of the IEEE* **105**, 1253-1266 (2017).

53. Zeyringer M, Price J, Fais B, Li P-H, Sharp E. Designing low-carbon power systems for Great Britain in 2050 that are robust to the spatiotemporal and inter-annual variability of weather. *Nature Energy* **3**, 395-403 (2018).
54. Bennett JA, *et al.* Extending energy system modelling to include extreme weather risks and application to hurricane events in Puerto Rico. *Nature Energy* **6**, 240-249 (2021).
55. Electricity supply business rules. http://www.nea.gov.cn/2012-01/04/c_131262676.htm (National Energy Administration, 2012).
56. Dong B, Jiang X, Yin F. Development and prospect of monitoring and prevention methods of icing disaster in China power grid. *IET Generation, Transmission & Distribution* **16**, 4480-4493 (2022).
57. Llasat MC, Turco M, Quintana-Seguí P, Llasat-Botija M. The snow storm of 8 March 2010 in Catalonia (Spain): a paradigmatic wet-snow event with a high societal impact. *Nat Hazards Earth Syst Sci* **14**, 427-441 (2014).
58. Busby JW, *et al.* Cascading risks: Understanding the 2021 winter blackout in Texas. *Energy Research & Social Science* **77**, 102106 (2021).
59. More than 90,000 people affected by snowstorm in Ensh, Hubei Province, with serious damage to power facilities. http://www.gov.cn/xinwen/2016-11/11/content_5131428.htm (Chinese government website, 2016).
60. Power supply restored to 110,000 homes in Sichuan after snowstorm. http://www.gov.cn/xinwen/2016-02/24/content_5045383.htm (Chinese government website, 2016).
61. Haerberli W, Whiteman C. Snow and ice-related hazards, risks, and disasters: Facing challenges of rapid change and long-term commitments. In: *Snow and ice-related hazards, risks, and disasters*. Elsevier (2021).
62. IPCC, 2021: Climate Change 2021: The Physical Science Basis. Contribution of Working Group I to the Sixth Assessment Report of the Intergovernmental Panel on Climate Change. <https://www.ipcc.ch/report/ar6/wg1/about/how-to-cite-this-report/> (2021).

REVIEWER COMMENTS

Reviewer #1 (Remarks to the Author):

The authors have done an outstanding work in addressing the comments of the reviewers. All my major concerns have been addressed. While I may not fully agree with some of the authors' choices regarding certain assumptions (e.g. solar system technology choice, average size of future wind turbines etc.) these are made with a reasoned approach and in the broader scheme would not have an impact on the results.

There are two minor points that could be improved in my view, or at least explained better:

1. Fig. 2 pg 11, has as y-axis title: "Times of Reference Dispatch Capacity". This title is confusing especially when in Ln 226, it is referred to as "reference transmission capacity".

Recommend to keep one term (transmission) and name the y-axis and figure caption accordingly.
Recommend: "Reference Transmission Capacity Factor"

2. Using the Anhui province as reference for reconstructing the hourly demand of the other provinces is a bit surprising to this reviewer (would have expected that this data should be available). Since, evidently they are not, it would be at least a good practice to time-align the demand of other provinces with the local time so that the overall effect is more realistic (e.g. when demand peaks in the East during noon, the solar output of the West would be low being a few hours delayed).

Reviewer #2 (Remarks to the Author):

The author has made modifications, and I have no other suggestions for modification.

Responses to Reviewers' Comments

To Reviewer #1:

The authors have done an outstanding work in addressing the comments of the reviewers. All my major concerns have been addressed. While I may not fully agree with some of the authors' choices regarding certain assumptions (e.g. solar system technology choice, average size of future wind turbines etc.) these are made with a reasoned approach and in the broader scheme would not have an impact on the results.

Authors' Response: Thank you for your positive comments. We have fully responded to your concerns and further revised the manuscript. Please see the subsequent one-by-one responses for details.

There are two minor points that could be improved in my view, or at least explained better:

1. Fig. 2 pg 11, has as y-axis title: "Times of Reference Dispatch Capacity". This title is confusing especially when in Ln 226, it is referred to as "reference transmission capacity".

Recommend to keep one term (transmission) and name the y-axis and figure caption accordingly.

Recommend: "Reference Transmission Capacity Factor"

Authors' Response: Thank you for your careful comments. Based on your suggestion, we have modified the y-axis titles of both Figure 2 in the main text and Figure S6 in the supplement. In addition, we have kept the term statement "transmission" with necessary changes in the main text.

In the revised version, we have revised Figure 2 and Figure S6 in *Results* and in *Supplementary Information* respectively, as follows.

Fig. 2 The relationship between national entire power shortage rates, transmission capacity, and short-term energy storage hours.

Figure S6. Power shortage and cost comparison under 16% abated fossil fuel and zero fossil fuel with long-term energy storage scenarios.

2. Using the Anhui province as reference for reconstructing the hourly demand of the other provinces is a bit surprising to this reviewer (would have expected that this data should be available). Since, evidently they are not, it would be at least a good practice to time-align the demand of other provinces with the local time so that the overall effect is more realistic (e.g. when demand peaks in the East during noon, the solar output of the West would be low being a few hours delayed).

Authors' Response: Thank you for your valuable comment and we apologize that we did not explain this point clearly in last version. Indeed, due to data availability, we could only obtain hourly electricity consumption for Anhui Province in 2019 and estimate that of other provinces based on the characteristics of Anhui province combining with other various information, as shown in section of *Provincial real-time hourly electricity demand predictions* in *Methods*.

However, it should be noted that in calculating hourly electricity demand for other provinces, we not only adopt the daily variations of Anhui province as reference but also refer to the actual hourly electricity load for typical workdays and non-workdays for 30 provinces themselves in China,

which are used to simulate the hourly variation characteristics of daily electricity demand for different provinces and refer to the actual monthly electricity consumption to capture the monthly variation characteristics of each province. Therefore, the calculated hourly electricity demand for each day in different provinces could be consistent with their local time (e.g., the daily peak of electricity consumption in the western region will occur later than that of the eastern region). To facilitate the reader's understanding, we additionally present one representative province from each of the eight regions in this study, as shown in Figure S13.

Furthermore, since the variable renewable power output potential in each province is calculated using actual local climatic conditions at different hours, the situation you mentioned would indeed appear in our analysis (e.g., when electricity demand peaks in the East during noon, the solar output in the West is low due to a few hours delay).

In summary, our analysis has addressed your concerns by capturing the hourly variation characteristics of each day both for provincial electricity consumption and variable renewable power output potential that are consistent with the local time scale of each province.

In the revised version, we have added Figure S12 in *Supplementary Figures* as follows and cited it in the *Method* section.

Figure S12. Hourly electricity load curves for typical workdays and non-workdays in representative provinces of eight regions.

To Reviewer #2:

The author has made modifications, and I have no other suggestions for modification.

Authors' Response: Thank you. We appreciate your valuable suggestions in this revision process which are essential to improve the quality of our paper.